

# Simulating marine neodymium isotope distributions using ND v1.0 coupled to the ocean component of the FAMOUS-MOSES1 climate model: sensitivities to reversible scavenging efficiency and benthic source distributions

Suzanne Robinson[1], Ruza Ivanovic[1], Lauren Gregoire[1], Julia Tindall[1], Tina van de Flierdt[2], Yves Plancherel[2], Frerk Pöppelmeier[3], Kazuyo Tachikawa[4], Paul Valdes[5]

[1]School of Earth and Environment, University of Leeds, Leeds, LS2 9JT, UK
[2]Department of Earth Science and Engineering, Imperial College London, London, SW7 2AZ, UK
[3]Climate and Environmental Physics, Physics Institute and Oeschger Center for Climate Change Research, University of
Bern, 3012 Bern, Switzerland
[4]Aix Marseille Univ, CNRS, IRD, INRAE, Coll France, CEREGE, Aix-en-Provence, France
[5]School of Geographical Sciences, University of Bristol, Bristol, UK

*Correspondence to*: Suzanne Robinson (ee14s2r@leeds.ac.uk)

**Abstract.** The neodymium (Nd) isotopic composition of seawater is a widely used ocean circulation tracer. However,
uncertainty in quantifying the global ocean Nd budget, particularly constraining elusive non-conservative processes, remains a major challenge. A substantial increase in modern seawater Nd measurements from the GEOTRACES programme coupled with recent hypotheses that a seafloor-wide benthic Nd flux to the ocean may govern global Nd isotope distributions ($\varepsilon_{Nd}$) presents an opportunity to develop a new scheme specifically designed to test these paradigms. Here, we present the implementation of Nd isotopes ([143]Nd and [144]Nd) into the ocean component of the FAMOUS coupled atmosphere-ocean general
circulation model (ND v1.0), a tool which can be widely used for simulating complex feedbacks between different Earth system processes on decadal to multi-millennial timescales.

Using an equilibrium pre-industrial simulation tuned to represent the largescale Atlantic Ocean circulation, we perform a series of sensitivity tests evaluating the new Nd isotope scheme. We investigate how Nd source/sink and cycling parameters govern
global marine $\varepsilon_{Nd}$ distributions, and provide an updated compilation of 6,048 Nd concentration and 3,278 $\varepsilon_{Nd}$ measurements to assess model performance. Our findings support the notions that reversible scavenging is a key process for enhancing the Atlantic-Pacific basinal $\varepsilon_{Nd}$ gradient, and is capable of driving the observed increase in Nd concentration along the global circulation pathway. A benthic flux represents a major source of Nd to the deep ocean. However, model-data disparities in the North Pacific highlight that the source of $\varepsilon_{Nd}$ from seafloor sediment is too unradiogenic in our model with a constant benthic
flux. Additionally, model-data mismatch in the northern North Atlantic suggests a missing source of Nd that is much more unradiogenic than the bulk sediment, alluding to the possibility of preferential contributions from 'reactive' detrital sediments under a benthic flux driven model of marine Nd cycling.





The new Nd isotope scheme forms an excellent tool for exploring global marine Nd cycling and the interplay between climatic

and oceanographic conditions under both modern and palaeoceanographic contexts.

# 1 Introduction

The Neodymium (Nd) isotope composition of seawater shows a clear provinciality between different ocean basins and is often used as a water mass provenance tracer (e.g., Frank, 2002; Goldstein and Hemming, 2003). The measured $^{143}Nd/^{144}Nd$ ratio is denoted relative to the bulk earth standard:

$$\varepsilon_{Nd} = \left( \frac{(^{143}Nd/^{144}Nd)_{sample}}{(^{143}Nd/^{144}Nd)_{CHUR}} - 1 \right) \times 10^4 \, , \tag{1}$$

where $(^{143}Nd/^{144}Nd)_{CHUR}$ relates to the Chondritic Uniform Reservoir (CHUR; 0.512638: Jacobsen and Wasserburg, 1980). Distinct variations in the Nd isotope signal of water masses originate from different continental regions and their isotopic fingerprints, and subsequent influence by ocean circulation, water mass mixing and particle cycling, as well as interaction with sediments (e.g. Tachikawa et al., 2017; van de Flierdt et al., 2016 for recent reviews). Neodymium in the deep ocean

has a residence time that is shorter than the global overturning of the deep ocean (Arsouze et al., 2009; Rempfer et al., 2011; Gu et al., 2019; Pöppelmeier et al., 2020a, 2021b; Tachikawa et al., 2003). Unlike other tracers of ocean circulation (e.g. $\delta^{13}C$, $\Delta^{14}C$), the measured Nd isotope composition of seawater is not actively involved marine biological cycling, giving rise to its promise as a carbon cycle independent ocean circulation tracer (Blaser et al., 2019a). Yet, a fundamental caveat in the application of $\varepsilon_{Nd}$ as a reliable oceanographic tracer is that a universal understanding of the exact mechanisms controlling

marine geochemical Nd cycling remains incomplete (Abbott et al., 2015a; Haley et al., 2017; van de Flierdt et al., 2016).

Numerical models are useful tools for investigating Nd cycling since they can specify the processes that govern the spatiotemporal variability in Nd isotope distributions in the ocean. Neodymium isotopes have been simulated in a range of different modelling studies testing specific hypotheses relating to Nd fluxes and thermohaline redistribution (Ayache et al.,

2016; Pöppelmeier et al., 2020a; Rempfer et al., 2011; Siddall et al., 2008; Jones et al., 2008; Roberts et al., 2010; Tachikawa et al., 2003; Arsouze et al., 2009; Gu et al., 2019; Oka et al., 2021, 2009; Pöppelmeier et al., 2022; Ayache et al., 2022; Pasquier et al., 2021; Du et al., 2020). However, recent work suggests that a seafloor wide benthic flux, resulting from early diagenetic reactions, may dominate the marine Nd cycle (Haley et al., 2017; Abbott, 2019; Abbott et al., 2015a, b, 2019; Du et al., 2016). These observations, alongside an ever-growing body of high-quality and highly-resolved measurements of dissolved seawater

Nd concentrations ([Nd]) and $\varepsilon_{Nd}$ from the GEOTRACES programme (GEOTRACES Intermediate Data Product Group, 2021), present an opportunity to re-evaluate, revise and explore constraints on the marine Nd cycle.



Initially, the predominant lithogenic fluxes of Nd to the ocean were believed to be only at the surface (aeolian dust and riverine fluxes; Goldstein et al., 1984; Goldstein and Jacobsen, 1987). Early modelling studies applying surface fluxes reproduced reasonable $\varepsilon_{Nd}$ in the North Atlantic (Bertram and Elderfield, 1993; Tachikawa et al., 1999). However, considering only dust
and river fluxes alone led to an unrealistic calculated residence time of Nd in seawater on the order of 5,000 years (Bertram and Elderfield, 1993; Jeandel et al., 1995). Through applying a simple box model to calculate the oceanic Nd budget, it was then found that considering dust and river surface inputs alone failed to balance both [Nd] and $\varepsilon_{Nd}$, thus indicating a 'missing source' of Nd to the ocean that accounted for $\approx 90\%$ of the Nd flux to the ocean (Tachikawa et al., 2003). This led to a new hypothesis relating to other Nd sources to seawater that could account for this 'missing source', including submarine
groundwater discharge (SGD) (Johannesson and Burdige, 2007), and input from the dissolution of sediment deposited on the continental margins (Lacan and Jeandel, 2005). The term 'boundary exchange' was coined to describe strong Nd isotopic interactions between continental margins and water masses though the co-occurrence of sediment dissolution and boundary scavenging (Lacan and Jeandel, 2005). Arsouze et al. (2007) simulated realistic global $\varepsilon_{Nd}$ distributions using boundary exchange as the only source-sink term, and since then, boundary exchange along continental margins has represented the major
flux of Nd to seawater in recent global Nd isotope models (Arsouze et al., 2009; Rempfer et al., 2011; Gu et al., 2019).

Nonetheless, boundary exchange alone cannot fully reconcile the global marine Nd cycle. Specifically, it cannot explain the observed vertical profiles of [Nd], which are decoupled from $\varepsilon_{Nd}$ (i.e., the 'Nd paradox': Goldstein and Hemming, 2003), with low concentrations near the surface increasing with depth. This is a common characteristic of isotopes/elements (e.g. thorium) that are reversibly scavenged (i.e., where the element is scavenged onto sinking particles at the surface and is subsequently
remineralised in the deep ocean) (Bertram and Elderfield, 1993; Bacon and Anderson, 1982). Siddall et al. (2008) first addressed numerically a hypothesis that the 'Nd paradox' can be explained by a combination of lateral advection and reversible scavenging by applying the reversible scavenging model pioneered by Bacon and Anderson (1982) to Nd cycling. In their study, both [Nd] and $\varepsilon_{Nd}$ were modelled simultaneously and explicitly to explore internal cycling of Nd in the ocean. Their findings demonstrated that scavenging and remineralisation processes are important active components in the marine cycling
of Nd, driving the increase of [Nd] with depth, but still allowing $\varepsilon_{Nd}$ to act as an effective water mass tracer.

Although inclusion of reversible scavenging can explain aspects of marine Nd cycling, the use of over-simplified fixed surface [Nd] and $\varepsilon_{Nd}$ boundary conditions in the model by Siddall et al. (2008) limited what could be determined about the full marine cycling of Nd and hence the 'Nd paradox'. The most comprehensive Nd isotope enabled ocean models to date now explicitly represent and quantify a wider range of distinct Nd fluxes that are both external and internal to the marine realm. For example,
Arsouze et al. (2009) used a fully prognostic coupled dynamic and biogeochemical model to simulate [Nd] and $\varepsilon_{Nd}$, considering dust fluxes, dissolved riverine sources, boundary exchange and reversible scavenging. In their study, a boundary source from the continental margins represented the major source of Nd to seawater ($\approx 95\%$ of the total source). Rempfer et al. (2011) continued this work, undertaking a more detailed and comprehensive investigation of Nd sources and particle scavenging using



a coarse resolution intermediate complexity model (Bern3D ocean model) and extensive sensitivity experiments. This later scheme was then closely followed by Gu et al. (2019) for the implementation of Nd isotopes in the ocean component of a more comprehensive Earth System Model (ESM, specifically the Community Earth System Model: CESM1.3) to explore in detail the changes to end-member $\varepsilon_{Nd}$ signatures in response to ocean circulation and climate changes. Overall, these comprehensive models, capable of quantifying the major sources implicated in marine Nd cycling indicated that dust and river fluxes were important for representing [Nd] and $\varepsilon_{Nd}$ distributions in the surface, but that the main flux of Nd to seawater is via a boundary source operating along the continental margins.

Recent pore fluid concentration profiles measured on the Oregon margin in the Pacific Ocean indicate that there may be a *benthic* flux of Nd from sedimentary pore fluids, presenting a new, potentially major seafloor-wide source of Nd to seawater (Abbott et al., 2015b, a). Additional measurements from the Tasman Sea suggest the likely presence of a benthic source of similar magnitude to that inferred for the North Pacific, which may indicate that regions with dominantly calcareous sediment also contribute a significant benthic source of Nd (Abbott, 2019). Evidence of this previously overlooked abyssal benthic sedimentary source of Nd has led to a shifting paradigm that challenges the current 'top down' model of marine Nd cycling to one of a 'bottom up' model (Haley et al., 2017). The bottom up model contends that the dominant addition of Nd to the ocean is from a diffuse sedimentary source at depth, rather than surface point sources from rivers and dust and the shallow continental margins ('top down'). The benthic flux hypothesis provides a compelling yet unproved mechanism to explain deep water $\varepsilon_{Nd}$ alteration alongside vertical [Nd] gradients in the North Pacific in the absence of modern deep-water formation, via exposure of old bottom water to a substantial benthic flux (Abbott et al., 2015a; Du et al., 2016).

Simple box models have been employed to investigate, to a first order, the non-conservative effects from a benthic flux (Du et al., 2016, 2018, 2020; Haley et al., 2017; Pöppelmeier et al., 2020b), suggesting overprinting of deep water mass $\varepsilon_{Nd}$ is linked to benthic flux exposure time and the difference between the Nd isotope composition of the benthic flux and the bottom water (Abbott et al., 2015a; Du et al., 2018). However, these models lack comprehensive descriptions of both the marine Nd cycle and of physical ocean circulation and climate interactions, limiting a clear interpretation of precisely how (and under what physical/environmental conditions) the benthic flux may determine global marine Nd distributions. Applying an intermediate complexity model, Pöppelmeier et al. (2021) investigated the benthic flux hypothesis in more detail by updating the Nd isotope enabled Bern3D model (Rempfer et al., 2011) to represent recent observations that indicate a Nd flux from bottom waters could occur across the entire seafloor. This was done by removing the depth limitation of the boundary exchange (previously 3 km) and invoking a constant benthic flux that escapes from all sediment-water interfaces. The scheme was further extended by revising key source-sink parameterisations, for subsequent investigation of non-conservative Nd isotope behaviour under different ocean circulation states (Pöppelmeier et al., 2022). The authors demonstrate substantial non-conservative effects occur even under strong circulation regimes with low benthic flux exposure times, and are not strictly limited to the deep ocean. This work highlights the importance of downward vertical fluxes via reversible scavenging alongside the benthic flux



to describe non-conservative marine Nd isotope behaviour. Nonetheless, the low horizontal resolution of the intermediate complexity model limits full resolution of key circulation patterns such as distinct deep-water formation in the Labrador and in the Nordic Seas, inhibiting the capabilities of the scheme to fully capture and explore water mass end member $\varepsilon_{Nd}$ distributions.

Thus, despite substantial progress to explicitly describe seawater Nd budgets, it is apparent that outstanding questions remain, alongside divergent lines of argument amongst subject specialists, limiting a full, quantitative description of marine Nd cycling. The decoupled nature of marine [Nd] and $\varepsilon_{Nd}$, which is yet to be fully understood, and new emerging observations (e.g., Stichel et al., 2020) emphasise the critical need to progress our understanding of modern marine Nd cycling, in the light of continued use of $\varepsilon_{Nd}$ as a valuable tracer of ocean circulation. In this context, Nd isotope enabled ocean models remain an effective way

to progress with testing and constraining processes governing marine Nd cycling, which can then feedback key information to the wider GEOTRACES and ocean-tracer modelling community.

With this is mind, there is a current gap in the toolkit for modelling marine Nd cycling between the class of high complexity, state-of-the-art ESMs (e.g., Gu et al., 2019) and the more efficient intermediate complexity models (e.g., Rempfer et al., 2011; Pöppelmeier et al., 2020a). To bridge this gap, there is a need for a model with the full complexity of an Atmosphere-Ocean

General Circulation Model (AOGCM), allowing the exploration of how Nd isotopes vary under changing climate conditions (including extensive palaeoceanographic applications), that is also capable of running very quickly to facilitate efficient parameter space exploration, performance optimisation and long integrations. The FAMOUS GCM fills this gap (Smith et al., 2008; Jones et al., 2005; Smith, 2012; Jones et al., 2008).

Here we present the new global marine Nd isotope scheme (ND v1.0) implemented in FAMOUS. We utilise the new sediment

$\varepsilon_{Nd}$ maps from Robinson et al. (2021) as boundary conditions for a mobile global sediment Nd flux, with the end-goal of further constraining the major sources, sinks and cycling of Nd isotopes, and exploring instances of non-conservative behaviour related to changes in Nd source distributions and scavenging processes. We develop sensitivity experiments (Sect. 3) to isolate the physical effects of varying two key parameters that detail major Nd fluxes and cycling to the global ocean, verifying foremost that the new isotope scheme is responding to Nd source/sink and biogeochemical processes as expected, and contextualising

how, and where, reversible scavenging processes and the benthic flux can govern marine [Nd] and $\varepsilon_{Nd}$ distributions. Finally (Sect. 3 and 4), we evaluate overall model performance through comparison with modern measurements and assess the model's ability to simulate observed spatial and vertical gradients between ocean basins, encompassing underling structural uncertainty/model bias.



## 2 Methods

**2.1 Model description**

We use the FAMOUS GCM  (Smith et al., 2008; Jones et al., 2005; Smith, 2012; Jones et al., 2008), a fast coupled AOGCM, derived from the Met Office's Hadley Centre Coupled Model V3 (HadCM3) AOGCM (Gordon et al., 2000). The atmospheric component of FAMOUS is based on quasi-hydrostatic primitive equations and has a horizontal resolution of 5° latitude by 7.5° longitude, 11 vertical levels on a hybrid sigma-pressure coordinate system and a 1-hour timestep. The ocean

component is a rigid-lid model, with a horizontal resolution of 2.5° latitude by 3.75° longitude and 20 vertical levels, spaced unequally in thickness from 10 m at the near-surface ocean to over 600 m at depth, and a 12-hour timestep. The ocean and atmosphere are coupled once per day.

FAMOUS's parameterisations of physical and dynamical processes are almost identical to those of HadCM3, but it has approximately half the spatial resolution and a longer timestep, allowing it to run ten times faster. Thus, FAMOUS achieves a

current speed of up to 650 model years per wall clock day on 16 processors, making it ideal for running large ensembles (Gregoire et al., 2011, 2016), more bespoke sensitivity studies (Smith and Gregory, 2009; Gregoire et al., 2015), and multi-millennial simulations, e.g. to examine ice sheet behaviours (Gregoire et al., 2012, 2016, 2015), ocean drifts (Dentith et al., 2019) and biogeochemistry (Dentith et al., 2020).

We added Nd isotopes ($^{143}$Nd and $^{144}$Nd) as optional passive tracers into the ocean component of FAMOUS, using a version

of the model with the Met Office Surface Exchange Scheme (MOSES) version 1 (Cox et al., 1999:  FAMOUS-MOSES1). It was a pragmatic choice to avoid the more recent FAMOUS-MOSES2.2 configuration (Essery and Clark, 2003; Valdes et al., 2017; Williams et al., 2013; Essery et al., 2001), because evaluation of our new Nd scheme would be hindered by the collapsed Atlantic Ocean convection and strong deep Pacific MOC produced by FAMOUS-MOSES2.2 under pre-industrial boundary conditions (see Dentith et al., 2019). Nonetheless, the Nd isotope code implementation presented here is directly

transferable to other versions of the UK Met Office Unified Model (UM) version 4.5, including HadCM3/L or FAMOUS-MOSES2.2.

**2.2 A new reference simulation for this study**

The use of Nd isotopes as a water provenance tracer comes from measurements of distinct $\varepsilon_{Nd}$ signatures in different water masses. This is perhaps best demonstrated in the south Atlantic 'zig-zag' depth profiles, where $\varepsilon_{Nd}$ displays large heterogeneity

and distinguishes southward flowing North Atlantic Deep Water (NADW) from northward flowing Antarctic Intermediate Water (AAIW) and Antarctic Bottom Water (AABW)  (Goldstein and Hemming, 2003; Jeandel, 1993). As such, it is desirable to implement the Nd isotope scheme in a version of FAMOUS that best positions these basinal water masses in the correct locations.



The standard pre-industrial FAMOUS setup (XFHCC; Smith, 2012) has certain known limitations, including over-ventilated

abyssal Atlantic waters characterised by strong, over-deep NADW formation, with 'North Atlantic' convection occurring only in the Norwegian-Greenland Sea (there is no deep water formation in the Labrador Sea), and insufficient Atlantic sector AABW formation (Dentith et al., 2019; Smith, 2012). This known physical bias would dominate simulated Nd distributions, thus hampering validation of the new Nd isotope scheme against modern measurements. To mitigate this, we chose to employ a new reference simulation with improved basin-scale physical ocean circulation. This simulation was obtained from a perturbed

parameter ensemble varying 13 physical tuning parameters (see Supplementary Information: Text S1 and Table S1 for brief description and for a list of perturbed parameters from this multi-sweep ensemble of FAMOUS MOSES1: Smith, 1990, 1993; Crossley and Roberts, 1995). We screened the resulting 549 simulations based on a set of pre-defined targets, with a particular focus on Atlantic Ocean structure and water mass composition. Specifically, we sought a simulation with appropriate modern AMOC strength (14-19 Sverdrup, Sv; where $1 \text{ Sv} = 10^6 \text{ m}^3 \text{ s}^{-1}$) (Frajka-Williams et al., 2019), AMOC structure (Talley et al.,

2011), regions of AMOC convection as indicated by mixed layer depth (specifically in both the Labrador Sea and the Nordic Seas as these represent key regions for NADW formation and the resultant end member Nd isotope compositions; Lambelet et al., 2016), depth of maximum overturning ($\approx 1,000$ m), and presence of AABW in the abyssal Atlantic (Frajka-Williams et al., 2019; Talley et al., 2011; Kuhlbrodt et al., 2007; Ferreira et al., 2018). Furthermore, we assessed the capabilities of the simulation to represent appropriate water mass structure in the Pacific (Talley et al., 2011). Through this approach, we

identified four possible candidate simulations from the large ensemble as the basis for the new Nd isotope scheme: XPDAA, XPDAB, XPDAC and XPDEA, here denoted by their unique five-letter Met Office UM identifier. See Supplementary Information: Table S2 for initial boundary conditions, and Figures S1-4 for Atlantic meridional stream function and mixed layer depth for the four experiments.

These simulations were integrated for a further 5,000 years to ensure adequate spin-up of the physical circulation. They were

then assessed for their ability to simulate global modern observations of salinity and temperature (compared to the NOAA World Ocean Atlas salinity and temperature database; Locarnini et al., 2019; Zweng et al., 2019) as an objective and quantitative basis for selecting the best-performing parameter configuration to be our *control* (Fig. 1).



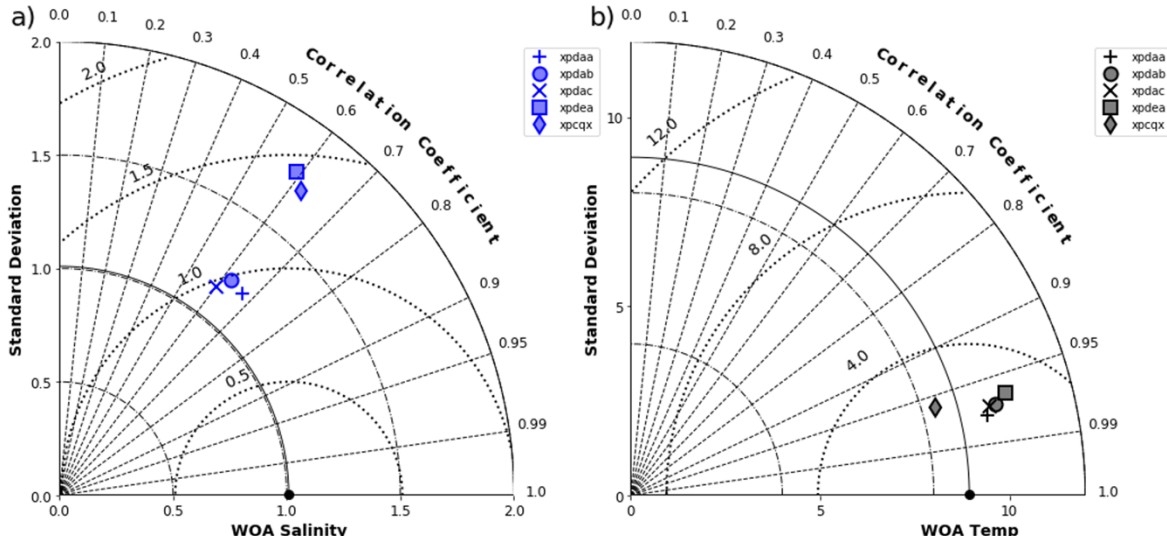

**Figure 1: Taylor diagrams summarising the performance of the four control-candidate simulations (XPDAA, XPDAB, XPDAC, XPDEA) in terms of their correlation, centred root mean square error, and ratio of their variances to the NOAA World Ocean Atlas**
**(a) salinity and (b) temperature databases (Locarnini et al., 2019; Zweng et al., 2019). The simulations were selected from a large FAMOUS pre-industrial perturbed parameter ensemble following the circulation performance screening described in Sect. 2.2. XPCQX represents the initial experiment, upon which the four control-candidates aim to improve.**

From this analysis, XPDAA returned the lowest root mean square error (and hence best performance using this metric; Fig. 1)

for simulating both salinity and temperature and so was used as our *control* simulation (Fig. 2).

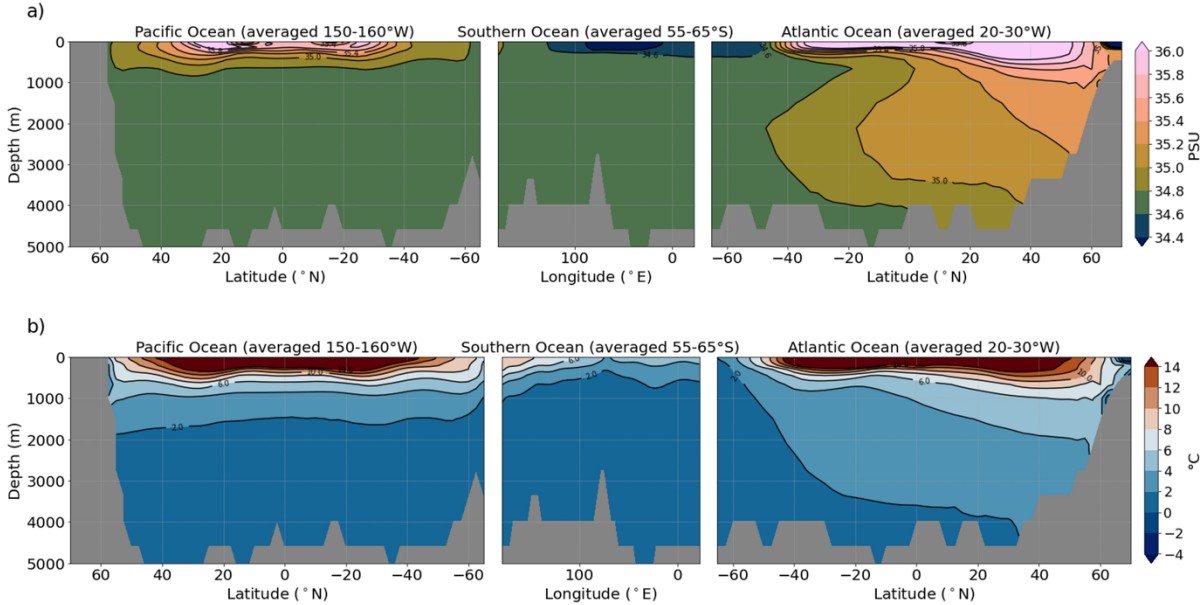

**Figure 2: (a) Salinity and (b) temperature profiles for the *control* simulation (centennial mean from final 100 years of a 5,000-year simulation) along a transect crossing the Pacific-Southern-Atlantic Ocean.**



The simulated steady state AMOC strength in FAMOUS under fixed pre-industrial boundary conditions is approximately 17 Sv (Fig. 3a), which we consider to be in excellent agreement with direct modern AMOC observations from the RAPID AMOC array at 26.5° N of 17.2 Sv from April 2004-October 2012, and the depth of maximum meridional stream function at

26.5° N is around 800 m (Fig. 3b), slightly shallower than RAPID observations of 900-1100 m (McCarthy et al., 2015; Sinha et al., 2018). In terms of the Atlantic circulation structure, the overturning cell of NADW descends to depths of 3,000 m as it bridges into the South Atlantic, and AABW fills the bottom of the Atlantic basin with southern-sourced waters up to 20° N. For the Nd isotope implementation, all sensitivity studies and model tuning described subsequently are based upon this simulation, XPDAA.

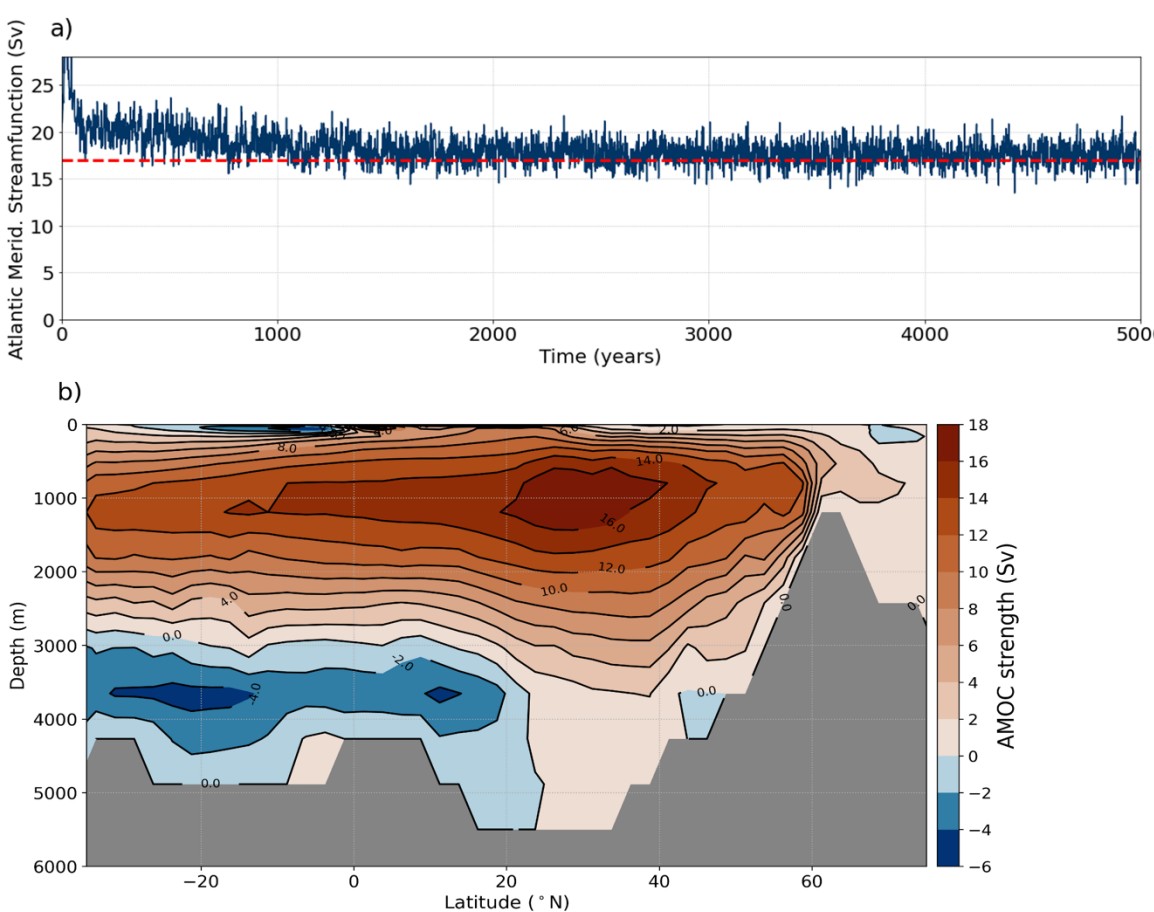

**Figure 3: Atlantic Meridional streamfunction for a 5,000-year simulation using *control* (XPDAA), showing (a) the maximum in time series (red dashed line indicates RAPID-AMOC 2004-2012 averaged AMOC strength at 26.5° N of 17.2 Sv; McCarthy et al., 2015), and (b) the zonal integration calculated from the last one hundred years.**





## 2.3 Neodymium isotope implementation in FAMOUS

In our simulated Nd isotope scheme (ND v1.0), we represent three different global sources of Nd into seawater: aeolian dust
flux, dissolved riverine input and dissolution of seafloor sediment. Neodymium ($Nd$) here refers to the sum of $^{143}Nd$ and $^{144}Nd$,
and the isotopic ratio ($IR$) relates to the ratio of $^{143}Nd$ to $^{144}Nd$, as shown:

$$Nd = {}^{143}Nd + {}^{144}Nd \tag{2}$$

$$IR = \frac{{}^{143}Nd}{{}^{144}Nd} \tag{3}$$

By rearranging Eq. (2) and using the isotopic ratio ($IR$: Eq. (3)), individual fluxes of each isotope can be calculated (Eq. (4)
and Eq. (5)) using information about Nd fluxes from each specific source and their associated $IR$ distributions:

$$^{143}Nd = \frac{Nd}{(1+(1/IR))} \tag{4}$$

$$^{144}Nd = \frac{Nd}{(IR+1)} \tag{5}$$

Neodymium isotopes are thus simulated and transported individually and independently as two separate tracers in our scheme,
explicitly resolving the concentration and distribution of each Nd isotope, allowing for [Nd] and $\varepsilon_{Nd}$ to be calculated offline
from the model output. It should be noted that of the Nd isotopes, $^{143}Nd$ and $^{144}Nd$ together only account for 36% of total Nd.
As such, obtaining absolute fluxes of $^{143}Nd$ and $^{144}Nd$ from total Nd fluxes requires a scaling of 0.36, and correspondingly, to
compare the total simulated Nd from Eq. (2) with observed total Nd requires scaling by 1/0.36. Due to this counteraction,
unscaled fluxes can be used, i.e. total Nd fluxes are used in the model and as such, the sum of the simulated $^{143}Nd + {}^{144}Nd$ (Eq.
(2)) can be easily converted to observed [Nd] for direct comparison, as carried out previously by Gu et al. (2019); Pöppelmeier
et al. (2020b); Rempfer et al. (2011).

The implementation of each source/sink term is described in detail in the following sections. To summarise these different
components, Nd sources from aeolian dust fluxes and dissolved riverine input enter the ocean only via the uppermost near-
surface ocean layer of the model. Seafloor sedimentary fluxes, an umbrella term that refers to a multitude of processes
encompassing boundary exchange (Lacan and Jeandel, 2005), submarine groundwater discharge (Johannesson and Burdige,
2007), and a benthic flux released from pore waters (Abbott et al., 2015a), are simulated via a combination of a sedimentary
source applied across sediment-water interfaces together with a separate sink occurring via particle scavenging. Removal of
Nd from the ocean model occurs when Nd scavenged/adsorbed onto sinking biogenic and lithogenic (dust) particles reaches
the seafloor via vertical fluxes and undergoes sedimentation, removing the particle-associated Nd from the ocean.





In the model numerics, fluxes of each Nd isotope into the ocean (kg m$^{-3}$ s$^{-1}$) are multiplied by a factor of $10^{18}$. This technique minimises the mathematical error associated with carrying small numbers, and so concentrations of each Nd isotope in model units are in ($10^{18}$ kg (Nd) m$^3$). A full list of all variables described in the text and their abbreviations are given in Table 1.

**Table 1: Nd scheme model parameters, abbreviations, fixed model parameter values, and units.**

| Variable | Abbreviation | Fixed parameter value | Unit |
|---|---|---|---|
| Total Nd concentration | $[Nd]_t$ | - | pmol kg$^{-1}$ |
| Dissolved Nd | $[Nd]_d$ | - | pmol kg$^{-1}$ |
| Particle-associated Nd | $[Nd]_p$ | - | pmol kg$^{-1}$ |
| Nd source, total | $f_{total}$ | - | g Nd yr$^{-1}$ |
| Nd source, density | $S_{total}$ | - | g Nd m$^{-3}$ yr$^{-1}$ |
| Dust source, total | $f_{dust}$ | $3.3 \times 10^8$ | g Nd yr$^{-1}$ |
| Dust source, density | $S_{dust}$ | - | g Nd m$^{-3}$ yr$^{-1}$ |
| Flux of dust | $F_{dust}$ | 2D horizontal global field from Hopcroft *et al.,* 2015 | g m$^{-2}$ yr$^{-1}$ |
| Nd concentration dust | $\overline{C}_{dust}$ | 20 | µg g$^{-1}$ |
| Nd dust dissolution | $\beta_{dust}$ | 0.02 | |
| Riverine source, total | $f_{river}$ | $4.4 \times 10^8$ | g Nd yr$^{-1}$ |
| Riverine source, density | $S_{river}$ | - | g Nd m$^{-3}$ yr$^{-1}$ |
| River discharge | $RIVER$ | - | m$^3$ yr$^{-1}$ |
| Riverine scaling factor | $\alpha_{river}$ | 1 | |
| Nd concentration river | $C_{river}$ | - | µg g$^{-1}$ |
| Nd removal, estuaries | $\overline{\gamma}_{river}$ | 0.7 | |
| Sediment source, total | $f_{sed}$ | - | g Nd yr$^{-1}$ |
| Sediment source, density | $S_{sed}$ | - | g Nd m$^{-3}$ yr$^{-1}$ |
| Total sediment surface | $A_{total}$ | - | m$^2$ |
| Gridbox sediment surface | $A(i, k)$ | - | m$^2$ |
| Gridbox volume | $V(i, k)$ | - | m$^3$ |
| Thickness of euphotic layer | $z_{eu}$ | 81 | m |
| Penetration depth of opal | $l_{opal}$ | 10,000 | m |
| Penetration depth of CaCO3 | $l_{calcite}$ | 3,500 | m |
| Particle settling velocity | $\omega$ | 1,000 | m yr$^{-1}$ |



| | | | |
|---|---|---|---|
| Ratio [Nd]$_p$ to [Nd]$_d$ | $[Nd]_p/[Nd]_d$ | - | |
| Global average density of seawater | $p$ | 1,024.5 | kg m$^{-3}$ |
| Reversible scavenging, density | $S_{rs}$ | - | g Nd m$^{-3}$ yr$^{-1}$ |
| Ratio between average POC concentration and density of seawater | $R_{POC}$ | $2.93 \times 10^{-9}$ | |
| Ratio between average CaCO3 concentration and density of seawater | $R_{CaCO3}$ | $6.27 \times 10^{-9}$ | |
| Ratio between average opal concentration and density of seawater | $R_{opal}$ | $5.21 \times 10^{-9}$ | |
| Ratio between average dust concentration and density of seawater | $R_{dust}$ | $1.73 \times 10^{-9}$ | |
| Total Nd inventory after equilibrium | $Nd(I)$ | - | $10^{12}$ g |

### 2.3.1 Dust source

Surface dust deposition ($F_{dust}$) is prescribed in the model from the annual mean dust deposition for the pre-industrial as simulated by the atmosphere component of the Hadley Centre Global Environmental Model version 2 (HadGEM2-A) GCM (Collins et al., 2011) (Fig. 4a). The dust deposition scheme (described by Woodward, 2011) has been shown to be in generally

good agreement with observations, with concentrations in the Atlantic well simulated across the whole of the Saharan dust plume, although some discrepancies occur, including an overestimation at some Pacific sites during spring (Collins et al., 2011). The simulation of pre-industrial climate conditions within HadGEM2 are described by Hopcroft and Valdes (2015), and the dust results specifically are described in full by Hopcroft et al. (2015). Based on these simulated dust fluxes, we apply an Nd source per volume ($S_{dust}$: kg m$^{-3}$ yr$^{-1}$) in the uppermost layer of the ocean model, assuming a global mean concentration

of Nd in dust $\overline{C}_{dust}$ ($\overline{C}_{dust}$ = 20 µg g$^{-1}$) (Goldstein et al., 1984; Grousset et al., 1988, 1998), from which only a certain fraction $\beta_{dust}$ ($\beta_{dust}$ = 0.02: Greaves et al., 1994) dissolves in seawater.

$$S_{dust}(i,k) = \frac{F_{dust}(i,1) \times \overline{C}_{dust} \times \beta_{dust}}{dz(i,1)} \qquad (6)$$

Here $i,k$ represents the horizontal and vertical indexing of model grid cell and $dz$ is the grid cell's thickness (10 m in the uppermost surface layer where $k$ =1). The total global flux of Nd from surface aeolian dust deposition to seawater ($f_{dust}$) is $3.3 \times$

$10^8$ g(Nd) yr$^{-1}$. Although we use an updated dust deposition field compared to previous studies, prescribed $\overline{C}_{dust}$ and $\beta_{dust}$ are




broadly consistent with earlier Nd isotope schemes, providing a comparable total Nd dust source ($1.0 \times 10^8$ g-$5.0 \times 10^8$ g (Nd)

yr$^{-1}$; Arsouze et al., 2009; Gu et al., 2019; Pöppelmeier et al., 2020b; Rempfer et al., 2011; Tachikawa et al., 2003).

**Figure 4: (a) Annual dust deposition taken from the pre-industrial annual mean dust deposition simulated by HadGEM2-A GCM**
**(Hopcroft et al., 2015), and (b) $\varepsilon_{Nd}$ signal from dust deposition following Tachikawa et al. (2003) and updated with information from**
**Mahowald et al. (2006) and Blanchet (2019).**

For the Nd isotope compositions of the dust flux, we started with the first-order estimate by Tachikawa et al. (2003), as follows:

North Atlantic > 50° N: $\varepsilon_{Nd}$ = -15, Atlantic < 50° N: $\varepsilon_{Nd}$ = -12, North Pacific > 44° N: $\varepsilon_{Nd}$ = -5, Indopacific < 44° N: $\varepsilon_{Nd}$ = -7,





and remainder: $\varepsilon_{Nd}$ = -8. This was revised with additional constraints, accounting for the dust plume expansions as reported by

Mahowald et al. (2006), in combination with the mean Nd isotope signatures of the respective source regions as determined

by the global compilation of detrital Nd isotope data by Blanchet (2019) (Fig. 4b).

### 2.3.2 Dissolved riverine source

To represent the Nd source from dissolved river fluxes, we used the river outflow to the ocean simulated by FAMOUS (*RIVER*)

as our river water discharge (kg m$^{-2}$ s$^{-1}$) and combined this outflow with both the Nd concentration *($C_{\text{river}}$; µg g$^{-1}$)* and isotopic

concentration (used to calculate the flux from each Nd isotope using Eq. (4) and Eq. (5)) of river water dissolved material, as

estimated by Goldstein and Jacobsen (1987; see Table 3). All river source Nd fields are shown in Fig. 5.





**Figure 5: (a) Simulated river outflow (*RIVER*) in FAMOUS, (b) major river ε$_{Nd}$, and (c) major river [Nd]; (b) and (c) are prescribed following estimates by Goldstein and Jacobsen (1987).**



River outflow in FAMOUS is based on a routing scheme that instantaneously delivers terrestrial runoff (precipitation - [evaporation + soil moisture]) from the location that precipitation falls to the designated coastal grid cell that the runoff would reach due to river routing (i.e., the river mouth). Relating the riverine Nd flux to the model's prognostic river discharge allows the Nd river source to respond to different climatic conditions, making it a more dynamic and predictive tool for examining the impact of changes in the global hydrological balance, such as wetting/drying events, or shifts in the

monsoons. Essentially, river outflow plumes are 'tagged' with the estimated ε$_{Nd}$ (which is provided as an input map along the coasts; Fig. 5b) according to the model's projected water discharge at that location and $C_{river}$ (also provided as an input map along the coasts; Fig. 5c). For palaeo or future climate applications where river routing is significantly different to today, the input maps controlling ε$_{Nd}$ and $C_{river}$ tagging at the coast would need to be updated to reflect Nd dissolution from the modified fluvial pathway over land, the model should predict the rest.

Estuaries are important biogeochemical reactors of rare earth elements (REEs), with sea salt driving flocculation of river-dissolved organic matter, which results in estuarine REE removal (Elderfield et al., 1990). This removal is known to be important in balancing the marine REE budgets, however the complex processes involved are still not fully constrained (Elderfield et al., 1990). Rousseau et al. (2015) summarised published observations of Nd removal (%) in estuaries from observations of estuarine dissolved Nd dynamics (Rousseau et al., 2015; their Table 1). Published values (n = 17) range from

40% in Tamar Neaps (Elderfield et al., 1990) to 97% in the Amazon (Sholkovitz, 1993), with a mean Nd removal of 71 ± 16% (s.d.). Based upon this, and parallel to previous model schemes (e.g., Rempfer et al., 2011; Gu et al., 2019; Arsouze et al., 2009), we assume that 70% of dissolved Nd from river systems are removed in estuaries (i.e., $\overline{\gamma}_{river}$ = 0.7).

The dissolved riverine source per unit volume of Nd ($S_{river}$: kg m$^{-3}$ yr$^{-1}$) in the uppermost layer of the ocean is therefore calculated as:

$$S_{river}(i,k) = \frac{RIVER(i,1) \times C_{river} \times (1.0 - \overline{\gamma}_{river})}{dz(i,1)} \qquad (7)$$

The total global flux of river sourced dissolved Nd to seawater ($f_{river}$) is $4.4 \times 10^8$ g(Nd) yr$^{-1}$. Previous Nd isotope schemes have applied either fixed annual mean continental river discharge estimates from Goldstein and Jacobsen (1987) and Dai and Trenberth (2002), as applied in the Bern3D Nd isotope scheme (Rempfer et al., 2011; Pöppelmeier et al., 2021a), or, similar to this study, using the model's own river routing and discharge schemes, as with NEMO-OPA (Arsouze et al., 2009) and

CESM1 (Gu et al., 2019). Our estimated global total riverine Nd source to the oceans sits within previous model estimates ($2.6 \times 10^8$–$1.7 \times 10^9$ g(Nd) yr$^{-1}$; Arsouze et al., 2009; Gu et al., 2019; Pöppelmeier et al., 2020b; Rempfer et al., 2011; Tachikawa et al., 2003).





There is a larger range in estimated riverine Nd flux to the ocean relative to the estimated dust flux ranges. It should be noted that the largest simulated Nd river source amongst these studies ($1.7 \times 10^9$ g(Nd) yr$^{-1}$; Pöppelmeier et al., 2020b) in the updated

Bern3D Nd isotope scheme applied a river scaling factor, used as a tuning parameter and based on findings by Rousseau et al. (2015), who suggest a globally significant release of Nd to seawater by dissolution of river sourced lithogenic suspended sediments grounded upon observations in the Amazon estuary. Our model does not attempt to fully resolve all complex estuarine processes, and in this study we chose to represent the dissolved riverine flux as a single source to seawater. Early findings by Goldstein and Jacobsen (1987) document that the Nd isotope composition of dissolved and suspended river loads

can vary by up to four epsilon units. Observations presented by Rousseau et al. (2015) showed the measured Amazon dissolved river end member value ($\varepsilon_{Nd}$ = -8.9) was more radiogenic than the typical suspended river material ($\varepsilon_{Nd}$ = -10.6) and as such, combining dissolved and particulate sources in river $\varepsilon_{Nd}$ budgets is non-trivial. Furthermore, our sediment Nd source to the ocean (described Sect. 2.3.3), which occurs across sediment-water interfaces, utilises the continental margin and seafloor $\varepsilon_{Nd}$ distribution maps by Robinson et al. (2021), thus using the most recent compilation of published global observations of Nd

isotope compositions of river sediment samples deposited on the continental shelf and slopes (alongside geological outcrops and marine sediment samples). It is therefore likely that this margin source encompasses at least in part the Nd isotope fingerprint from a river particle flux. Notwithstanding, our model does permit a scaling of the Nd river flux ($\alpha_{river}$), which, although out of the scope of this study, could be applied in future model development to explore in more detail a particulate river source as a major Nd flux to seawater.

### 340   2.3.3 Continental margin and seafloor sediment source

The sediment source describes the flux of Nd into seawater entering each model grid cell adjacent to sediment, this source is not restricted to the uppermost surface layers and can be implemented across all vertical and horizonal sediment-water interfaces, as is the case for all experiments described in this study. As such, the sediment source represents (1) boundary exchange, as defined by Lacan and Jeandel (2005b), which describes the flux of Nd from particle-seawater exchange occurring

predominantly across the continental margins; (2) submarine groundwater discharge, as suggested by Johannesson and Burdige (2007), which releases Nd to seawater via discharge of fresh groundwater to coastal seas and is mainly limited to the upper 200 m; and (3) a benthic flux, which specifically refers to a transfer of Nd from sediment pore water to seawater resulting from early diagenetic reactions and is not depth limited (Abbott et al., 2015b, a; Haley et al., 2017).

The total sediment Nd source per unit volume ($S_{sed}$: kg m$^{-3}$ yr$^{-1}$) into any given ocean grid cell is dependent on the fraction of

the surface area of that cell that is in contact with sediment. Similar to previous schemes (e.g., Gu et al., 2019; Pöppelmeier et al., 2020b; Rempfer et al., 2011), the total globally integrated sediment-associated Nd source to the ocean ($f_{sed}$: g Nd yr$^{-1}$) is used as a tuning parameter.





$$S_{sed}(i,k) = f_{sed} \times \frac{A(i,k)}{A_{total}} \times \frac{1}{V(i,k)} \,, \tag{8}$$

where $A(i,k)$ is the total area of the sediment surface in contact with seawater per ocean grid cell (m²), $A_{total}$ is the total global
area of the sediment surface where a sediment source occurs (m²) and $V(i,k)$ is the volume of water per ocean grid cell (m³).

There remains no true consensus on whether (and how) to apply spatial variability to sediment Nd sources, as reflected in the
way previous schemes have adopted different approaches. Arsouze et al. (2009) limited the sediment flux to the upper 3,000
m of the ocean (to represent 'boundary exchange' processes), imposing a depth scaling factor, and considering estimated [Nd]
distributions across the continental margins using the earlier marine margin Nd compilation by Jeandel et al. (2007). Both
Rempfer et al. (2011) and Gu et al. (2019) simplified this method by applying spatially uniform sediment Nd fluxes, also
limited to the upper 3,000 m. In more recent work, Pöppelmeier et al. (2020b) removed the depth limitation (as we have done),
and incorporated a geographically-varying scaling factor that extrapolated modern Nd flux observations to try to capture more
localised features through increased benthic fluxes (Abbott et al., 2015b; Blaser et al., 2020; Grenier et al., 2013; Lacan and
Jeandel, 2005; Rahlf et al., 2020). Pasquier et al. (2021) presented the first inverse model of global marine cycling of Nd, and
in this latest scheme, the strength of the sediment Nd flux to the ocean was imposed with an exponential depth function,
resembling eddy kinetic energy and particulate organic matter fluxes, which are characteristically larger near surface and
coastal regions.

For our scheme, we do not assume spatial variations in the sediment source of Nd ($f_{sed}$). The flux per unit area is uniform with
depth, latitude and longitude, essentially avoiding making explicit inferences on the nature of the sedimentary Nd source. It
has been proposed that preferential mobilisation of certain components of the sediment drive spatial variations in sediment
fluxes (e.g., Abbott et al., 2015a; Wilson et al., 2013; Du et al., 2016; Abbott, 2019; Abbott et al., 2019), and that both detrital
and authigenic phases likely exchange Nd within pore water during early diagenesis (Du et al., 2016; Blaser et al., 2019b).
However, the elusiveness of marine Nd cycling alongside our limited knowledge of the specific mechanisms controlling
sediment-water Nd exchange means that determining generalisable rules for where and under what conditions (e.g., redox
environments or fresh labile detrital material) preferential mobilisation may occur is unknown and challenging to resolve.
Therefore, in accordance with Pöppelmeier et al. (2020a), Du et al. (2020), Gu et al. (2019) and Rempfer et al. (2011), we
adopt a constant detrital sediment flux as a first-order approximation. In fact, we contend that applying this simpler method,
as opposed to constructing a more complex source term that is arguably just as arbitrary (given the uncertainty in Nd cycling),
allows for a more explicit quantification of differences between observed and simulated Nd distributions. As such, without
overfitting our model, we allow for the clearest indication of those parts of the system that are well understood (and
represented), and those which prove deficient. Under this framework, we may separate out and test the effect of many of the
major Nd sources/sinks with dedicated sensitivity simulations, including the possibility, in future work, of incrementally



modifying the sediment source distributions to increase the complexity of the scheme and assess the impact of our various assumptions.

The isotopic ratio of the sediment Nd flux to seawater is prescribed using the recent updated global gridded map of bulk detrital $\varepsilon_{Nd}$ at the continental margins and seafloor (Robinson et al., 2021, Fig. 6a). Using Eq. (4) and Eq. (5), fluxes of each Nd isotope are calculated from this condition and bi-linearly regridded to the model's native resolution (Fig. 6b). Previous studies used the predecessor continental $\varepsilon_{Nd}$ map by Jeandel et al. (2007) (e.g. Arsouze et al., 2009, 2007; Gu et al., 2019; Rempfer et al., 2011); see Fig. 5 in Robinson et al. (2021) for a comparison of the differences). Marginal $\varepsilon_{Nd}$ distributions in our improved
map are broadly more radiogenic across the Arctic Shield, Northern Eurasia, South America, north-eastern Africa and Antarctica and more unradiogenic over southern Greenland, north-eastern Europe, western and eastern Africa and parts of the Americas. Crucially, the newer map of Robinson et al. (2021) provides the first estimate of global seafloor sediment Nd isotope composition, which is necessary for considering a global (e.g., rather than depth limited) benthic flux.



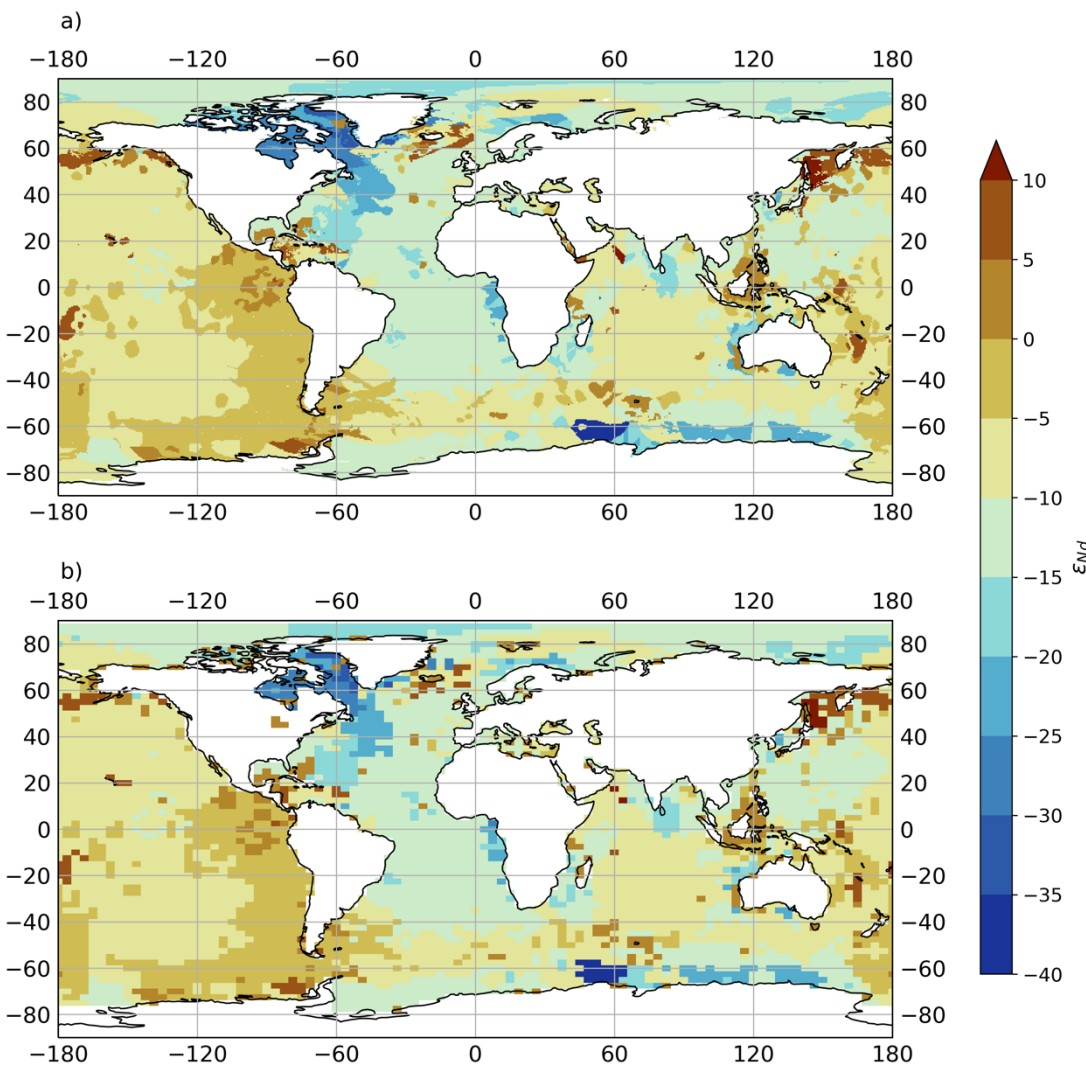

**Figure 6: (a) Map of the global $\varepsilon_{Nd}$ distributions at the sediment-ocean interface from Robinson et al. (2021), and (b) as used as a model input in this study (bi-linearly regridded from (a) onto the coarser FAMOUS ocean grid).**

Pasquier et al. (2021) first applied the sedimentary $\varepsilon_{Nd}$ map from Robinson et al. (2021) in their recent global marine Nd isotope scheme, imposing positive (i.e., radiogenic) modifications to the Pacific sedimentary $\varepsilon_{Nd}$ and using a reactivity scaling factor (linked to sediment lability) that favours more extreme $\varepsilon_{Nd}$ signals. Here, we again adopt a simpler approach, imposing the unmodified sediment $\varepsilon_{Nd}$ distributions from Robinson at al. (2021), allowing for a presentation of our new scheme based on what we confidently know about Nd cycling without the complication of over-conditioning our model inputs. Future research may then use this work as a foundation to robustly explore different choices for model inputs (i.e., boundary conditions) to revisit these fundamental questions about Nd sediment source.



### 2.3.4 Internal cycling via reversible scavenging

Vertical cycling and removal of Nd from the water column via sinking particles ('reversible scavenging') is parameterised using the same approach as previous Nd isotope implementations (Siddall et al., 2008; Arsouze et al., 2009; Rempfer et al., 2011; Gu et al., 2019; Pöppelmeier et al., 2020a; Oka et al., 2021). Based on the original scheme of Bacon and Anderson (1982), the method captures the physical process of absorption/incorporation and desorption/dissolution of Nd onto particle surfaces in seawater. The scheme assumes that particle associated Nd is in dynamic equilibrium with falling particles

throughout the water column, with continuous exchange between the particle and dissolved pools. This process redistributes Nd within the water column, acting as a net sink of dissolved Nd at shallower depths as they adsorb onto particle surfaces, and a net source at greater depths where dissolution of particles releases dissolved Nd back to seawater. The scavenging is the only sink of Nd in our model; Nd associated to particles (particulate organic carbon, POC; calcium carbonate, $CaCO_3$; opal; dust) reaching the seafloor is removed from the system, operating under a steady state assumption that acts to balance the three

external input sources (marine sediments, dust and rivers). Previous studies (e.g., Siddall et al., 2008; Arsouze et al., 2009; Rempfer et al., 2011; Gu et al., 2019; Wang et al., 2021) have demonstrated that reversible scavenging is an active and important component in global marine Nd cycling, and is necessary for successfully simulating both $[Nd]_d$ and $\varepsilon_{Nd}$ distributions.

Updating the approach employed by Siddall et al. (2008), we prescribe individual biogenic particle export fields based on satellite-derived primary production. FAMOUS does contain an optional ocean biogeochemistry module (Hadley Centre

Ocean Carbon Cycle; HadOCC), which includes simplified nutrient-phytoplankton-zooplankton-detritus (NPZD) classes (Palmer and Totterdell, 2001) and could instead be used as the basis for predicting vertical particle fluxes in the ocean (which was the approach adopted by Arsouze et al., 2009; Gu et al., 2019; and Rempfer et al., 2011). We favoured satellite-derived estimates in order to improve the accuracy of particle-associated cycling of Nd and reduce biases inherent to the intermediate complexity biogeochemistry model (Dentith et al., 2020; Palmer and Totterdell, 2001). This approach also optimises the

computational efficiency of our scheme, since the added ecosystem and geochemistry tracer fields in HadOCC slows the model down. However, future work could modify our implementation to instead associate Nd cycling with the prognostic particle fields in HadOCC.

In our scheme, biogenic particle fields (POC, $CaCO_3$, and opal) are prescribed using gridded, global satellite-derived particle export productivity from Dunne et al. (2012, 2007). The euphotic zone ($z_{eu}$), is set to a globally uniform depth in FAMOUS of

81 m, the closest bottom grid box depth to match that defined in the OCMIP II protocol (75 m) (Najjar and Orr, 1998). Below $z_{eu}$, appropriate depth-dependent dissolution profiles, derived from assumptions of particle degradability and sinking speed (Martin et al., 1987; Laws et al., 2000; Behrenfeld and Falkowski, 1997), and widely used to model flux attenuation in ocean models (e.g. all models used in the Ocean-Carbon Cycle Model Intercomparison Project; Doney et al., 2004; Sarmiento and LeQuéré, 1996), were applied to the biogenic export fluxes.






Downward fluxes of POC ($F_{POC}$) follow the power-law profile of Martin et al. (1987):

$$F_{POC}(z) = F_{POC}(z_{eu}) \times \left(\frac{z}{z_{eu}}\right)^{-\propto} (for\ z > z_{eu}) , \qquad (9)$$

where $z$ is depth (m), and $\propto$ represents a dimensionless dissolution constant for POC set to 0.9 (Najjar and Orr, 1998). Although a widely used parameterisation of dissolution, it should be noted this so-called 'Martin curve' is known to underestimate the

flux to the sediment in the off-equatorial tropics and subtropics and overestimates the flux in subpolar regions, indicating particles penetrate deeper than the Martin curve in the tropics and shallower in sub-polar regions (Dunne et al., 2007).

Downward fluxes of opal ($F_{opal}$) and CaCO3 ($F_{CaCO3}$) follow exponential dissolution profiles with particle-specific length scales $l_{opal}$ (10,000 m) and $l_{calcite}$ (3,500 m) (Maier-Reimer, 1993; Henderson et al., 1999; Najjar and Orr, 1998):

$$F_{opal}(z) = F_{opal}(z_{eu}) \times exp\left(\frac{-z-z_{eu}}{l_{opal}}\right), \qquad (10)$$

$$F_{CaCO_3}(z) = F_{CaCO_3}(z_{eu}) \times exp\left(\frac{-z-z_{eu}}{l_{CaCO_3}}\right), \qquad (11)$$

The sinking of dust is prescribed according to the pre-industrial annual mean dust deposition simulated by Hopcroft et al. (2015) (see Sect. 2.3.1). We assume that dust does not dissolve significantly with depth and so dust export fluxes are constant throughout the water column. In line with previous schemes (e.g. Arsouze et al., 2009; Pöppelmeier et al., 2020b; Rempfer et al., 2011; Siddall et al., 2008), a uniform settling velocity ($\omega$) of 1,000 m yr[-1] is applied to all particle fields, capturing the

mean particle flux to the seafloor.

Annual averaged export of biogenic fields are shown in Fig. 7 and the total annual export production of POC (9.6 Gt(C) yr[-1]), CaCO3 (0.45 Gt(C) yr[-1]), and opal (90 Tmol(Si) yr[-1]) are comparable with previous estimates, although export of CaCO3 and opal are at the lower end, as highlighted in Table 3 by Dunne et al. (2007). Note, the annual export of CaCO3 reflects the new

optimised surface calcite parameterisation as described in Dunne et al. (2012).



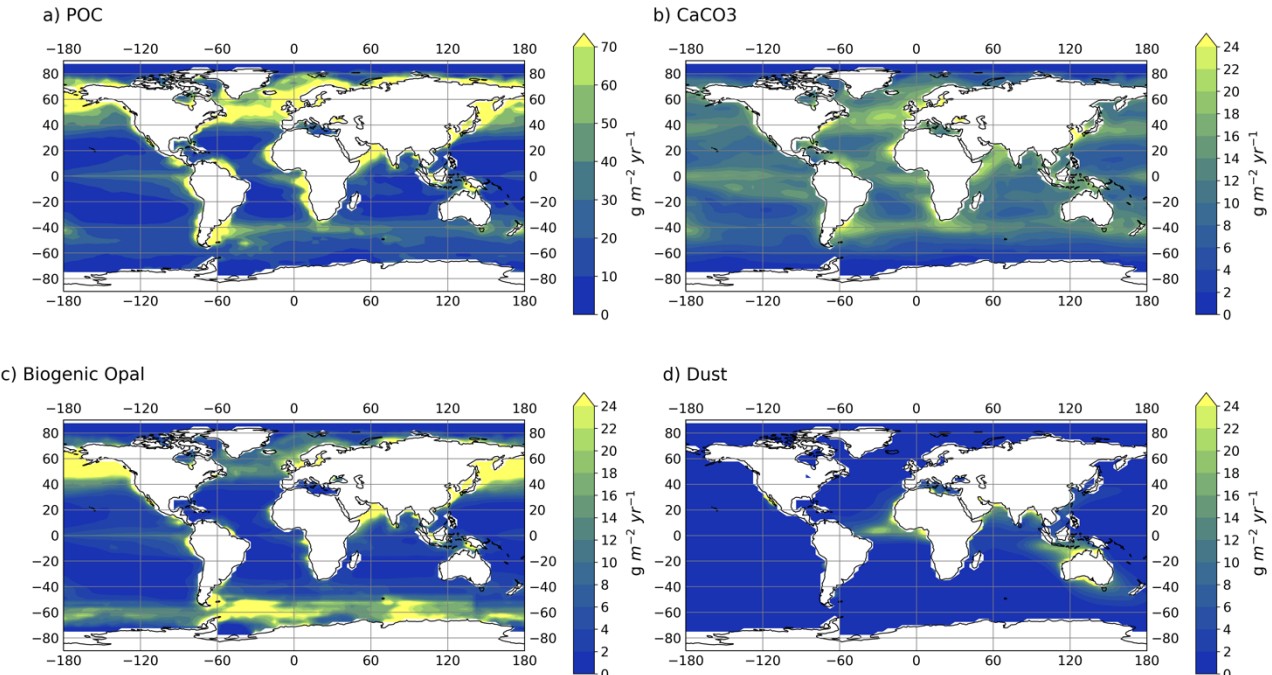

**Figure 7: Particle export fields from the ocean surface (g m$^{-2}$ yr$^{-1}$). Biogenic particle fields are prescribed using satellite-derived export productivity fields for (a) POC, (b) CaCO$_3$ and (c) opal (Dunne et al., 2007, 2012). The dust input fields (d) are annual mean dust deposition simulated for the pre-industrial by the HadGEM2-A GCM (Hopcroft et al., 2015). Note the different scale used for panel (a).**

Reversible scavenging ($Nd_{rs}$) considers total Nd for each isotope ($j$) as the dissolved Nd ($Nd_d$) and particulate Nd ($Nd_p$) associated with the different particle fields ($\chi$; where $\chi$ = POC, CaCO$_3$, opal and dust).

$$[Nd]_t^j = [Nd]_d^j + [Nd]_p^j = [Nd]_d^j + \sum_i [Nd]_{p,\chi}^j \tag{12}$$

Nd$_p$ sinks in the water-column with the particles due to gravitational force. Dissolution of biogenic particles with increasing depth below the euphotic zone releases Nd incorporated/adsorbed onto particles back to seawater (i.e., the Nd is dissolved/desorbed). Thus, particles act as internal sinks for marine Nd$_d$ in shallower depths and as sources at greater depths. This combined process for reversible scavenging ($S_{rs}$) in the model can be described by:

$$S_{rs}(i,k) = \frac{-\partial\left(\omega[Nd]_p(i,k)\right)}{\partial z(i,k)} \tag{13}$$

Where $[Nd]_p$ can be determined within the model from total Nd for each isotope:





$$[Nd]_p^j(i,k) = [Nd]_t^j(i,k) \times \left(1 - \frac{1}{1+\sum_i R_\chi(i,k) \times K_\chi^j}\right),$$  (14)

$R_\chi(i,k)$ describes the dimensionless ratio between particle concentration for each particle type ($C_\chi(i,k)$) and the average density of seawater ($p$:1024.5 kg m$^{-3}$), input as fixed boundary conditions in our scheme. $C_\chi$ is calculated from the prescribed particle fluxes ($F_\chi$: Fig. 7) by assuming a globally uniform settling velocity ($\omega$ =1,000 m yr$^{-1}$; Arsouze et al., 2009; Dutay et al., 2009; Gu et al., 2019; Kriest and Oschlies, 2008; Rempfer et al., 2011), i.e., $C_\chi = F_\chi / \omega$. The equilibrium between dissolved

concentration ($[Nd]_d$) and concentration associated to particle-type ($[Nd]_p$) can be described by the equilibrium partition coefficient ($K_\chi^j$):

$$K_\chi^j = \frac{[Nd]_{p,\chi}^j}{[Nd]_d^j} \times \frac{1}{R_\chi},$$  (15)

here $[Nd]_p/[Nd]_d$ represents the scavenging efficiency in the model. It is independent of particle type and is used as a tuning

parameter. $\overline{R}_\chi$, however, is dependent on particle type (where $\overline{R}_\chi = \overline{C}_\chi/p$) and thus $K_\chi$ is different for different particles. Our global mean particle concentrations ($\overline{C}_\chi$; Table 2) are similar to those reported in previous schemes; see Supplementary Information: Table S3 for a comparison.

**Table 2: Global mean particle concentrations for each particle type used to calculate equilibrium scavenging coefficients following Eq. (15). In summary export fluxes of POC, CaCO3 and opal are from Dunne et al. (2007; 2012) and dust fluxes are from Hopcroft**
**et al. (2015).**

| Particle type | Acronym | Concentration (kg m$^{-3}$) |
|---|---|---|
| POC | $\overline{C}_{POC}$ | $3.0 \times 10^{-6}$ |
| CaCO$_3$ | $\overline{C}_{CaCO3}$ | $6.43 \times 10^{-6}$ |
| Opal | $\overline{C}_{opal}$ | $5.33 \times 10^{-6}$ |
| Dust | $\overline{C}_{dust}$ | $1.78 \times 10^{-6}$ |

Isotopic fractionation during absorption/incorporation and desorption/dissolution is neglected due to similar masses of [143]Nd and [144]Nd, avoiding undue complexity arising from any assumption about preferential scavenging (Siddall et al., 2008). Adsorption occurs everywhere that particles are present and we do not allow for preferential scavenging onto different particle

types, consistent with previous models (e.g. Rempfer et al., 2011). In contrast, Siddall et al. (2008) optimised $K_\chi$ to fit observed $[Nd]_d$ for each particle type. Their optimised solution effectively implied no scavenging of Nd by POC, this result was considered tentative due to similar dissolution profiles of CaCO$_3$ and POC, which likely biased scavenging to CaCO$_3$ that may have been more correctly attributable to POC.



Therefore, and by including advection and diffusion processes (*Transport*), it follows that the total conservation equation for
each Nd isotope in the model scheme can be written as:

$$\frac{\delta [Nd]_t^j}{\delta t} = S_{dust}^j + S_{river}^j + S_{sed}^j + S_{rs}^j + Transport, \tag{16}$$

## 2.4 Evaluation methods & data sets

To validate the new Nd isotope scheme (ND v1.0) and to assess the model's performance, we compare the simulated $[Nd]_d$
and $\varepsilon_{Nd}$ to modern seawater measurements, with a focus on describing the ability of the model to represent key spatial and
vertical distributions across ocean basins.

As part of this assessment, a basic indication of model skill is returned by the mean absolute error (MAE):

$$MAE = \frac{1}{N}\sum_{k=1}^{N}|obs_k - sim_k|, \tag{17}$$

where $obs_k$ and $sim_k$ are measured and simulated $[Nd]_d$ or $\varepsilon_{Nd}$ respectively, and $k$ is an index over all observational data. For
each measurement – based on its longitude, latitude, and depth – the value predicted by the model is extracted and the mean
deviation of simulated and observed $[Nd]_d$ and $\varepsilon_{Nd}$ is presented in pmol kg$^{-1}$ and epsilon units respectively. Here we chose
specifically not to apply a grid box volume weighting to the MAE, which would act to emphasise abyssal Pacific results in our
assessment of model skill, where there are few observations and relatively low variability in Nd distributions. The advantage
of using an unweighted MAE is that the assessment metric better scrutinises regions with larger (spatial) gradients in both
$[Nd]_d$ or $\varepsilon_{Nd;}$ i.e., at the surface and high latitudes. The observational data used in this assessment are from the seawater REE
compilation used by Osborne et al. (2017, 2015), augmented with more recent measurements including data in the
GEOTRACES Intermediate Data Product 2021 (GEOTRACES Intermediate Data Product Group, 2021) from GEOTRACES
cruises (GA02, GA08, GP12, GN02, GN03 and GIPY05). Combined, our observational database represents a total of 6,048
$[Nd]_d$ and 3,278 $\varepsilon_{Nd}$ measurements, making it the largest compilation of seawater Nd data used to date to validate the
performance of an Nd isotope enabled model. Notably, we omit measurements of $[Nd]_d > 100$ pmol kg$^{-1}$ from the model data
comparison because they represent very localised signals which we do not attempt to resolve. These include extreme surface
concentrations present in restricted basins such as fjords, the Baltic Sea and the Gulf of Alaska (Chen et al., 2013; Haley et al.,
2014), or input from hydrothermal activity (Chavagnac et al., 2018), which is not believed to govern global marine Nd
distributions due to the immediate removal of hydrothermal Nd at the vent site (Goldstein and O'Nions, 1981). The location
and spatial distribution of all observational records used in this study are shown in Fig. 8, and full details of the seawater
compilation including a full list of all the references for the data sources are provided in Supplementary Information: Table
S4.





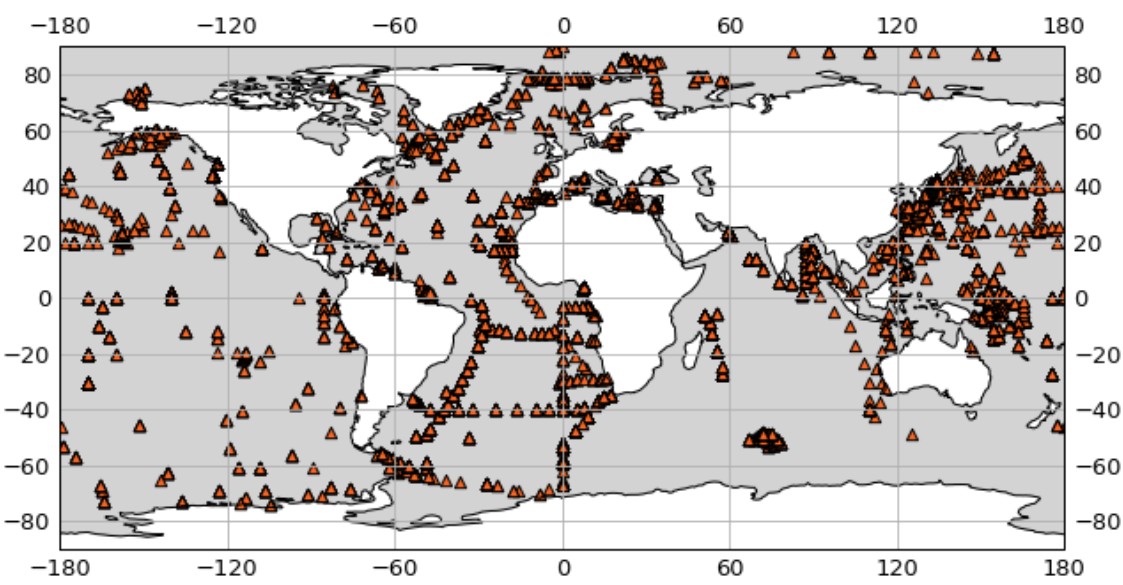

**Figure 8: Location of marine observational records used in this study, (a) filled orange triangles show the location of dissolved Nd concentration records, and (b) filled sky-blue circles show the location of dissolved $\varepsilon_{Nd}$ records.**

Neither the horizontal nor vertical distribution of global seawater $[Nd]_d$ and $\varepsilon_{Nd}$ observational data are even. Most $[Nd]_d$ measurements are in the Pacific Ocean followed by the Atlantic Ocean (representing 37% and 35% of measurements respectively, noting that the Pacific has more than twice the volume of the Atlantic), with far fewer measurements in the Indian




Ocean (12%) and the Southern Ocean (5%). In comparison, $\varepsilon_{Nd}$ measurements are most frequent in the Atlantic Ocean (45%), followed by Pacific Ocean (29%), and again there are fewest measurements from the Indian Ocean (8%) and Southern Ocean
(7%). Both $[Nd]_d$ and $\varepsilon_{Nd}$ observational data are biased towards shallower depths, with median depths of 496 m and 505 m, with only 10% and 11% of measurements at depths below 3,000 m respectively. It is therefore important to highlight that, as with previous studies, skew in the distribution of seawater Nd measurements will act to bias our assessment of model performance; $MAE_{[Nd]}$ towards the Pacific Ocean and Atlantic Ocean, $MAE_{\varepsilon Nd}$ towards the Atlantic Ocean, and both towards shallower depths.

In some instances, near land grid cells, the location of the compared measured and modelled Nd may not match due to the coarseness of the model grid. In such cases, we employ a nearest neighbour algorithm to extract the modelled value from the closest ocean model grid cell. Furthermore, if multiple measurements occur within one model grid cell, the arithmetic mean of the values is used for our comparison to model results, and as such, $n = 3,471$ and 2,136 for the calculation of both $MAE_{[Nd]}$ and $MAE_{\varepsilon Nd}$ respectively.

To ensure that our evaluation is not overly reliant on the cost-function analysis alone, and hence reduce the spatial biases in our assessment from the geographically uneven spread of measured $[Nd]_d$ and $\varepsilon_{Nd}$, we also evaluate the capability of the model to reproduce appropriate global Nd inventories and to simulate large-scale horizontal and vertical gradients. For example, we compare $[Nd]_d$ and $\varepsilon_{Nd}$ patterns across the global thermohaline circulation, through inter basin gradients, depth profiles and between distinct water masses, critically assessing the contributions of distinct Nd sources and cycling processes. We compare
our results briefly with findings from previous modelling studies, but we highlight that the purpose of this study is to understand the behaviour of our model, and not to undertake a comprehensive calibration of its performance. Optimisation (or 'tuning') of the Nd scheme will follow this work, and this needs to be considered when comparing the cost function performance to previous schemes.

**2.5 Sensitivity experiment design**

We designed a number of sensitivity experiments (Table 3) to systematically vary individual model parameters describing the reversible scavenging efficiency ($[Nd]_p/[Nd]_d$) and the main Nd source ($f_{sed}$) in order to better understand our model's behaviour and performance. These two parameters ($[Nd]_p/[Nd]_d$ and $f_{sed}$) were chosen primarily as they represent important and largely unconstrained non-conservative processes that are understood to govern simulated global distributions of both seawater $[Nd]_d$ and $\varepsilon_{Nd}$ (Rempfer et al., 2011; Pöppelmeier et al., 2020a; Gu et al., 2019; Arsouze et al., 2009; Siddall et al., 2008). By isolating
individual effects, the primary aim was to understand in detail the model's sensitivity to different forcings, identify which parameters are important for $[Nd]_d$ and/or $\varepsilon_{Nd}$ patterns across different ocean basins and ocean depths, and to identify assumptions within the explored parameters that require further constraining (through further field campaign, laboratory analysis or model experimentation).





Firstly, $[Nd]_p/[Nd]_d$ is systematically varied in six sensitivity simulations ($[Nd]_p/[Nd]_d$ ranging 0.001-0.006), these values are
based upon results from similar modelling schemes (Rempfer et al., 2011; Arsouze et al., 2009; Gu et al., 2019) and considers
the few direct observations of $[Nd]_p/[Nd]_d$ (Jeandel et al., 1995; Stichel et al., 2020; Zhang et al., 2008). Here, and based upon
simulations undertaken when validating the scheme, alongside estimates from previous optimised Nd isotope schemes
(Arsouze et al., 2009; Rempfer et al., 2011; Gu et al., 2019; Pöppelmeier et al., 2020a), $f_{sed}$ is fixed at $4.5 \times 10^9$ g yr$^1$ throughout.

Secondly, $f_{sed}$ is varied in four sensitivity simulations ($f_{sed}$ ranging $1.5 \times 10^9$-$6.0 \times 10^9$ g yr$^{-1}$), using values based upon previous
recent estimates of a global sediment flux to seawater ($3.3 \times 10^9$–$5.5 \times 10^9$ g yr$^{-1}$; Gu et al., 2019; Pöppelmeier et al., 2020b;
Rempfer et al., 2011), and encompassing a larger parameter space in order to explore the sensitivity of Nd distributions.
Notably, our $f_{sed}$ sensitivity studies alter the percentage contribution of the sediment Nd flux to the total Nd flux to seawater
from 66% (where $f_{sed}= 1.5 \times 10^9$ g yr$^{-1}$) to 89 % (where $f_{sed} = 6.0 \times 10^9$ g yr$^{-1}$). In all simulations $[Nd]_p/[Nd]_d$ is fixed at 0.003,
based upon reasons outlined for the $[Nd]_p/[Nd]_d$ sensitivity simulations.

**Table 3: Suite of FAMOUS simulations designed to assess the sensitivity of simulated $[Nd]_d$ and $\varepsilon_{Nd}$ distributions to two systematically
varied parameters: reversible scavenging efficiency ($[Nd]_p/[Nd]_d$) and the global rate of direct Nd transfer from sediment to ocean
water ($f_{sed}$). Simulation name refers to the title given to each sensitivity simulation in this manuscript, and the simulation identifier
refers to the unique five-letter Met Office identifier (which, for example, can be used to call down full experiment details from the
NERC PUMA facility: puma.nerc.ac.uk).**

| Simulation name | Simulation identifier | $f_{sed}\,(g(Nd)\,yr^{-1})$ | $[Nd]_p/[Nd]_d$ |
|---|---|---|---|
| *Varying $[Nd]_p/[Nd]_d$* | | | |
| *EXPT_RS1* | XPDAI | $4.5 \times 10^9$ | 0.001 |
| *EXPT_RS2* | XPDAD | $4.5 \times 10^9$ | 0.002 |
| *EXPT_RS3* | XPDAH | $4.5 \times 10^9$ | 0.003 |
| *EXPT_RS4* | XPDAE | $4.5 \times 10^9$ | 0.004 |
| *EXPT_RS5* | XPDAF | $4.5 \times 10^9$ | 0.005 |
| *EXPT_RS6* | XPDAG | $4.5 \times 10^9$ | 0.006 |
| *Varying $f_{sed}$* | | | |
| *EXPT_SED1* | XPDAL | $1.5 \times 10^9$ | 0.003 |
| *EXPT_SED2* | XPDAM | $3.0 \times 10^9$ | 0.003 |
| *EXPT_SED3* | XPDAH | $4.5 \times 10^9$ | 0.003 |
| *EXPT_SED4* | XPDAN | $6.0 \times 10^9$ | 0.003 |

Dissolved seawater Nd in all simulations are initialised from zero and integrated for at least 9,000 years under constant pre-
industrial boundary conditions to allow the deep ocean circulation and marine Nd cycle to reach steady state, which we define
as being when the Nd inventory becomes [near] constant with time (< 0.0025 % change per 100 years). All the presented
results refer to or show the centennial mean from the end of the 9,000-year simulations.





# 3 Results and discussion

**3.1 Model sensitivity to reversible scavenging efficiency ($[Nd]_p/[Nd]_d$)**

The first set of sensitivity experiments test the response of simulated $[Nd]_d$ and $\varepsilon_{Nd}$ to a systematic variation of the reversible scavenging tuning parameter ($[Nd]_p/[Nd]_d$) while all other parameters are kept constant. In the first 2,000 years, the Nd inventory increases exponentially in all simulations (Fig. 9). *EXPT_RS1* continues to increase rapidly until the end of the experiment, but after 2,000 years for simulations *EXPT_RS2* to *EXPT_RS6*, the rate of global Nd accumulation begins to slow

as the inventories tend towards equilibrium. By year 6,000, all experiments with a global Nd inventory $< 8.0 \times 10^{12}$ g reach steady state and remain so until the end of the experiment (9,000 years). We therefore deem these simulations to have reached an acceptable equilibrium state. We explain why *EXPT_RS1* and *EXPT_RS2* do not reach equilibrium below, but otherwise, because of their unrealistic condition after 9,000 years (i.e. they reach an Nd inventory far past the target inventory of $4.2 \times 10^{12}$ g; Tachikawa et al. (2003)), these two simulations are largely omitted from further discussion and analysis.


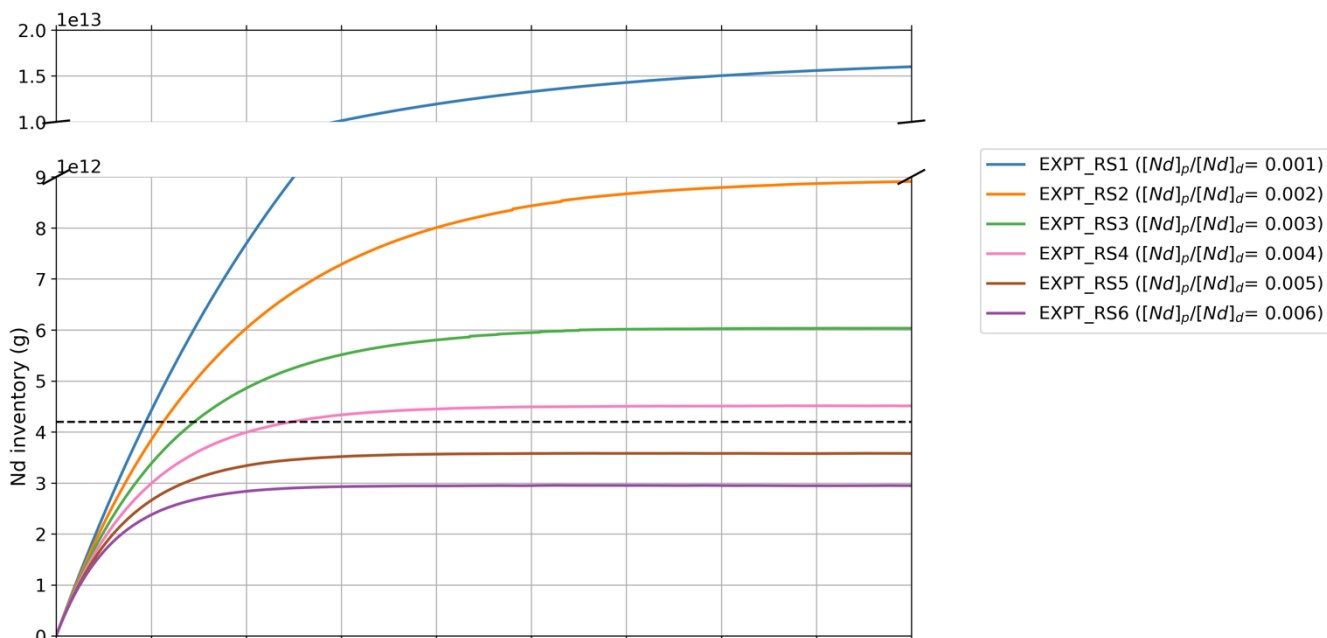

**Figure 9: Global Nd inventory (g) simulated with different values for the reversible scavenging tuning parameter, $[Nd]_p/[Nd]_d$, as indicated. Dashed line represents the estimated global marine Nd inventory of $4.2 \times 10^{12}$ g from Tachikawa et al. (2003) used as an approximate target for our simulations.**

Despite total Nd flux to seawater being kept constant, varying the scavenging efficiency ($[Nd]_p/[Nd]_d$) leads to different Nd

inventories and residence times (Table 4), consistent with previous studies (Siddall et al., 2008; Arsouze et al., 2009; Rempfer et al., 2011; Gu et al., 2019). A higher $[Nd]_p/[Nd]_d$ increases both the Nd scavenging efficiency and removal via sedimentation



by enabling a larger fraction of seawater Nd to adsorb onto particles, in turn leading to a lower Nd inventory and a lower residence time (where; residence time = Nd inventory/total Nd flux).

**Table 4: Overview of simulations exploring model sensitivity to the reversible scavenging tuning parameter $[Nd]_p/[Nd]_d$. Displaying**
**global mean absolute error (MAE) for $[Nd]_d$ and $\varepsilon_{Nd}$.**

| Simulation | $f_{sed}$ ($\times$ $10^9$ g yr$^{-1}$) | $[Nd]_p/[Nd]_d$ | Total Nd flux ($\times$ $10^9$ g yr$^{-1}$) | Nd inventory ($\times$ $10^{12}$ g) | Residence time (years) | $MAE_{[Nd]}$ (n = 3471) | % within 10 pmol kg$^{-1}$ | $MAE_{\varepsilon Nd}$ (n = 2136) | % within 3 $\varepsilon_{Nd}$-units |
|---|---|---|---|---|---|---|---|---|---|
| *EXPT_RS1* | 4.5 | 0.001 | 5.27 | 10.6 | 3037 | 44.46 | 6 | 3.34 | 48 |
| *EXPT_RS2* | 4.5 | 0.002 | 5.27 | 8.91 | 1691 | 15.96 | 41 | 3.11 | 53 |
| *EXPT_RS3* | 4.5 | 0.003 | 5.27 | 6.03 | 1145 | 10.13 | 58 | 2.88 | 57 |
| *EXPT_RS4* | 4.5 | 0.004 | 5.27 | 4.51 | 856 | 9.66 | 62 | 2.67 | 60 |
| *EXPT_RS5* | 4.5 | 0.005 | 5.27 | 3.58 | 679 | 10.52 | 56 | 2.46 | 64 |
| *EXPT_RS6* | 4.5 | 0.006 | 5.27 | 2.95 | 559 | 11.7 | 50 | 2.27 | 70 |

Neodymium in *EXPT_RS1*, which has the lowest $[Nd]_p/[Nd]_d$, has a residence time of 3,037 years. This is much larger than the global ocean overturning time of 1,500 years, resulting in an Nd inventory of $1.06 \times 10^{13}$ g after 9,000 years, almost a factor of four larger than that of the estimated Nd budget in the oceans of $4.2 \times 10^{12}$ g from Tachikawa et al. (2003). The $[Nd]_p/[Nd]_d$, and consequently the sink in *EXPT_RS1*, is too small to balance the input of Nd from the sources, causing such high Nd to
accumulate in the simulation, which, as a result, does not reach steady state (rate of change 2.11% per 100 years at the end of the simulation; Fig. 9). Thus, *EXPT_RS1* returns the largest $MAE_{[Nd]}$ and $MAE_{\varepsilon Nd}$ from these simulations, indicating the worst model-data fit, especially for representing $[Nd]_d$ measurements. *EXPT_RS2*, although tending towards equilibrium at the end of the 9,000-year simulation (Fig. 9), just fails to reach steady state as defined above, still experiencing a rate of change of 0.003% per 100 years, highlighting that the rate of Nd removal from the ocean in this model configuration is still too low.
*EXPT_RS6*, the experiment with the largest $[Nd]_p/[Nd]_d$, returns the lowest $MAE_{\varepsilon Nd}$. However, reaching only $2.95 \times 10^{12}$ g, the final Nd inventory is too low to be considered as capturing Nd cycling well. On balance, we surmise that *EXPT_RS4* has the most reasonable combination of prognostic skill in terms of simulated Nd inventory ($4.51 \times 10^{12}$ g) and residence time (856 years). This simulation, with $[Nd]_p/[Nd]_d$ of 0.004, shows the best balance between Nd accumulation and removal, and hence returns the lowest $MAE_{[Nd]}$ alongside a moderately well performing $MAE_{\varepsilon Nd}$ that falls in the middle of the range of results.

In contrast to our model, the schemes by Rempfer et al. (2011) and Gu et al. (2019) required a lower $[Nd]_p/[Nd]_d$ of 0.001 and 0.0009 respectively for their optimised experiment with Nd inventories of $\approx 4.2 \times 10^{12}$ g. Despite having similar scavenging schemes, a direct comparison of the parameter values used in the different Nd isotope modelling studies is difficult to make. This is because the divergence in sensitivity to reversible scavenging efficiency can be attributed to a combination of the



differing magnitude and spatial distributions of model biogeochemical particle fields and Nd inputs, which are also partly
controlled by the different architecture and horizontal resolution of the physical models. In other words, every study has run a
different experiment, and so the results are not directly comparable. We thus propose that a future modelling protocol for
intercomparing different global Nd isotope schemes would be well suited to exploring these differing sensitivities
comprehensively.


The different values of $MAE_{[Nd]}$ and $MAE_{\varepsilon Nd}$ across our sensitivity experiments (Table 4) demonstrates the distinctive and
uncoupled behaviour of $[Nd]_d$ and $\varepsilon_{Nd}$ within the ocean, as broadly described by the Nd paradox, indicating that different
processes govern the global distributions of each. In our results, increasing the efficiency of vertical cycling improves model-
data fit for $\varepsilon_{Nd}$ in a global sense, reducing $MAE_{\varepsilon Nd}$ to a minimum of 2.27 where $[Nd]_p/[Nd]_d$ is 0.006. However, the story is

more complicated for $[Nd]_d$ performance, where increasing $[Nd]_p/[Nd]_d$ from 0.001 up to 0.004 also reduces $MAE_{[Nd]}$ (to a
minimum of 9.66 pmol kg$^{-1}$), but then subsequent increases in $[Nd]_p/[Nd]_d$ (0.005 and 0.006) worsen model-data fit for $[Nd]_d$
because the sink term becomes too strong, removing too much Nd from the ocean, as reflected in the low accumulated Nd
inventories ($3.58 \times 10^{12}$ g and $2.95 \times 10^{12}$ g respectively).

Generally, simulated $[Nd]_d$ distributions in *EXPT_RS4* match observational data well (Fig. 10). The lowest concentrations
occur in the surface layers, and deep water $[Nd]_d$ increases along the global circulation pathway and with increasing age of
water masses, with lowest $[Nd]_d$ in the North Atlantic Ocean and the highest concentrations in the deep North Pacific Ocean
(Bertram and Elderfield, 1993; van de Flierdt et al., 2016; Tachikawa et al., 2017). These $[Nd]_d$ distributions are consistent
with previous schemes that also take into account reversible scavenging (Arsouze et al., 2009; Gu et al., 2019; Oka et al., 2021;

Rempfer et al., 2011; Siddall et al., 2008). Thus, we may infer that the model scheme in FAMOUS does have the broad
capability of representing the physical processes governing global marine $[Nd]_d$ distributions. However, the scheme does tend
to simulate a too pronounced global vertical $[Nd]_d$ gradient (Fig. 10 and  Supplementary Information: Fig. S7 for major ocean
basin averaged depth profiles), a feature reported in previous similar model schemes (e.g. Arsouze et al., 2009; Gu et al., 2019),
indicating that the representation of processes governing vertical $[Nd]_d$ does not yet fully capture all processes occurring in the

ocean, leading to an underestimation of $[Nd]_d$ in the surface ocean and overestimation at abyssal depths.





**Figure 10: Global volume-weighted distributions of $[Nd]_d$ (left) and $\varepsilon_{Nd}$ (right) in simulation *EXPT_RS4* split into four different depth bins, (a-b) shallow (0-200 m), (c-d) intermediate (200-1,000 m), (e-f) deep (1,000-3,000 m), and (g-h) deep abyssal ocean (>3,000 m). Water column measurements from within each depth bin (Osborne et al., 2017, 2015; GEOTRACES Intermediate Data Product Group, 2021) are superimposed as filled circles using the same colour scale.**

Overestimation of the $[Nd]_d$ at depth may be caused by biases within the simulated biogenic particle fields. Direct and comprehensive global observations of sinking particles fluxes – the central driver of ocean biogeochemical cycling – remain fundamentally difficult to obtain (Dunne et al., 2007), and our particle fluxes may be inaccurate as a consequence (see Sect. 2.3.4 for assumptions and respective limitations). It also seems likely that the reversible scavenging parameterisations, which





are simple by design due to incomplete understanding (also Sect. 2.3.4), restrict the model's ability to precisely capture all aspects of the measured Nd distributions. Additionally other particles such as Fe and Mn oxides and hydroxides, not considered here, may also play an important role for scavenging of Nd (e.g., Bayon et al., 2004). Further observational evidence of the processes involved and their importance for Nd cycling by combined particulate and dissolved measurements and laboratory

experiments (e.g., Stichel et al., 2020; Pearce et al., 2013; Rousseau et al., 2015; Wang et al., 2021), plus experimentation within a modelling framework, may help to improve this limitation.

However, the largest model-data disparities for $[Nd]_d$ occur in the shallow ocean (above 200 m) at specific locations close to continental margins where Nd is input to the ocean through major point sources that are not well resolved by the model. This includes continental margins in the Labrador Sea (where simulated $[Nd]_d$ is 3 pmol kg$^{-1}$ compared to measured $[Nd]_d$ of

70 pmol kg$^{-1}$) and the Sea of Japan (where simulated $[Nd]_d$ is 2 pmol kg$^{-1}$ compared to measured $[Nd]_d$ of 50 pmol kg$^{-1}$), for example. Such low $[Nd]_d$ in the surface layers may be exacerbated by operational constraints in the scheme, such as the extensive and immediate dilution of point sourced Nd across the whole of its containing grid cell combined with the instantaneous nature of simulated reversible scavenging, which may be much faster in the model than would occur normally.

Nonetheless, consistent with compilations of water column measurements (Tachikawa et al., 2017; van de Flierdt et al., 2016),

overall global distributions of simulated $\varepsilon_{Nd}$ are broadly most unradiogenic in the North Atlantic and more radiogenic in the North Pacific (Fig. 10), with intermediate values in the Southern and Indian Oceans. The most unradiogenic $\varepsilon_{Nd}$ occurs in the surface layers of the Hudson Bay and Labrador Sea regions, and they closely match measured data ($\varepsilon_{Nd}$ = -18). However, the most radiogenic $\varepsilon_{Nd}$, simulated in the surface layers of marginal regions in the North and equatorial western Pacific ($\varepsilon_{Nd}$ = -3), is significantly lower than measured ($\varepsilon_{Nd}$ = +3), and in the central and North Pacific; particularly above 1,000 m, simulated $\varepsilon_{Nd}$

is -7, but measurements are closer to -1 $\varepsilon_{Nd}$. In fact, these specific comparisons, which demonstrate a good match in the North Atlantic and weaker performance in the North Pacific, are congruent with a more general trend in the simulations. That is, at the basin scale, the magnitude of the $\varepsilon_{Nd}$ gradient from Pacific to Atlantic is underestimated by the model, and presents a familiar bias as seen in previous Nd isotope schemes (Arsouze et al., 2009; Rempfer et al., 2011; Pöppelmeier et al., 2020a; Jones et al., 2008; Gu et al., 2019). This is mainly due to the simulated Pacific being too unradiogenic (basinal mean $\varepsilon_{Nd}$ of -

7.5) compared to measured water samples ($\varepsilon_{Nd}$ = -4), while simulated and measured basinal mean Atlantic $\varepsilon_{Nd}$ values are in much better agreement ($\varepsilon_{Nd}$ = -11 and -12.5 respectively; Supplementary Information Fig. S7).

The discrepancy in simulated and measured Pacific $\varepsilon_{Nd}$ values may be amplified by sparse sampling that is biased towards shallow radiogenic continental regions and volcanic island areas, in which case, model performance is better than we are able to assess. In addition, and as highlighted in Sect. 2.4, $MAE_{\varepsilon Nd}$ is biased towards the Atlantic (45% of all observational data),

compared to the Pacific (29%). Consequently, due to biases in measurement density, a simulation better representing the Atlantic is more strongly favoured in the adopted cost function than one better representing the Pacific at the expense of the





Atlantic. One further explanation for why our simulated Pacific $\varepsilon_{Nd}$ is so much lower than recorded in modern water measurements is that the model boundary conditions, specifically the marine sediment source of Nd (taken directly from Robinson et al., 2021), may not be sufficiently radiogenic; a point we will return to later (Sect. 3.2).

Simulated $[Nd]_d$ depth profiles in all the reversible scavenging sensitivity experiments (Fig. 11) generally (though not always) exhibit similar depth profiles to the observational data. The best model-data fit is seen especially within depth profiles in the Pacific and Southern Ocean, and notably more so under higher $[Nd]_p/[Nd]_d$. The largest model-data offsets across all sensitivity experiments in terms of both the magnitude and the depth gradients in $[Nd]_d$ occur in the North Atlantic Ocean. In particular, the large near-surface concentrations are not resolved, with the largest disparities occurring under higher scavenging

parameterisations, where simulated $[Nd]_d < 10$ pmol kg$^{-1}$ in the upper 1,000 m are too low compared to observed concentrations around 18-21pmol kg$^{-1}$.

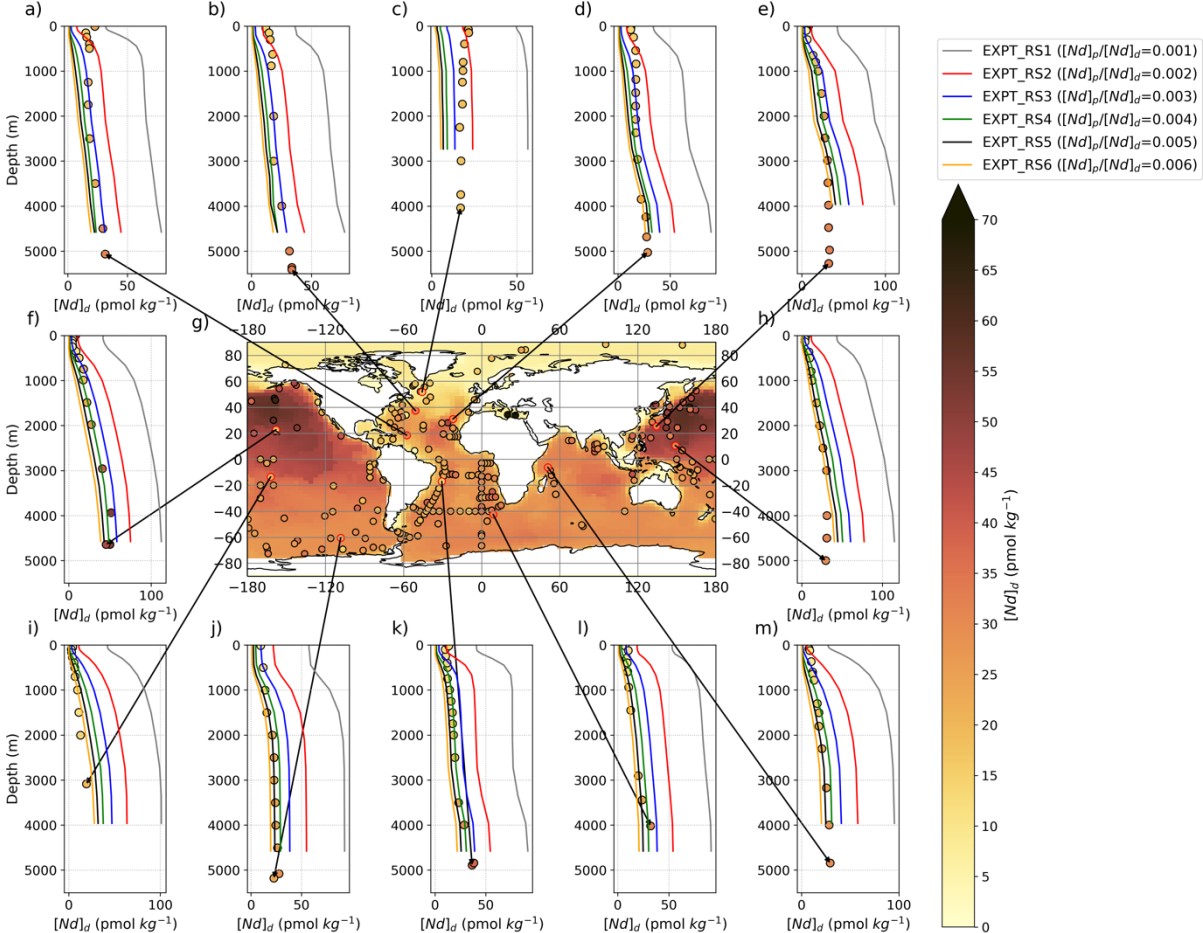

**Figure 11: Central panel (g) displays $[Nd]_d$ at the seafloor in simulation *EXPT_RS4* (100-year mean from the end of the run), with superimposed water column measurements (Osborne et al., 2017, 2015; GEOTRACES Intermediate Data Product Group, 2021) from ≥ 3,000 m shown by filled coloured circles on the same colour scale. Surrounding panels (a-f) and (h-m) display depth profiles**





**of simulated (coloured lines, one per sensitivity simulation with varied $[Nd]_p/[Nd]_d$ and measured (filled circles) $[Nd]_d$. Larger shifts in the $[Nd]_d$ between simulations highlight regions most sensitive to the efficiency of reversible scavenging.**

Interestingly, we see a different sensitivity to varying $[Nd]_p/[Nd]_d$ in $[Nd]_d$ (Fig. 11) compared with $\varepsilon_{Nd}$ (Fig. 12) within different ocean basins and depths. For $[Nd]_d$, we find that all ocean basins are sensitive to the parameterisation of reversible scavenging efficiency (i.e., wider divergence between the sensitivity for each depth profile), particularly at depths below 1,000 m, which

is broadly consistent with the findings reported in previous work by Rempfer et al. (2011). However, Siddall et al. (2008) showed a strong sensitivity of Pacific $[Nd]_d$ alongside a weak sensitivity of Atlantic $[Nd]_d$ to the reversible scavenging efficiency, the differences were attributed by the authors to the dominance of vigorous Atlantic advective lateral transport, whereas we demonstrate similar sensitivities in the Pacific, Atlantic, Indian and Southern Oceans (Fig. 11). The more simplified fixed surface boundary conditions applied in Siddall et al. (2008) are not influenced by changing $[Nd]_p/[Nd]_d$, unlike

the fluxes in the scheme presented here and by Rempfer et al. (2011), which may explain the contradictory response of the Atlantic basin across these studies.

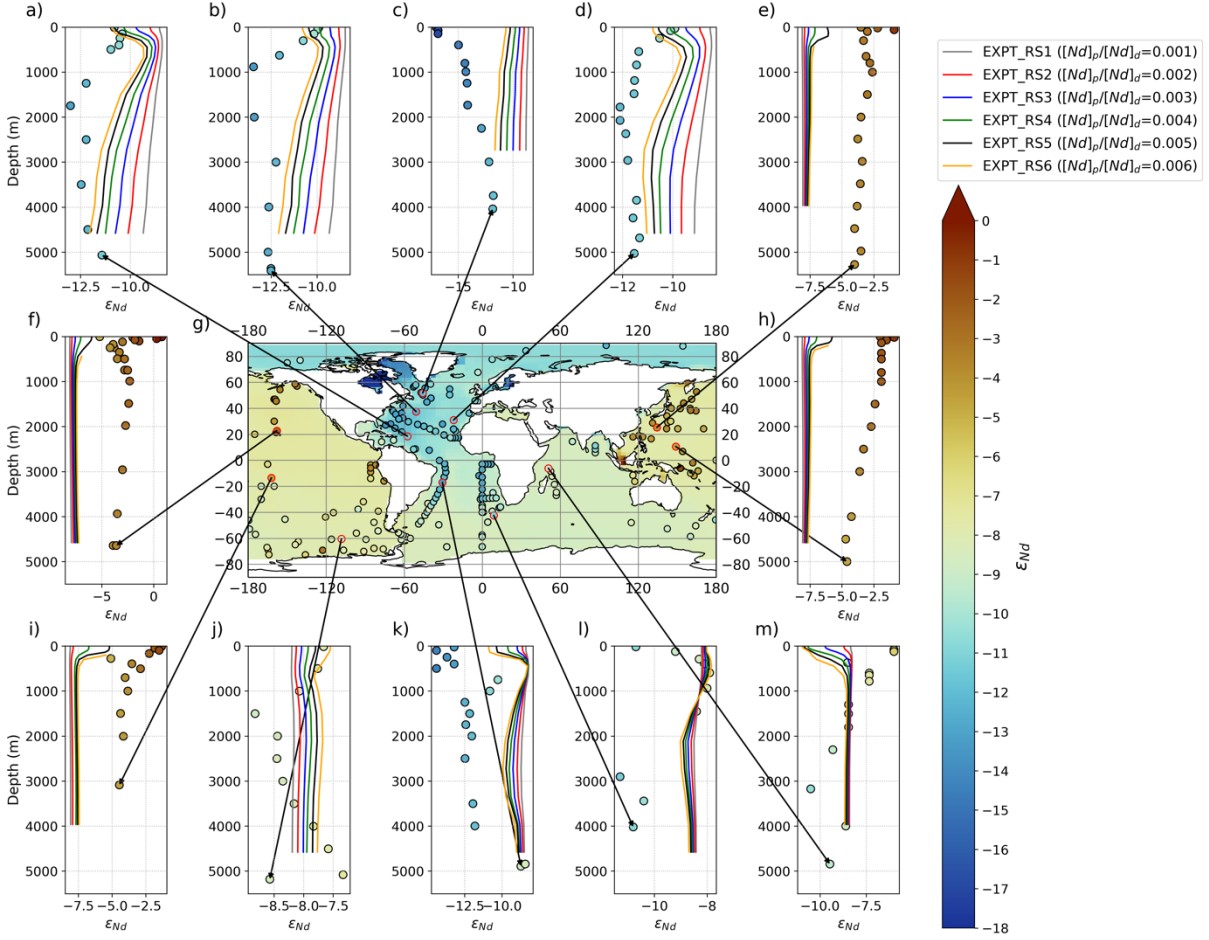

**Figure 12: Central panel (g) displays $\varepsilon_{Nd}$ at the seafloor in simulation *EXPT_RS4* (100-year mean from the end of the run), with superimposed water column measurements (Osborne et al., 2017, 2015; GEOTRACES Intermediate Data Product Group, 2021)**





**from ≥ 3,000 m shown by filled coloured circles on the same colour scale. Surrounding panels (a-f) and (h-m) display depth profiles of simulated (coloured lines, one per sensitivity simulation with varied [$Nd$]$_p$/[$Nd$]$_d$) and measured (filled circles) ε$_{Nd}$. Larger shifts in the ε$_{Nd}$ between simulations highlight regions most sensitive to the efficiency of reversible scavenging.**

For ε$_{Nd}$, the response to varying scavenging efficiency has varied effects across depths and between ocean regions, indicating a more complex relationship between how reversible scavenging can delineate global ε$_{Nd}$ distributions in comparison to [$Nd$]$_d$. The North Atlantic is the most sensitive basin to changes in reversible scavenging (registered by the greater ε$_{Nd}$ profile shifts

between different experiments, Fig. 12a-d), particularly at depths below 2,000 m, indicating that reversible scavenging is important, there, for governing the simulated ε$_{Nd}$ signal of deep-water masses. We attribute this sensitivity to the fact that the Atlantic Ocean experiences strong convection and is surrounded by continental margins. As such, the basin is subject to substantial continental Nd inputs to the upper layers of the ocean, and the shallow ε$_{Nd}$ signal is transferred to the ocean interior, setting the unradiogenic deep ocean signature. Thus, a stronger reversible scavenging efficiency is needed to trap regional ε$_{Nd}$

provenance signals locally.

Typical depth profiles from measurements in the sub-tropical North Atlantic show contrasts in ε$_{Nd}$ that co-vary with the presence of major water masses (coloured circles in Fig. 12a-b). Across all sensitivity experiments, there is relative consistency with depth for the profile, here (e.g., Fig. 12b), ranging from -10 to -12 under increasing scavenging efficiency, which acts to drive more localised unradiogenic signals. This simulated uniformity can be partly explained by the lack of abyssal AABW,

which does not extend past ~ 20° N (Fig. 3). This insufficient AABW production and penetration into the Atlantic are known limitations of FAMOUS (Dentith et al., 2019; Smith, 2012), although these biases are reduced in our *control* compared to the previous studies (Sect. 2.2). Thus, the seawater basin below the surface mixed layer comprises only North Atlantic water, and in so doing, the model will not resolve the AABW signal inferred from measurements at depth in higher latitudes. Shifts in ε$_{Nd}$ between the sensitivity studies in this region therefore relate to reversible scavenging efficiency changing the ε$_{Nd}$ of NADW.

Higher up the water column, the more unradiogenic NADW seen in the seawater measurements is conspicuously absent (note the ε$_{Nd}$ minima in subtropical North Atlantic measurements ~1,000-2,000 m deep, Fig. 12a-b), even though we know NADW does reside there in the model (see Fig. 3 for verification). We may relate this to the high latitude North Atlantic ε$_{Nd}$ being too radiogenic to tag NADW with its characteristically unradiogenic ε$_{Nd}$ signal, particularly around the mouth of the Labrador Sea (Fig. 10a-b). This could be exacerbated by a dampening of the unradiogenic NADW ε$_{Nd}$ from a relatively radiogenic seafloor

benthic flux along the water flow path (Fig. 6). Consequently, despite a close correlation to seawater ε$_{Nd}$ in the subtropics and lower-NADW (where measurements of ε$_{Nd}$ are -12.4 and simulated is -12), upper-NADW end member ε$_{Nd}$ is not sufficiently unradiogenic, even under the highest reversible scavenging efficiency ([$Nd$]$_p$/[$Nd$]$_d$= 0.006) where simulated ε$_{Nd}$ is -12 in comparison with seawater records -13.2 ε$_{Nd}$ (Lambelet et al., 2016).

In contrast, ε$_{Nd}$ in the Pacific Ocean is least sensitive to changes in [$Nd$]$_p$/[$Nd$]$_d$, particularly below 500 m (Fig. 12e-f, h-i).

Depth variability is greatest in the equatorial Pacific, where ε$_{Nd}$ decreases from -2.5 at the surface to -5 at depth, but, overall,



simulated vertical $\varepsilon_{Nd}$ gradients are small in the Pacific, consistent with seawater measurements. This is expected, due to the absence of major ocean convection and ventilation, meaning the Pacific contains an older, more homogenised pool of water in comparison to the Atlantic, and thus, water masses are far less distinct. This does make it possible for reversible scavenging to convey a more localised surface signal into the interior of the Pacific under a lower $[Nd]_p/[Nd]_d$ than can be achieved in the
Atlantic, because the localised $\varepsilon_{Nd}$ signal does not get dispersed via convection as rapidly.

Our less sensitive response of Pacific $\varepsilon_{Nd}$ to reversible scavenging efficiency contrasts with results from Rempfer et al. (2011), who found a greater response of Pacific $\varepsilon_{Nd}$ (compared to the Atlantic). We attribute this difference primarily to the spatial variation in the sediment Nd flux, which, in the study from 2011, is constrained to shallower radiogenic continental sources with no external marine Nd source below 3,000 m. In comparison, our deep seafloor wide sediment source governs simulated
$\varepsilon_{Nd}$ distributions in the intermediate-deep Pacific, due to its larger flux relative to that transported vertically via particle scavenging and dissolution, or horizontally by the sluggish convection of the Pacific.

On the whole, the relatively greater sensitivity of $\varepsilon_{Nd}$ in the surface Pacific, compared with the deep Pacific, to the reversible scavenging efficiency produces a more radiogenic signal closer to that of the radiogenic measurements. This is likely caused by the downward transportation of radiogenic surface inputs to subsurface layers through scavenging. The stronger response
in the upper ocean layers is consistent with previous modelling (e.g. Gu et al., 2019), and occurs due to the inherent presence of larger particle fluxes and therefore greater influence of reversible scavenging here compared to the deeper ocean, where particle dissolution acts to reduce particle concentrations and their associated scavenging mechanisms. One further aspect to note, is that under a high sink ($[Nd]_p/[Nd]_d = 0.006$), in some Pacific regions $[Nd]_d$ in the surface layers tends towards zero. This causes numerical instabilities in modelled Nd ratios and thus, where $[Nd]_d < 0.2$ pmol kg$^{-1}$, $\varepsilon_{Nd}$ is meaningless and so is
masked-out from the results shown by Fig. 12, to avoid misinterpretation.

The simulated depth profile in all experiments in the Indian Ocean matches the observed intermediate $\varepsilon_{Nd}$ signal of -8 between 500-2,000 m (Fig. 12m). However, the more radiogenic $\varepsilon_{Nd}$ signal of -6 in the surface layers is not captured, nor the shift in $\varepsilon_{Nd}$ below 2,000 m to more unradiogenic values reaching a minimum of -10.5 at 3,000 m, with all experiments simulating a relatively uniform $\varepsilon_{Nd}$ with depth. Under higher reversible scavenging parameters for this depth profile (*EXPT_RS4-*
*EXPT_RS6*), the surface concentrations are too low (simulated $[Nd]_d = 3$ pmol kg$^{-1}$ and observed $[Nd]_d = 8$ pmol kg$^{-1}$), indicating that the model is not fully resolving either the concentration or the $\varepsilon_{Nd}$ from a surface flux here, and scavenging may be too intense at the surface. In the deeper ocean ($\approx 3,000$ m), these simulations show the model represents the $[Nd]_d$ profiles better, but is missing an unradiogenic exchange of $\varepsilon_{Nd}$ at depth, where there is an insensitivity to reversible scavenging. This could point to an unradiogenic sediment source or exchange that is misrepresented by the bulk sediment boundary conditions
and sediment source assumptions (e.g., our application of a global seafloor sediment source of Nd irrespective of sedimentary characteristics).





Interestingly, apart from the Indian Ocean, a model configuration with more efficient scavenging generally tends to produce results closest to the observational data (Table 4). In the Indian Ocean, seawater $\varepsilon_{Nd}$ distributions are subject to monsoon systems, which facilitate the [seasonal] delivery of large riverine fluxes of Nd to seawater, for example from the Ganges and

Brahmaputra Delta (Gupta and Naqvi, 1984). Large freshwater fluxes also deliver large amounts of freshly eroded and labile sediment to the continental margins, which likely contribute significantly to governing marine $\varepsilon_{Nd}$ distributions through boundary particle-seawater exchange processes.

In the Southern Ocean, simulations with $[Nd]_p/[Nd]_d \geq 0.003$ broadly match the general measured $\varepsilon_{Nd}$ at depths above 1,000 m (decreasing with depth from -7.8 at the surface to -8.2 at 1,000 m). Below this, and down to 3,000 m, observed diversions in

$\varepsilon_{Nd}$ are not captured in any of the sensitivity experiments. This discrepancy can be attributed to the simulation of quite homogenous AABW throughout the water column in the region, which represents a physical bias of FAMOUS (Dentith et al., 2019; Smith, 2012). Specifically, in the Pacific sector of the Southern Ocean (Fig. 12j), the model cannot resolve the measured unradiogenic spike at 1,500 m, which captures the distinct presence of lower Circumpolar Deep Water (CDW, $\varepsilon_{Nd} = -8.4 \pm 1.6$: Lambelet et al., 2018) formed from mixing of Atlantic, Pacific and Indian sourced waters.


The main conclusion to be made regarding model sensitivity to varying the reversible scavenging tuning parameter $[Nd]_p/[Nd]_d$, is that scavenging and removal via sedimentation is necessary to balance the simulated input sources from dust, rivers and seafloor sediment, and enables the scheme to reach equilibrium around reasonable Nd inventories. We find that reversible scavenging is an important physical process that enhances the $\varepsilon_{Nd}$ gradient between oceans by maintaining localised basinal

$\varepsilon_{Nd}$ signals throughout the water column. The strength of this process is particularly important for maintaining the simulated unradiogenic $\varepsilon_{Nd}$ in the well ventilated North Atlantic Ocean, but less important for the more stagnant modern Pacific Ocean.

Nonetheless, parameterising reversible scavenging efficiency alone cannot account for the correct trends and magnitude within $\varepsilon_{Nd}$ gradients observed between basins and in depth profiles. Currently, the scheme assumes that all particles reaching the

seafloor via reversible scavenging are buried in the sediment, and as such are decoupled from the seafloor sediment source. A future evolution of the scheme could explore the dissolution of authigenic sedimentary phases on the seafloor during diagenesis, to investigate how this may influence the $\varepsilon_{Nd}$ distributions of the benthic flux. Moreover, our results demonstrate the importance of further constraining other aspects of marine Nd cycling under a holistic framework, including surface inputs from dust and river sources, sediment-seawater exchanges and the redistribution and mixing via physical ocean circulation

which also governs global $[Nd]_d$ and $\varepsilon_{Nd}$ distributions. Furthermore, we note that the scheme described here may not be fully resolving the end member $\varepsilon_{Nd}$ of different water masses due to an imperfect representation of the sources of Nd to seawater (i.e., the model boundary conditions and strength of the source fluxes). Thus, in some instances, the scheme carries inappropriate/dampened $\varepsilon_{Nd}$ signals, coupled alongside particular structural model biases in the physical circulation (e.g.,





limited AABW intrusion in the North Atlantic), although in the case of the latter point, we note that this is not a limitation of

the presented scheme, and that the implementation can be useful for identifying such physical biases.

### 3.2 Model sensitivity to Nd flux from the sediment ($f_{sed}$)

The second tranche of sensitivity simulations tests the response of $[Nd]_d$ and $\varepsilon_{Nd}$ to systematically varying the total Nd flux from the sediment ($f_{sed}$), while all other parameters were kept constant. In this experiment, Nd accumulates rapidly from the start of the simulations (Fig. 13) and tapers off thereafter to varying degrees depending on the rate of accumulation. For

*EXPT_SED1* and *EXPT_SED2*, the rate of increase in the Nd inventory begins to reduce by 1,000 years, as these simulations, which have the smallest sediment source ($f_{sed}$), approach steady state. By year 2,000 the rate of Nd accumulation in *EXPT_SED3* and *EXPT_SED4* reduces, and these simulations also reach steady state, albeit above that of the target global inventory reference ($4.2 \times 10^{12}$ g). By year 6,000 all $f_{sed}$ sensitivity experiments have reached steady state ($< 0.0025$ % change per 100 years).

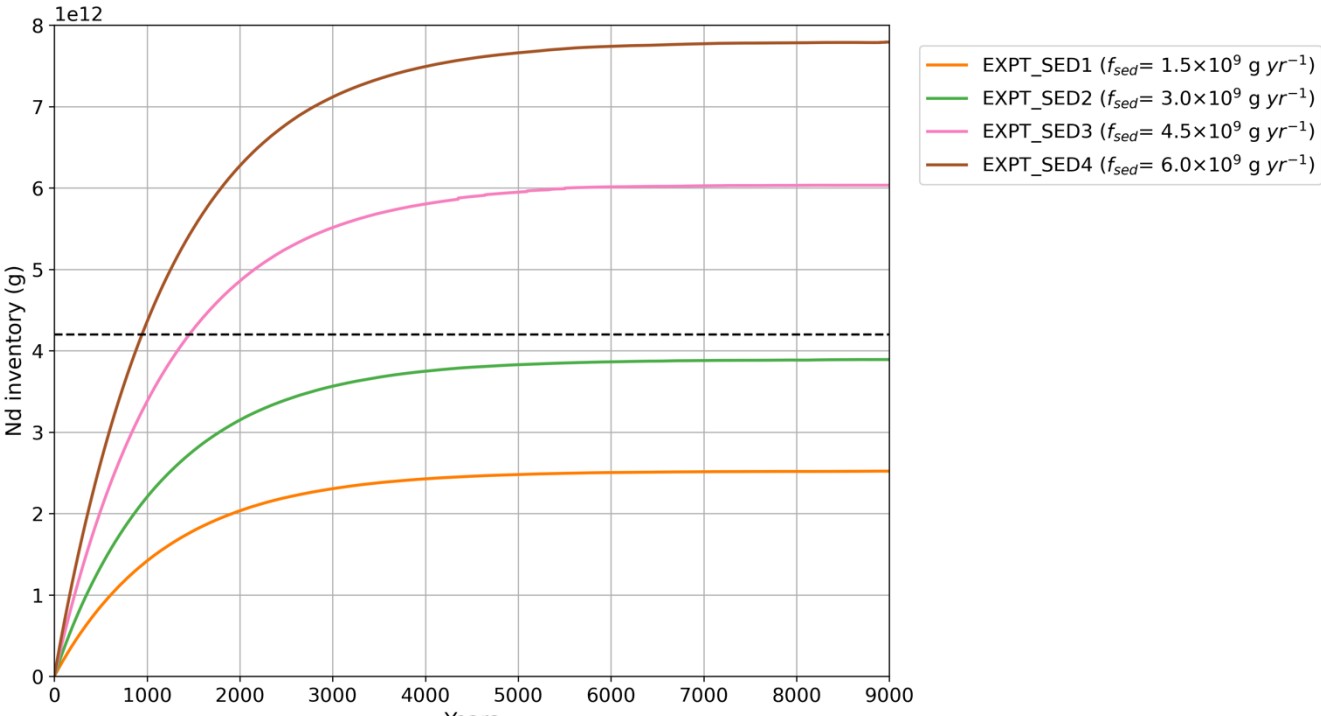

**Figure 13: Global Nd inventory (g) simulated with different values for the total sediment flux tuning parameter ($f_{sed}$) as indicated. Dashed line represents the estimated global marine Nd inventory of $4.2 \times 10^{12}$ g from Tachikawa et al. (2003) used as an approximate target for the simulations.**

Table 5 summarises the final global mean marine Nd inventory and residence time of the simulations, and reports their skill according to our global cost functions, MAE$_{[Nd]}$ and MAE$_{\varepsilon Nd}$. Modifying the sediment flux while keeping the sink term

$[Nd]_p/[Nd]_d$ constant returns a varied equilibrium Nd inventory across the suite of simulations (Fig. 13). However, without





changing the scavenging efficiency (i.e. Nd sink), there is a more moderate range of 40 years difference in the $f_{sed}$ simulations' Nd residence time, consistent with the findings of Rempfer et al. (2011). In general, the relatively low scavenging efficiency of all of the $f_{sed}$ sensitivity simulations ($[Nd]_p/[Nd]_d = 0.003$, see Sect. 3.1 for context) yields long residence times greater than 1,000 years.

**Table 5: Overview of simulations exploring model sensitivity to the total sediment Nd flux tuning parameter ($f_{sed}$). Displaying global mean absolute error ($MAE$) for $[Nd]_d$ and $\varepsilon_{Nd}$.**

| Simulation | $f_{sed}$ ($\times 10^9$ g yr$^{-1}$) | $[Nd]_p/[Nd]_d$ | Total Nd flux ($\times 10^9$ g yr$^{-1}$) | Nd inventory ($\times 10^{12}$ g) | Residence time (years) | $MAE_{[Nd]}$ (n = 3471) | % within 10 pmol kg | $MAE_{\varepsilon Nd}$ (n = 2136) | % within 3 $\varepsilon_{Nd}$ - units |
|---|---|---|---|---|---|---|---|---|---|
| *EXPT_SED1* | 1.5 | 0.003 | 2.27 | 2.5 | 1110 | 10.71 | 53 | 2.71 | 60 |
| *EXPT_SED2* | 3.0 | 0.003 | 3.77 | 3.89 | 1032 | 7.96 | 73 | 2.82 | 62 |
| *EXPT_SED3* | 4.5 | 0.003 | 5.27 | 6.03 | 1145 | 10.13 | 58 | 2.88 | 57 |
| *EXPT_SED4* | 6.0 | 0.003 | 6.77 | 7.79 | 1150 | 13.23 | 47 | 2.93 | 55 |

*EXPT_SED1*, which has the lowest $f_{sed}$ and hence the lowest total Nd flux to the ocean, returns the smallest total Nd inventory of $2.5 \times 10^{12}$ g (only half the total estimated by Tachikawa et al., 2003*)*, but did well with $\varepsilon_{Nd}$ distributions, returning the lowest

global $MAE_{\varepsilon Nd}$. Conversely, *EXPT_SED4*, which has the largest $f_{sed}$, results in the greatest Nd inventory ($7.8 \times 10^{12}$ g), producing both the worst $MAE_{[Nd]}$ and $MAE_{\varepsilon Nd}$.

Compared to varying the reversible scavenging efficiency, varying $f_{sed}$ drives relatively discrete changes in Nd distributions, as demonstrated by the minor differences in $MAE_{\varepsilon Nd}$ between sensitivity experiments. This makes sense, since varying $f_{sed}$ has no direct impact on the relative distribution of $^{143}$Nd and $^{144}$Nd once it is in the water, it only acts to change the fractional

contribution from each specific Nd source, e.g. an enhanced $f_{sed}$ reduces the fraction of total Nd flux coming from dust and rivers (which are inputs constrained to the surface and point sources close to the continents), concentrating the flux across the global seafloor, and vice versa. However, in all simulations, and consistent with previous studies, the sediment Nd source to seawater remains the major source, and $f_{sed}$ would need to be reduced much more to greatly influence $MAE_{\varepsilon Nd}$.

Overall, from the simulations in this study *EXPT_SED2* demonstrates the best skill at reaching the target Nd inventory

($3.89 \times 10^{12}$ g compared to the $4.2 \times 10^{12}$ g target). It returns the lowest $MAE_{[Nd]}$ because it achieves the most balanced Nd source and sink terms, and although the simulation does not represent the lowest $MAE_{\varepsilon Nd}$, the range across the $f_{sed}$ experiment



is small (2.71 to 2.93; Table 5) and it does simulate the highest percentage of simulated $\varepsilon_{Nd}$ within 3-$\varepsilon_{Nd}$ units of measurements (62%).

Altogether, simulated $[Nd]_d$ in *EXPT_SED2* matches the general observational data trend of Nd concentration increasing with depth (Fig. 14), with good model-data fit especially in the upper 1,000 m across all ocean basins (except the North Atlantic, discussed below). However, in the deep layers of the North Pacific (below 3,000 m) simulated $[Nd]_d$ is underestimated (36 pmol kg$^{-1}$ compared to 50 pmol kg$^{-1}$ from seawater measurements). This underestimation of $[Nd]_d$ at depth can be explained by a combination of having a seafloor sediment source at the lower end of our range ($f_{sed}$ is $3.0 \times 10^9$ g yr$^{-1}$) and slow release from reversible scavenging ($[Nd]_p/[Nd]_d$ of 0.003; our experiments suggest 0.004 may be a more suitable efficiency to use, Sect 3.1),

highlighting the importance of both non-conservative processes in governing deep $[Nd]_d$ distributions, especially in the deep North Pacific.





**Figure 14: Global volume-weighted distributions of [*Nd*]*d* (left) and ε*Nd* (right) in simulation *EXPT_SED2* split into four different depth bins, (a-b) shallow (0-200 m), (c-d) intermediate (200-1,000 m), (e-f) deep (1,000-3,000 m), and (g-h) deep abyssal ocean (>3,000 m). Water column measurements from within each depth bin (Osborne et al., 2017, 2015; GEOTRACES Intermediate Data Product Group, 2021) are superimposed as filled circles using the same colour scale.**

Moreover, the longer lifetime of simulated Nd in the *EXPT_SED2* ocean (1,032 years) compared to that estimated by previous work (360-800 years; Gu et al., 2019; Rempfer et al., 2011; Siddall et al., 2008; Tachikawa et al., 2003) means that Nd becomes well mixed in the deep ocean (below 1,000 m), homogenising the ε*Nd* signal. This causes the observed inter-basin gradients (a critical feature in the use of Nd as an ocean circulation tracer) to become severely damped, particularly away from direct input of fresh reactive phases with distinctive ε*Nd* (Robinson et al., 2021; Abbott et al., 2019). However, where including a reduced





sediment source (e.g. compared to *EXPT_SED3* and *EXPT_SED4*) increases the relative importance of dust and river inputs, such as in the shallow Atlantic (Lambelet et al., 2016), simulated $\varepsilon_{Nd}$ matches unradiogenic measurements with reasonable skill. Consistent with earlier studies (e.g., Arsouze et al., 2009; Jones et al., 2008), the largest model-data disparities occur in

the North and equatorial Pacific, where simulated $\varepsilon_{Nd}$ is far too unradiogenic compared to the observational data, pointing to a number of processes that may be better optimised in our Nd scheme. For example, a larger (and also more radiogenic) sediment source (i.e., greater $f_{sed}$) may be needed, particularly around shallow-intermediate marginal settings; a suggestion also supported by the too low $[Nd]_d$. Additionally, increased scavenging and a lower simulated residence time would improve the representation of localised Nd isotope signatures.

Total Nd concentrations and isotopic distributions show different responses to varying $f_{sed}$. We find that $[Nd]_d$ is sensitive to $f_{sed}$ across all ocean basins (i.e., a wide divergence in the depth profiles show in Fig. 15), mostly at depths below 500 m where there is no direct influence from river and dust inputs and the relatively large area of the deep abyssal seafloor as an Nd interface becomes important (particularly with low reversible scavenging).

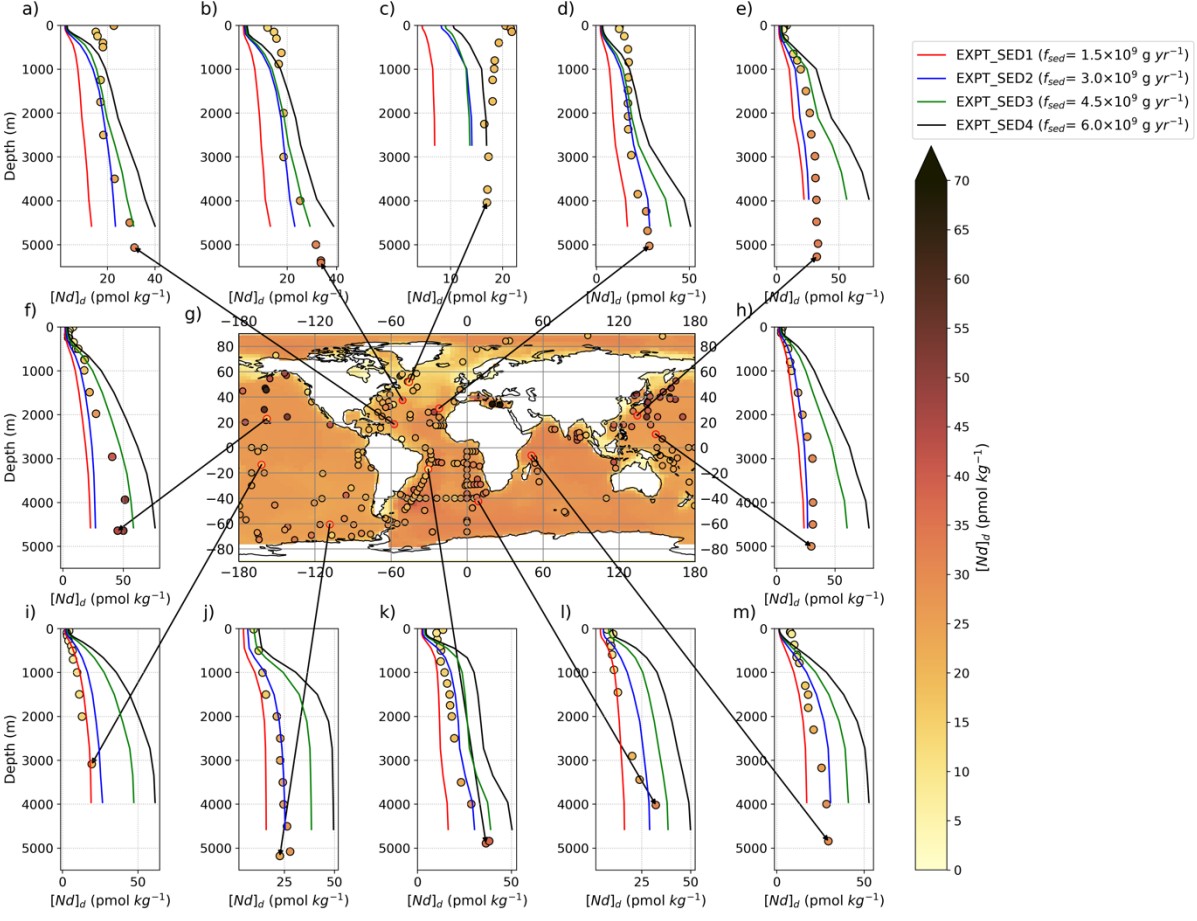





Typically, the best model-data fit for [*Nd*]$_d$ depth-profiles is achieved under lower $f_{sed}$ ($1.5 \times 10^9$ to $3.0 \times 10^9$ g yr$^{-1}$), and particularly in the Pacific, Indian and Southern Ocean (Fig. 15). This indicates that under a relatively low reversible scavenging parameter, a corresponding smaller flux of Nd from the sediment is needed to balance the marine Nd budget, and by inference, a higher $f_{sed}$ would be more appropriate with stronger reversible scavenging fluxes. Correspondingly, under high $f_{sed}$ and particularly in the deeper interior of the ocean (below 3,000 m), simulated [*Nd*]$_d$ diverges to higher concentrations than

measured due to too much Nd being sourced to the deep ocean from the sediment. Additionally, in these high $f_{sed}$ scenarios, the strength of the sediment source obscures horizontal seafloor [*Nd*]$_d$ gradients across basins (see the near-uniform seafloor [*Nd*]$_d$ in Fig. 15g), masking the influence of reversible scavenging, which is controlled by the location and dissolution of particle fields and is important for governing [*Nd*]$_d$ patterns (Sect. 3.1). Rempfer et al. (2011) found the opposite. In their study, doubling and halving $f_{sed}$ both resulted in well pronounced nutrient-like profiles of [*Nd*]$_d$. This highlights the impact of having

a global sediment source unrestricted in depth (our study) compared to limiting that flux to the upper 3,000 m (Rempfer et al., 2011) paired with a more efficient reversible scavenging (Rempfer et al., 2011) that allows the biogenic particle adsorption and desorption processes to dominate the deep-water distributions.

Similar to the reversible scavenging sensitivity experiment (Sec. 3.1), the largest simulated offsets between simulated and measured [*Nd*]$_d$ under all $f_{sed}$ experiments occur in the North Atlantic and sub-tropical Atlantic at depths above 1,000 m, where,

even under the largest sediment fluxes, simulated [*Nd*]$_d$ is too low. The depth profile south of Greenland (Fig. 15c) shows the greatest sensitivity to varying $f_{sed}$ in the upper 1,000 m, with *EXPT_SED4*'s simulated surface concentrations of 10 pmol kg$^{-1}$ ($f_{sed} = 6.0 \times 10^9$ g yr$^{-1}$) being closest to the measured concentrations of 22 pmol kg$^{-1}$. By implication, the accurate representation of sediment fluxes is required to reproduce upper ocean [*Nd*]$_d$ in this region, and an enhanced sediment flux alone cannot account fully for the observed high surface concentrations. Either, a combination of the surface and near-surface fluxes

resolved here are too diluted in the model (possibly due to grid box resolution), or the model is missing a significant surface/near-surface Nd source. Previous schemes have likewise simulated too low surface [*Nd*]$_d$ in the North Atlantic, likely also due to difficulties in representing highly localised and variable surface features in global models (Gu et al., 2019; Rempfer et al., 2011).

Whereas [*Nd*]$_d$ is sensitive to $f_{sed}$ globally, the height of ε$_{Nd}$ sensitivity is more regional (Fig. 16). The Atlantic, Indian and

Southern Ocean ε$_{Nd}$ are significantly more sensitive to changes in the bulk seafloor sediment Nd flux than the Pacific Ocean. Overall, differences in $f_{sed}$ tend to drive whole depth profile shifts of low magnitude in ε$_{Nd}$, this contrasts the findings of Rempfer et al. (2011), who varied a margin constrained $f_{sed}$ and reported deep water ε$_{Nd}$ (below 1,000 m) were affected less



than in our results. If this conflicting difference in deep ocean sensitivity to $f_{sed}$ is because of the applied spatial distributions of a sediment Nd source to seawater ($< 3,000$ m for Rempfer et al. (2011), all depths for us), then further constraints of sediment

Nd fluxes across space and time are crucial when interpreting ocean circulation from $\varepsilon_{Nd}$. Interestingly, there is not a linear/direct response of $\varepsilon_{Nd}$ to changing $f_{sed}$. Broadly and globally, the largest $f_{sed}$ value of $6.0 \times 10^9$ g yr$^{-1}$ (*EXPT_SED4*) leads to more radiogenic $\varepsilon_{Nd}$ and an intermediate $f_{sed}$ value of $3.0 \times 10^9$ g yr$^{-1}$ (*EXPT_SED2*) leads to the most unradiogenic $\varepsilon_{Nd}$ shift. The largest response in $\varepsilon_{Nd}$ to changing $f_{sed}$ occurs with a sediment flux between $3.0 \times 10^9$ yr$^{-1}$ and $4.5 \times 10^9$ g yr$^{-1}$. Increasing the flux beyond this produces a very weak response, indicating a threshold in $\varepsilon_{Nd}$ sensitivity that results from an already

dominant sediment source encompassing over 85% of the total Nd flux to seawater.

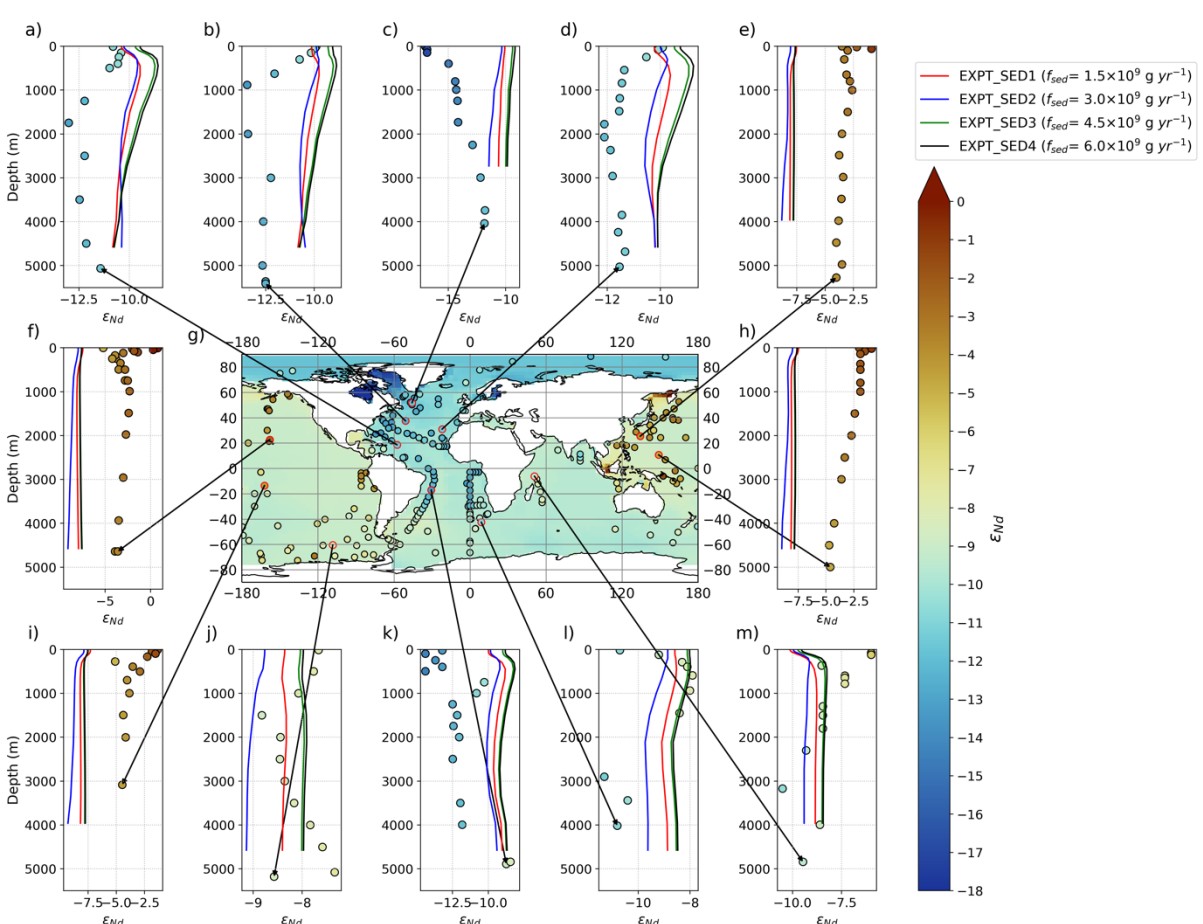

**Figure 16: Central panel (g) displays $\varepsilon_{Nd}$ at the seafloor in simulation *EXPT_SED2* (100-year mean from the end of the run), with superimposed water column measurements (Osborne et al., 2017, 2015; GEOTRACES Intermediate Data Product Group, 2021) from $\geq 3,000$ m shown by filled coloured circles on the same colour scale. Surrounding panels (a-f) and (h-m) display depth profiles**

**of simulated (coloured lines, one per sensitivity simulation with varied $f_{sed}$) and measured (filled circles) $\varepsilon_{Nd}$. Larger shifts in the $\varepsilon_{Nd}$ between simulations highlight regions most sensitive to the magnitude of the seafloor sediment source.**





Once again, the most notable $\varepsilon_{Nd}$ model-data mismatch occurs within the depth profiles of the North Atlantic. Due to major rivers delivering a high [Nd] load to the Atlantic (Fig. 5c), in their vicinity, simulations with low $f_{sed}$ are conditioned towards riverine $\varepsilon_{Nd}$, providing the typical unradiogenic $\varepsilon_{Nd}$ signature of NADW and demonstrating the important balance between

surface inputs and a sediment flux for simulating $\varepsilon_{Nd}$ in the deep North Atlantic (Fig. 16a-d). Conversely, enhancing $f_{sed}$ increases the fraction of Nd supplied to the ocean from the bulk seafloor sediment, which has more uniform, intermediate $\varepsilon_{Nd}$ values in the central North Atlantic (-12.5) in contrast to the more unradiogenic $\varepsilon_{Nd}$ signals of the continental margins and riverine source in the Labrador (-28) and West Atlantic basins (-15.6), with localised near-surface marginal sediment extreme minimums of (-34) in the northern Labrador Sea (Robinson et al., 2021). Greater $f_{sed}$ thus acts to overprint and hence mix away

the more distinct surface $\varepsilon_{Nd}$ signal of NADW gained at their sites of deep-water formation in favour of a more general intermediate $\varepsilon_{Nd}$ signal as it becomes exposed to Nd fluxes along its southward seafloor flow path. Model-measurement disparity at the mouth of the Labrador Sea and south of Greenland in all simulations at shallow and intermediate depths strongly suggests that a specific fraction of the bulk sediment with more unradiogenic $\varepsilon_{Nd}$ than is captured by Robinson et al. (2021) is interacting with seawater in place of the bulk sediment $\varepsilon_{Nd}$ signal (Fig. 17). This is further supported by recent core-top particle-

seawater interaction investigations of the region (Blaser et al., 2016). It therefore follows that either existing seafloor sediment measurements do not characterise this region well, or a deep ocean benthic flux, or at least, an indiscriminate whole-ocean floor flux, is neither reasonable nor necessary (in fact, it is counterproductive) for providing accurate $\varepsilon_{Nd}$ tagging of Atlantic seawater.

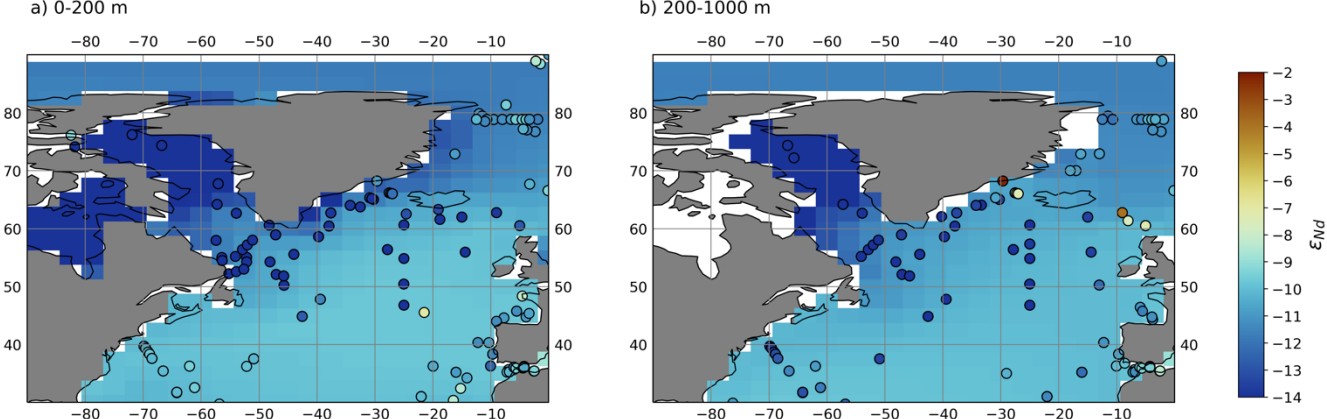

**Figure 17: Volume-weighted distributions of $\varepsilon_{Nd}$ in simulation *EXPT_SED2* split into two different depth bins, (a) shallow (0-200 m),**
**and (b) intermediate (200-1,000 m) within the North Atlantic and Labrador Sea basins. Water column measurements from within each depth bin (Osborne et al., 2017, 2015; GEOTRACES Intermediate Data Product Group, 2021) are superimposed as filled circles on the same colour scale.**

In contrast to the Atlantic, Pacific Ocean $\varepsilon_{Nd}$ is the least sensitive to varying $f_{sed}$ (Fig. 16e-f, h-i). Characteristically, the Pacific encompasses vast open ocean oligotrophic expanses with low biogenic particle export, in tandem with smaller relative dust

and river sources, which means the seafloor sediment source already dominates the simulated Nd fluxes and distributions, even





under lower sediment fluxes. Furthermore, the bulk detrital $\varepsilon_{Nd}$ (-8) released from the seafloor sediment in the Northwest Pacific (Fig. 6) is the same as the signal from the dust flux off the Asian continent, which contributes significant terrigenous material to the seafloor here (Nakai et al., 1993; Han et al., 2011), which makes it hard to distinguish between these two sources. In the North Pacific, the simulations with higher $f_{sed}$ match the observed $[Nd]_d$ better than those with lower $f_{sed}$ (Fig.
15), but the $\varepsilon_{Nd}$ signal from the bulk seafloor sediment cannot explain the radiogenic $\varepsilon_{Nd}$ observed here (Fig. 16), and so the model likely is not fully capturing the correct source (or the prescribed boundary condition is not correctly representing the distribution and fraction of the sediment phase contributing to the sediment flux) of Nd to seawater. Consistent with previous literature, the greatest imbalances in the Nd isotope budget occur in the reservoirs of the vast deep Pacific (Jones et al., 2008). This could imply that a 'reactivity weighted' sedimentary flux is most important for the deep North Pacific compared with
other basins, providing evidence to support the preferential dissolution of reactive Nd sediment phases in the Pacific under a model of marine Nd cycling dominated by a seafloor sediment source.

To conclude, we surmise that a sediment-seawater flux represents a key major source of $[Nd]_d$ that is particularly fundamental to the intermediate and deep ocean Nd budgets, and as such plays an important role in governing marine Nd cycling. Notably,
we find that $\varepsilon_{Nd}$ is much less sensitive than $[Nd]_d$ to changing the rate of this flux ($f_{sed}$), and a high $f_{sed}$ coupled with a weak reversible scavenging cannot account for $[Nd]_d$ gradients of increasing concentration along the thermohaline circulation in the deep ocean. Indeed, a strong sediment source pushes the ocean towards a more globally uniform and too high $[Nd]_d$ than measurements from the deep ocean suggest. This could be interpreted as evidence against a 'bottom up' model (Haley et al., 2017) with constant benthic flux across the whole seafloor in favour of the more distinct $[Nd]_d$ distributions that may be
achieved under a 'top-down' driven model with greater dominance of reversible scavenging. As such, although a benthic flux *can* represent the depth profiles of linearly increasing $[Nd]_d$ with depth, reversible scavenging is *necessary* and should not be considered a secondary process in controlling the global thermohaline variations in deep water $[Nd]_d$. Nonetheless, we acknowledge that employing a more horizontally nuanced benthic flux tied to local environmental and sedimentary conditions may introduce more spatial patterning in simulated $[Nd]_d$, reducing the importance of the reversible scavenging terms.

Certainly, these sensitivity simulations demonstrate openly that the bulk $\varepsilon_{Nd}$ of seafloor detrital sediment (Robinson et al., 2021) cannot be considered fully representative of the $\varepsilon_{Nd}$ composition of the sediment that is interacting with seawater in all instances. They highlight the need for observational and experimental quantification of the broad mobile Nd phases globally, and their $\varepsilon_{Nd}$ signal, as well as constraints on the spatial distribution of such a benthic flux (e.g., identifying where and under what environmental conditions a benthic flux occurs, and at what strength). The model's response to $f_{sed}$ sets the stage for
further testing of global sedimentary $\varepsilon_{Nd}$ on marine Nd cycling, providing the foundation for resolving the inherent complex multitude of processes.



## 4 Summary and conclusions

In this study, we describe the implementation of Nd isotopes ($^{143}$Nd and $^{144}$Nd) into the ocean component of the FAMOUS GCM (ND v1.0), providing a powerful tool designed for comprehensively exploring global marine Nd cycling, especially
through representing explicit non-conservative processes to explore the extent of their influence on global seawater Nd distributions. We present a new reference equilibrium pre-industrial simulation (XPDAA) with improved basin scale physical ocean circulation performance relative to a standard version (XFHCC; Smith, 2012), though we note that for this study it was not important to evaluate the performance of the surface climate. Our Nd isotope scheme starts from previous Nd isotope implementations (Rempfer et al., 2011; Gu et al., 2019; Pöppelmeier et al., 2020a; Arsouze et al., 2009; Siddall et al., 2008),
but revisits and updates Nd sources, sinks and tracer transformation in line with increased observations and recent findings relating to global marine Nd cycling. Our model represents the main features of [Nd] and $\varepsilon_{Nd}$ well, although there is a tendency to simulate an over-pronounced vertical [Nd] gradient, and to produce a too unradiogenic $\varepsilon_{Nd}$ signal in the Pacific.

The presented sensitivity experiments demonstrate that the Nd isotope scheme in the FAMOUS GCM is sensitive to Nd source and biogeochemical processes. Model sensitivity to reversible scavenging efficiency demonstrates its importance for
determining the increase in Nd concentration with depth and along the circulation pathway. Moreover, reversible scavenging acts to homogenise $\varepsilon_{Nd}$ vertically in the water column, enhancing regional basinal gradients in simulated $\varepsilon_{Nd}$ by maintaining the localised provenance signal. On the other hand, a seafloor benthic flux, a term referring to a multitude of processes encompassing boundary exchange (Lacan and Jeandel, 2005), submarine groundwater discharge (Johannesson and Burdige, 2007), and a benthic flux released from pore waters (Abbott et al., 2015a), presents a major deep ocean source of Nd, the
magnitude of which governs horizonal seafloor Nd concentrations across ocean basins. The weak sensitivity of simulated $\varepsilon_{Nd}$ in the deep North Pacific implies that with a seafloor-wide benthic flux of marine Nd, the $\varepsilon_{Nd}$ of the sediment flux, as captured by the bulk $\varepsilon_{Nd}$, is not a true representation in all instances of the labile sediment phase interacting with seawater in this basin. Alternatively, it may also indicate that there is a significant missing source of radiogenic Nd to seawater, likely of volcanic origin; or possibly a combination of both explanations. Furthermore, model-data mismatch at the mouth of the Labrador Sea
suggests that the labile sediment Nd phases interacting with seawater in the northern North Atlantic are considerably more unradiogenic than captured by the bulk sediment.

Exploring in detail the behaviour of simulated $[Nd]_d$ and $\varepsilon_{Nd}$ distributions also highlighted some of the structural limitations of the model (e.g., difficulties representing highly localised and surface features) and influential biases in the physical ocean circulation (e.g., limited northward intrusion of AABW in the North Atlantic). These results provide the groundwork for a
future comprehensive optimisation of the marine Nd isotope scheme in FAMOUS. In the first instance, we suggest calibration of the key tuning parameters ($[Nd]_p/[Nd]_d$ and $f_{sed}$) to achieve target Nd inventories and residence times. Additionally, it would be beneficial to obtain additional observational constraints on the broad labile sediment $\varepsilon_{Nd}$ interacting with seawater across different seafloor regions, including constraining the exchange between authigenic and detrital sediment phases during early



diagenesis. This would improve the boundary conditions we can feed into the model, giving it the best chance to simulate Nd distributions accurately and making the interpretation of model performance (and thus also of the relative importance of different sources and sinks) more straightforward. Future sensitivity studies could also focus on the influence of river particulate and continental marginal sources on marine Nd in order to provide further insight to (and possibly constrain) the relative importance of these inputs, as opposed to a predominantly benthic seafloor-wide source.

Implementing Nd isotopes in a fast GCM provides a useful tool for exploring model-scheme sensitivities and uncertain marine biogeochemical processes. This framework allows for performing the long integrations necessary to spin-up ocean physics and biogeochemistry, which can be simulated for multiple time periods (e.g. in palaeo studies) in equilibrium (i.e. with fixed forcing, Haywood et al., 2016; Lunt et al., 2017) and transient (i.e. with temporally evolving forcing, e.g. Ivanovic et al., 2016; Menviel et al., 2019) scenarios. Alternatively, or in addition, large ensembles of simulations can be run for [re]calibrating (i.e., 'tuning') the model or quantifying uncertainty in the inputs (boundary conditions, parameter values etc.). In this way, it becomes possible to build on new knowledge gained by running the scheme – such as the identification of physical ocean biases – to improve the model and refine what we know about the respective sources and sinks of Nd in the ocean (e.g. the strength and isotopic composition of seafloor fluxes). Furthermore, the sophisticated, complex model physics in FAMOUS and intrinsic coupling of the ocean GCM to an atmosphere GCM and dynamic ice sheet model enables the oceanographic changes and how these may manifest in Nd distributions to be examined in conjunction with associated atmospheric and cryospheric changes, including the feedbacks between the different Earth system components.

This new model scheme can aid in the delivery of more robust applications of $\varepsilon_{Nd}$ as a modern and palaeo-tracer. It provides a platform for dynamic modelling under different modes of marine Nd cycling (e.g., the balancing of 'top down' versus 'bottom up' fluxes) or varied climatic and oceanographic conditions, enabling current hypotheses to be tested rigorously with the aim of constraining Nd cycling under a complex, partially understood marine geochemical system.

**Code Availability**

The code detailing the advances for the marine Nd isotope scheme (ND v1.0) described in this manuscript is available via the Research Data Leeds Repository (Robinson 2022: *https://doi.org/10.5518/1136*) under a Creative Commons Attribution 4.0 International (CCBY 4.0) license. These files are known as code modification (i.e. 'mod') files and should be applied to the original FAMOUS model code, which is protected under UK Crown Copyright and can be obtained from the National Centre for Atmospheric Science (NCAS) Computational Modelling Services (CMS): https://cms.ncas.ac.uk/, with specific FAMOUS documentation available at: *https://cms.ncas.ac.uk/miscellaneous/um-famous/*. All files and corresponding information that needs to be applied to configure each individual simulation presented here are available from the same DOI (above); note that to complete the setup of these simulations, line 4909 in each simulation's tracer.f file needs updating with the corresponding



RS_TUNE value as listed in Table 3. A complete version of the modified code for the EXPT_RS4 simulation using FAMOUS-
MOSES1, including Nd isotope implementation, is archived at the Research Data Leeds Repository, and linked from the DOI
above.

Control candidate simulations for new FAMOUS reference

- XPDAA *control simulation* (0-5,000 years)
- XPDAB *control candidate simulation* (0-5,000 years)
- XPDAC *control candidate simulation* (0-5,000 years)
- XPDEA *control candidate simulation* (0-5,000 years)

Reversible scavenging efficiency ($[Nd]_p/[Nd]_d$) sensitivity simulations

- XPDAI $[Nd]_p/[Nd]_d=0.001$ (0-9,000 years)
- XPDAD $[Nd]_p/[Nd]_d=0.002$ (0-9,000 years)
- XPDAH $[Nd]_p/[Nd]_d=0.003$ (0-9,000 years)
- XPDAE $[Nd]_p/[Nd]_d=0.004$ (0-9,000 years)
- XPDAF $[Nd]_p/[Nd]_d=0.005$ (0-9,000 years)
- XPDAG $[Nd]_p/[Nd]_d=0.006$ (0-9,000 years)

Total Nd source from sediment ($f_{sed}$) sensitivity simulations

- XPDAL $f_{sed}=1.5 \times 10^9$ g yr$^{-1}$ (0-9,000 years)
- XPDAM $f_{sed}=3.0 \times 10^9$ g yr$^{-1}$ (0-9,000 years)
- XPDAH $f_{sed}=4.5 \times 10^9$ g yr$^{-1}$ (0-9,000 years)
- XPDAN $f_{sed}=6.0 \times 10^9$ g yr$^{-1}$ (0-9,000 years)

**Data Availability**

The data are available via the Research Data Leeds Repository (Robinson 2022a: https://doi.org/10.5518/1136)

**Supplementary Material**

The supplement related to this article is to be associated with a DOI, and currently attached as a .zip
(Supplementary_Robinson.et.al_inPrep.zip), containing:
**Supplementary Information**: Documented supplementary text, figures, and tables (Text S1, Table S1-S3, Figures S1-S9)
**Table S4:** Spreadsheet of seawater Nd concentration and isotope measurements and references used to validate model scheme



**simulation_files/:** Folder with all the simulation files needed to run each simulation

**standard_famous_mods/:** Folder containing all standard FAMOUS GCM modification files

## Author contributions

RFI designed the project with SMR, and supervised its completion. SMR wrote and implemented the code with technical input from JCT, RFI and LJG. SMR completed the experiment design, ran the simulations, analysed results, and prepared the manuscript with input from all co-authors. All co-authors contributed scientific expertise throughout. PV designed and ran the

pre-industrial FAMOUS GCM perturbed physics ensemble. FP updated the dust source $\varepsilon_{Nd}$ boundary conditions. YP provided the seawater REE compilation, which was updated by SMR.

## Competing interests

The authors declare that they have no conflict of interest.

## Acknowledgements

SMR was funded by the Natural Environment Research Council (NERC) SPHERES Doctoral Training Partnership (grant number: NE/ L002574/1). LJG was funded by UKRI Future Leaders Fellowship 'SMBGen' (grant number: MR/S016961/1). JCT was supported through the Centre for Environmental Modelling and Computation (CEMAC), University of Leeds. FP was supported by the European Union's Horizon 2020 program (grant number: 101023443). TvdF acknowledges funding from the NERC grant NE/ P019080/1. Our heartfelt thanks go to Dr Jamie D Wilson for valuable discussions when implementing

the reversible scavenging scheme, and to Professor Andy Ridgwell for helpful discussions during the design of code developments.

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
