# Peer review of "Simulating marine neodymium isotope distributions using Nd v1.0 coupled to the ocean component of the FAMOUS-MOSES1 climate model: sensitivities to reversible scavenging efficiency and benthic source distributions"

_EGUsphere, 2022_

## Referee Comment (RC1)

**egusphere-2022-606 Review**

**Title**: Simulating marine neodymium isotope distributions using ND v1.0 coupled to the ocean component of the FAMOUS-MOSES1 climate model: sensitivities to reversible scavenging efficiency and benthic source distributions

**Reviewer**: Benoît Pasquier

**General comments**

This is a welcome and timely manuscript describing a new marine Nd-cycling model embedded in a fast GCM that is well-suited for future exploration and optimization. The authors also present preliminary results of the sensitivity of their model to varying two important parameters of the Nd cycle, which already offer new insights into our understanding of the global Nd cycle and its isotope signature.

The model skill is thoroughly examined through quantitative metrics and expert assessment of the tracer distributions. To the best of my knowledge, the science supporting the model is sound, the context and references are properly presented, and many of the potential caveats of the model are presented.

Although I have not attempted to reproduce the scientific results myself, I commend the authors for making available what seems to be all the necessary code files and data for running the simulations.

The manuscript is structured well, the presentation is clear and easy to follow, and the figures are of high quality.

My biggest issue with the current manuscript is a small one and lies within the angle or sometimes plain omission of some necessary discussions around the caveats of the model.

**Overall, I would recommend the publication of this manuscript after minor revisions.**

Below is the list of minor suggestions and comments,

**Specific comments**

- **On the model's ocean circulation,** I have a few suggestions that the authors might or might not want to consider.

  At the end of the model description (Section 2.1), the authors explain that their choice of the older MOSES version was driven by the bad ocean circulation of the more recent version ("collapsed Atlantic Ocean convection and strong deep Pacific MOC"). Obviously, no ocean circulation model is perfect, and I commend the authors for detailing their choice of ocean circulation model in the following section (2.2), but I think a bit more could help there.

  - Suggestion 1: Add some discussion about how the quality/skill compares to other GCMs. This could be illustrated by "simply" adding other GCMs to Fig. 1. I believe this would help the reader assess the author's model choice

    Note this suggestion comes from my perspective as a data-assimilated ocean circulation user. RMS errors of about 2°C and 0.9 PSU for temperature and salinity, respectively, seem like large

biases, given, e.g., the "old" OCIM1 circulation model (*DeVries and Primeau*, 2011) and its RMS errors of less than about 0.2°C and 0.05 PSU (i.e., about 10–20 times better on that specific metric).

- Suggestion 2: Discuss how well the selected FAMOUS model does in reproducing other circulation tracers (e.g., those mentioned by the authors, $\delta^{13}C$ and $\delta^{14}C$).

- **"bottom-up" vs "top-down"**. The model presented is a "bottom-up" model, which means that roughly 85% of the Nd tracer is injected at the bottom of the ocean. Although that fraction varies from 66% to 89% in their experiments, discussions on the possibility of a potential "top-down" model (where said fraction would go much lower) are sparse. I think a more thorough examination of the possibility that baking-in strong sedimentary fluxes can be a caveat in itself and discussing alternatives in a more balanced way would strengthen the manuscript. (More details in the line-by-line points below)

- **Unit suggestion**. Throughout, maybe by convention or already established precedent, the authors express quantities that I believe could be simplified for clarity. For example, fluxes are expressed in g $yr^{-1}$ but are of the order of $10^9$ g $yr^{-1}$. This begs the question: Why not use Gg $yr^{-1}$? This would remove many "$\times 10^x$". Alternatively, since [Nd] is expressed in pmol $kg^{-1}$, maybe the authors could expres sources and sinks in Mmol $yr^{-1}$ instead of g $yr^{-1}$. (These are just suggestions.)

**Line-by-line suggestions, comments, typos, etc.**

- Eq. (1): While there is a mountain of established precedent publications that have adhered to this $\varepsilon_{Nd}$ notation, the authors might be interested in checking *Coplen, 2011 (doi:10.1002/rcm.5129)* for $\varepsilon_{Nd}$ notation and unit), which argues for writing it in $\delta$ notation, without the superfluous $10^4$ constants, and expressing it in parts per ten thousand (‰):

> $\delta_{Nd} = IR_{sample} / IR_{CHUR} - 1$

For what it is worth, in *Pasquier et al.* (2022), we opted to keep the $\varepsilon_{Nd}$ symbol but expressed the equation without a unit (i.e., without the $10^4$). No change is required here, just pointing at some potential improvements.

- L47 (missing "in"):

> he measured Nd isotope composition of seawater is not actively involved ***in*** marine biological cycling

- L56 (and all other occurrences) the year of our *Pasquier et al.* publication should be 2022 instead of 2021 (and a DOI should be added).

- L156: The *Jones et al.* (2008) citation should be removed (because it is not about the FAMOUS model).

- Fig. 1 (Taylor diagrams) is missing units.

- L258: Not that it is important, but I am curious, as this flew over my head:

> This technique minimises the mathematical error associated with carrying small numbers.

What is the reason here? Isn't FAMOUS written in Fortran and doesn't it deal with floating point arithmetic correctly for small numbers?

- Table 1:

    - The exponent of the yr unit is shoved to the next line (for several rows), slightly reducing readability.
    - As per the specific comment above, maybe better unit choices can improve clarity?

- L275 (and throughout): The context makes it clear that $f_{dust}$ is in grams of Nd. Maybe remove the " (Nd)" in "g(Nd) yr$^{-1}$"?

- L284–286: It is unclear how the additional constraints on the aeolian $\varepsilon_{Nd}$ are applied. It is probably worth expanding/detailing.

- Fig. 5:

    - Panel a: This filled contour map essentially looks bicolor to me. Could a log scale be applied to the colormap to distinguish different river discharge strengths?
    - Panels b and c: While the *Goldstein and Jacobsen* (1987) reference is given, it is unclear how the prescribed riverine $\varepsilon_{Nd}$ and [Nd] gridded datasets are created.
    - Panel d (missing): Could the authors add a map of the resulting riverine Nd source?

- Table 1 + Eq. (7): $S_{river}$ units issue. Substituting the units from Table 1 into Eq. (7) yields

> $m^2 \, yr^{-1}$

instead of

> $kg \, m^{-3} \, yr^{-1}$.

This begs the question: Is the equation correct?

- L396: Side note (not necessary for this manuscript, but could be a nice upstream fix): In *Pasquier et al.* (2022), one of the reasons for capping the north Pacific values of sedimentary $\varepsilon_{Nd}$ was because it appeared as if the source dataset from *Robinson et al.* (2021) had used disconnected seafloor areas during production, with a particularly visible jump along the 180° meridian. Another oddly aligned frontier also appeared in the South Pacific around 165°W:

[Figure]

These disconnected areas probably originated from the lithology type dataset used:

[Figure]

It would be oddly coincidental for those lithology areas to have frontiers that coincide with meridians by chance. Maybe these areas could be fused back and the $\varepsilon_{Nd}$ seafloor dataset updated? (This is not a big critique by any means and I would like to emphatically commend the authors for making such a map/dataset available in the first place!)

- Eqs. (10) and (11) typo: It should be either

  exp$((z - z_{eu}) / ...)$

or

  exp$((z_{eu} - z) / ...)$

depending on the vertical axis ($z$) orientation.

- Fig. 7: Colorbar units should be all upright (some are italic for some reason).

- Eqs. (12) and (14): The sum should be indexing over $\chi$ instead of $i$.

- L479 seems to start a new sentence right after the equation but does not. It is also unclear how $[Nd]_p/[Nd]_d$ is a tunable parameter. (It does not explicitly appear in Eq. (15).) Maybe this is an equation typo? Unsure what fix the authors would want.

- L480: It took me a while to realize that the authors have used $p$ instead of more usual $\rho$ (Greek rho) for seawater density. Could they replace $p$ with $\rho$?

- Eq. (17) Suggestion: Maybe the authors could also report RMSE (root mean square error, as done by *Sidall et al.* (2008) and *Pasquier et al.* (2022)) along MAE. (Also as a suggestion for the future work mentioned elsewhere: squared differences, like the mean square error (MSE), generally work well as the objective function for optimization routines, owing to their quadratic shape.)

- L563: missing minus in exponent: $yr^{-1}$ instead of $yr^1$.

- L564 and throughout: Notation suggestion: Probably clearer to write

  > $(1.5–6) \times 10^9 \ g \ yr^{-1}$

  than

  > $1.5 \times 10^9 – 6.0 \times 10^9 \ g \ yr^{-1}$

  but again,

  > $(1.5–6) \ Gg \ yr^{-1}$

  would be even better in my opinion.

- L575: I am likely wrong but I am unconvinced that all (any?) of the experiments fit that criterion. Back of the envelope calculation means a $(0.0025\% / 100yr)^{-1} = 4Myr$ stability timescale for the global mean [Nd] tendency. Maybe Figure 9 could also show the (centennial) tendencies of the mean [Nd], and prove me wrong (see Fig. 9 point 2 below).

- Fig. 9:

  - Maybe a $y$-axis log scale instead of the broken axis?
  - Maybe plot the tendencies in a separate panel below? It is sort of expected that the global inventory scales inversely with the scavenging strength. Therefore the only new information I am looking for at a glance in Fig. 9 is how quickly the system equilibrates. But then plotting the tendencies directly would be more straight to the point.

- Table 4:

  - would benefit from a smaller font.
  - The residence time of the first row (EXPT_RS1; 3037yr) does not match the formula:

    > residence time = Nd inventory/total Nd flux

- A suggestion: Move the columns for flux, inventory, and age to Table 3, and turn the "mismatch" columns into plots. Better would be detailed scatter plots of every model vs observation data point, for [Nd] and $\varepsilon_{Nd}$ (it can be a simple scatter with transparency or, even better in my opinion, a joint distribution density plot as was done in, e.g., Fig. 7 of *Pasquier et al*, 2022). I suggest this because only the "mismatch" columns (the last four) are conveying new information while the other columns are either constant, redundant (with Table 3 or Fig.9), or simple divisions (the residence time formula).

- L626:

  >  illustrates?

- L628: What about:

  >  the scavenging efficiency

- Figure 10: These are not

  > Global volume-weighted distributions of [Nd] (left) and $\varepsilon_{Nd}$ (right) (...) split into four different depth bins (...)

  Instead, these are maps of

  > [Nd] (left) and $\varepsilon_{Nd}$ (right) (...) vertically averaged over four different depth ranges

- L652–661: What about too strong a sedimentary source? While I agree with all the potential caveats listed in this paragraph, the authors should clarify why they don't consider an overestimate of the benthic source as the potential culprit for an overestimate of deep Nd.

- L662–668: Conversely to the preceding paragraph/point, my first impression is that a potential culprit is not discussed: What about too-weak surface sources? The simulated surface [Nd] underestimates observations beyond the coasts, particularly in the Atlantic (visible in the surface map of Fig. 9 but also in the profiles of Fig. S7). Larger surface point sources combined with a slower scavenging scheme can supply this missing surface Atlantic Nd. But a stronger dust source can, too. (That is what happens in our preliminary parameter space optimization in *Pasquier et al.* (2022): the dust solubility parameter is increased —to unrealistic levels— to better fit the observations.) Otherwise, could it be too small a (vertical) supply by the ocean circulation model? Maybe the authors can discuss these hypotheses (and rule them out)?

- L675 $\varepsilon_{Nd}$ should not denote both the value and the unit. Thus, for consistency, I would remove it there:

  > closer to -1 .

- L687–689: Maybe I missed this: Could it again be a case of surface sources instead? In the Pacific, it is not only that the $\varepsilon_{Nd}$ values are too low, but the vertical [Nd] profile also suggests a lack of surface-originating Nd (Fig. S7). Maybe this is another manifestation of too strong scavenging near pointwise sources near the coast (a large number of observations in the North West Pacific make it hard to see the simulated field underneath)? Or maybe the model is missing a radiogenic Pacific surface source?

- L690: I am probably missing something here, but

  > Simulated $[Nd]_d$ depth profiles in all the reversible scavenging sensitivity experiments (Fig. 11) generally (though not always) exhibit similar depth profiles to the observational data

  seems like an impossible achievement. The Nd inventory precisely scales inversely with the scavenging strength (data from Table 4):

[Figure]

  Although there are some variations in the spatial distributions, [Nd] generally does the same. This means if experiment A "exhibits similar depth profiles to the observational data", then the other experiments cannot all also match the data. Could the authors rephrase this paragraph so that it is clear what is similar? (It cannot be the profiles!)

- L708: What about using "suggests" instead of "demonstrates"? (Some, like me, usually assume "demonstrates" means "proves".)

- L716–725: What about a mention of the fact that increased scavenging efficiency, which means more local trapping of Nd, also means inter-basin separation? That is, the inter-basin $\varepsilon_{Nd}$ gradients are favored by strong scavenging and a short residence time, as confirmed by the relationship between MAE($\varepsilon$) and scavenging strength.

- L728: Where is "here"?

- L818:

  > By year 6,000 all $f_{sed}$ sensitivity experiments have reached steady state (< 0.0025 % change per 100 years).

  Is this correct? Looking at EXP_SED4 around 6000yr still shows a slope

that I estimate to be a change of about 0.5% over 1000yr, i.e., 0.05% / 100yr, which is 20 times more than the advertised threshold. At this stage, I am sure I am missing something therefore I hope that the authors can clarify this! I would again suggest a semilog plot of the total Nd inventory tendencies to accompany the existing plot.

- L837: What about:

> varying $f_{sed}$ drives relatively  **small** changes in Nd **spatial** distributions

---

## Referee Comment (RC2)

[referee-annotated manuscript omitted]

---

## Referee Comment (RC4)

[referee-annotated manuscript omitted]

---

## Community Comment (CC1)

[Figure]

*Figure 1: seafloor εNd and seafloor Nd concentration provided by Robinson et al. (2021)*

---

## Author Comment (AC1)

Correspondence: ee14s2r@leeds.ac.uk

Title: Simulating marine neodymium isotope distributions using ND v1.0 coupled to the ocean component of the FAMOUS-MOSES1 climate model: sensitivities to reversible scavenging efficiency and benthic source distributions

**Summary of Changes**

Blue text below is our response to the reviewer's comments (reproduced in black). Line numbers refer to the tracked-changes version of the manuscript.

**Response to reviewer 1: Benoît Pasquier**

**General comments**

This is a welcome and timely manuscript describing a new marine Nd-cycling model embedded in a fast GCM that is well-suited for future exploration and optimization. The authors also present preliminary results of the sensitivity of their model to varying two important parameters of the Nd cycle, which already offer new insights into our understanding of the global Nd cycle and its isotope signature.

The model skill is thoroughly examined through quantitative metrics and expert assessment of the tracer distributions. To the best of my knowledge, the science supporting the model is sound, the context and references are properly presented, and many of the potential caveats of the model are presented.

Although I have not attempted to reproduce the scientific results myself, I commend the authors for making available what seems to be all the necessary code files and data for running the simulations.

The manuscript is structured well, the presentation is clear and easy to follow, and the figures are of high quality.

We thank the reviewer for his positive comments recognising the genuine value of our work.

My biggest issue with the current manuscript is a small one and lies within the angle or sometimes plain omission of some necessary discussions around the caveats of the model.

We have revised the manuscript to address this point in accord with the detailed suggestions below, and we thank the reviewer for his thorough comments.

Overall, I would recommend the publication of this manuscript after minor revisions.

Below is the list of minor suggestions and comments.

**Specific comments**

On the model's ocean circulation, I have a few suggestions that the authors might or might not want

to consider.

0

At the end of the model description (Section 2.1), the authors explain that their choice of the older MOSES version was driven by the bad ocean circulation of the more recent version ("collapsed Atlantic Ocean convection and strong deep Pacific MOC"). Obviously, no ocean circulation model is perfect, and I commend the authors for detailing their choice of ocean circulation model in the following section (2.2), but I think a bit more could help there.

• Suggestion 1: Add some discussion about how the quality/skill compares to other GCMs. This could be illustrated by "simply" adding other GCMs to Fig. 1. I believe this would help the reader assess the author's model choice

Note this suggestion comes from my perspective as a data-assimilated ocean circulation user. RMS errors of about 2°C and 0.9 PSU for temperature and salinity, respectively, seem like large biases, given, e.g., the "old" OCIM1 circulation model (*DeVries and Primeau*, 2011) and its RMSerrors of less than about 0.2°C and 0.05 PSU (i.e., about 10–20 times better on that specific metric).

Done: we have added the following text to section 2.2, lines 284-288 to add more context to this evaluation, including a comparison to examples of other models whose data are available to us, namely HadCM3 and MIROC. We would not expect FAMOUS to outperform these two models since it is tuned to HadCM3, and both HadCM3 and MIROC4m have slightly higher complexity and higher resolution. However, this comparison demonstrates that the performance of the *control* simulation is comparable to similar/higher-order models, indicating appropriate model skill, which is useful context for understanding the limitations and advantages of the current study given our pragmatic choice to undertake the Nd isotope scheme development with a fast GCM.

We have not added the HadCM3 and MIROC data to Fig. 1 as we prefer to leave that figure focused on the new model results.

We have also added clarification to the model description (2.1) that HadCM3 is the parent model of FAMOUS, including line 235-237: '*FAMOUS is calibrated to the performance of HadCM3, taking the philosophy that this is the most appropriate evaluation target and it is unrealistic to expect the lower resolution, lower complexity model to out-perform its parent model* (Valdes et al., 2017).

Suggestion 2: Discuss how well the selected FAMOUS model does in reproducing other circulation tracers (e.g., those mentioned by the authors,  $\delta^{13}C$  and  $\delta^{14}C$ ).

Having considered it carefully, we have not adopted this suggestion because the control simulations are physically quite different between our simulations, and those of Dentith et al. (2019) and Dentith (2020), who untertook the C-isotope work. The earlier study highlighted that one of the main limitations to FAMOUS's ability to reproduce measured carbon-isotope ratios in the ocean was the over-deep North Atlantic Deep-Water formation and circulation and lack of Southern-sourced water in the abyssal North Atlantic, hence, we adopted a new control configuration. The ocean structure (including AMOC) is very different in the

different studies and therefore a comparison to the previous C-isotope results would not be appropriate. To include the C-isotopes in our new simulations would require many months of further model integration time and a large volume of additional analysis, and we do believe there is value in presenting the Nd implementation documented here as a standalone piece of work, also considering that the journals focus is on the description of model development.

"bottom-up" vs "top-down". The model presented is a "bottom-up" model, which means that roughly 85% of the Nd tracer is injected at the bottom of the ocean. Although that fraction varies from 66% to 89% in their experiments, discussions on the possibility of a potential "top-down" model(where said fraction would go much lower) are sparse. I think a more thorough examination of the possibility that baking-in strong sedimentary fluxes can be a caveat in itself and discussing alternatives in a more balanced way would strengthen the manuscript. (More details in the line-byline points below)

Done: we have revised the text (section 3.1 line 986-1002) to emphasise the explicit caveat that our experiment design assumes a dominant sediment source based on suggestions that the seafloor sediment is the 'missing' (approx. 90%) Nd source (Tachikawa et al., 2003; Rempfer et al., 2011; Gu et al., 2019; Arsouze et al., 2007, 2009) and the more recent evidence that this is mostly coming out from abyssal seafloor sediment (Abbott et al., 2015b, a, 2019; Pöppelmeier et al., 2020; Deng et al., 2022). Our experiment design is specifically geared towards facilitating a discussion within the Nd community on the appropriate emphasis to place on a benthic flux for solving the Nd paradox, since we know this is an area for intense debate.

To explore more thoroughly the 'top down' versus 'bottom up' paradigm would indeed make a useful area for additional study, though it is beyond the scope of the presented work, since we primarily aim here to present the new version of an Nd isotope enabled FAMOUS and explore the sensitivity of the two main parameters/processes that are currently thought to govern marine Nd cycling and yet have only poor constraints. This is a good opportunity to highlight our companion paper to this manuscript (in discussion:

https://egusphere.copernicus.org/preprints/2022/egusphere-2022-937/), where we present an optimized version of the Nd isotope scheme. Exploring the difference in a top down vs bottom up is something to achieve with this optimised model, to extend the work presented in the companion paper, which already begins down that path by assessing the margin vs benthic flux.

(Note, in response to another reviewer's comment, we have rephrased the way we refer to the 'top down' vs 'bottom up' paradigm in this manuscript).

Unit suggestion. Throughout, maybe by convention or already established precedent, the authors express quantities that I believe could be simplified for clarity. For example, fluxes are expressed in g yr-1 but are of the order of 109 g yr-1. This begs the question: Why not use Gg yr-1? This would remove many "× 10x". Alternatively, since [Nd] is expressed in pmol kg-1, maybe the authors could expres sources and sinks in Mmol yr-1 instead of g yr-1. (These are just suggestions.)

Done: units were presented in conventions similar to previous Nd isotope implementation

in GCMs. However, we are happy to update this to Gg yr-1 for easier reading and have made this change throughout, including figures – maybe it will catch on!

Line-by-line suggestions, comments, typos, etc.

Eq. (1): While there is a mountain of established precedent publications that have adhered to this  $\varepsilon_{Nd}$  notation, the authors might be interested in checking *Coplen*, 2011 (doi:10.1002/rcm.5129) for  $\varepsilon_{Nd}$  notation and unit), which argues for writing it in  $\delta$  notation, without the superfluous 104 constants, and expressing it in parts per ten thousand (‰):

 $\delta_{\rm Nd}$  = IRsample / IRCHUR - 1

For what it is worth, in *Pasquier et al.* (2022), we opted to keep the  $\varepsilon_{Nd}$  symbol but expressed the equation without a unit (i.e., without the 104). No change is required here, just pointing at some potential improvements.

Thank you for highlighting this.

L47 (missing "in"):

he measured Nd isotope composition of seawater is not actively involved *in* marine biological cycling

**Done.**

L56 (and all other occurrences) the year of our *Pasquier et al.* publication should be 2022 instead of 2021 (and a DOI should be added).

Done.

L156: The *Jones et al.* (2008) citation should be removed (because it is not about the FAMOUS model).

**Done.**

• Fig. 1 (Taylor diagrams) is missing units.

**Done.**

- L258: Not that it is important, but I am curious, as this flew over my head:
- This technique minimises the mathematical error associated with carrying small numbers. What is the reason here? Isn't FAMOUS written in Fortran and doesn't it deal with floating point arithmetic correctly for small numbers?

No problem! The choice for scaling Nd fluxes in the code was done to make the amounts of Nd in the model easier for humans to work with due to the very small numbers involved in simulating a trace element in a global model. Such scaling is often done (e.g., salinity and reporting values such as  $\varepsilon_{Nd}$ ). We have edited the text (line 336-339) to make this clearer.

- Table 1:
  - The exponent of the yr unit is shoved to the next line (for several rows), slightly reducing readability.
  - As per the specific comment above, maybe better unit choices can improve clarity?

Done (yr on the same line and updated to Gg and Tg).

• L275 (and throughout): The context makes it clear that  $f_{dust}$  is in grams of Nd. Maybe remove the " (Nd)" in "g(Nd) yr-1"?

**Done.**

L284–286: It is unclear how the additional constraints on the aeolian  $\varepsilon_{Nd}$  are applied. It is probably worth expanding/detailing.

**Done.**

. Fig. 5:

• Panel a: This filled contour map essentially looks bicolor to me. Could a log scale be applied to the colormap to distinguish different river discharge strengths?

**Done.**

• Panels b and c: While the *Goldstein and Jacobsen* (1987) reference is given, it is unclear how the prescribed riverine  $\varepsilon_{Nd}$  and [Nd] gridded datasets are created.

Done: figure caption updated to clarify this: 'Figure 5: (a) Simulated river outflow (RIVER) in FAMOUS, (b) major river  $\varepsilon_{Nd}$ , (c) major river [Nd], and (d) the resulting riverine Nd source. The coastal grids in (b) and (c) are prescribed following average [Nd] and  $\varepsilon_{Nd}$  estimates of dissolved river runoff to each of the oceans by Goldstein and Jacobsen (1987; see Table 3).'

• Panel d (missing): Could the authors add a map of the resulting riverine Nd source? Done.

Table 1 + Eq. (7): Sriver units issue. Substituting the units from Table 1 into Eq. (7) yields

m2 yr-1

instead of

kg m⁻³ yr⁻¹.

This begs the question: Is the equation correct?

Done: we corrected the RIVER units in Table 1 to  $g m^2 yr^1$ , and the intext units for the source from the river from kg m-3 yr-1. This is the correct unit as used in the code, and yields the correct units for Eq.(7).

L396: Side note (not necessary for this manuscript, but could be a nice upstream fix): In *Pasquier et al.* (2022), one of the reasons for capping the north Pacific values of sedimentary  $\varepsilon_{Nd}$  was because it appeared as if the source dataset from *Robinson et al.* (2021) had used disconnected seafloor areas during production, with a particularly visible jump along the 180° meridian. Another oddly aligned

frontier also appeared in the South Pacific around 165°W:

These disconnected areas probably originated from the lithology type dataset used

It would be oddly coincidental for those lithology areas to have frontiers that coincide with meridiansby chance. Maybe these areas could be fused back and the  $\varepsilon_{Nd}$  seafloor dataset updated? (This is not a big critique by any means and I would like to emphatically commend the authors for making such a map/dataset available in the first place!)

We take the opportunity to respond to this side-note because it is an interesting discussion, but please note that we do not revise the manuscript, or benthic boundary condition, in light of this comment because it mainly pertains to the previous work. When designing the simulations presented here, we thought about the points the reviewer now raises carefully and decided against making further tweaks to the published seafloor dataset. For some background on the methods for that previous paper and to explain our decision not to smooth over this feature at the date-line: the artificial disconnect across the meridian (and the South Pacific) is an artifact of the high-resolution gridded map characterising the major lithologies of seafloor sediments in the world's ocean basins (Dutkiewicz et al., 2015; see 'Seafloor Lithology Map' in Data Availability), which was used to constrain the interpolation of discrete detrital and pore water measurements to create the seafloor  $\varepsilon_{Nd}$  maps. Here, we adopted the assumption that dominant seafloor lithology types at least partially describe the major sedimentary source and characteristics of detrital  $\varepsilon_{Nd}$ . This lithology map was, at the time of the paper, the most up to date representation of seafloor lithology. However, limitations of the seafloor lithology map included missing coverage in the polar Arctic region and a disconnect across the meridian. In order to create seafloor  $\varepsilon_{Nd}$  maps, and facilitate new schemes testing a global benthic flux, an  $\epsilon_{Nd}$  signature needed to be assigned to all depositional sedimentary environments. We therefore had to make pragmatic (and often difficult) choices in order to best represent the  $\varepsilon_{Nd}$ distributions in abyssal seafloor regions with vast areas of no data and factor in boundaries of the map. In the previous work, we went some way towards correcting for these discontinuities in the gridded lithology file around the international date line using manual adjustments in the coastal areas of the Ross Sea (see supplementary: C10 and SF18) and the east Bering Sea (SF4). However, the Pacific seafloor, covering such a vast area but with limited measurements, proved the most challenging region to represent. To 'manually adjust' to correct for this disconnect across the Pacific would have meant either ignoring lithological bounds from the seafloor lithology map and applying a single mean  $\epsilon_{\mbox{\scriptsize Nd}}$  across the whole of the abyssal Pacific, which arguably would have imposed just as arbitrary value as using the lithology bounds. As such, and in the interest of transparency, we chose for this first evolution of the seafloor  $\varepsilon_{Nd}$  map to minimise the manual-tuning. Most importantly, we hope this data driven and easily reproducible map provides a blue-print for how to [re]make the map with new data, and we provide as much information as possible so that every user can apply their own preferred assumptions and adjustments. We particularly highlighted outstanding questions over labile benthic fluxes and we hope that a future influx of seafloor detrital and importantly pore water  $\varepsilon_{Nd}$  measurements from GEOTRACES as well as from other programs and the wider community will help feed in knowledge to revise the map in a second version. This updated knowledge of the benthic flux would then go hand in hand with a future update to revise the seafloor lithology map, including correcting for the arbitrary bounds across the meridian line.

Eqs. (10) and (11) typo: It should be either

 $exp((z - z_{eu}) / ...)$

or

 $exp((z_{eu} - z) / ...)$

depending on the vertical axis (z) orientation.

**Done: corrected the typos in Eq.(10) and (11) corrected to $exp((z_{eu} - z) / ...)$ : thank you.**

Fig. 7: Colorbar units should be all upright (some are italic for some reason).

Done: italics removed for consistency. However, we prefer to keep the colourbar units horizontal and we think this layout is sufficiently clear.

Eqs. (12) and (14): The sum should be indexing over  $\chi$  instead of *i*.

Done.

• L479 seems to start a new sentence right after the equation but does not. It is also unclear how  $[Nd]_p/[Nd]_d$  is a tunable parameter. (It does not explicitly appear in Eq. (15).) Maybe this is an equation typo? Unsure what fix the authors would want.

Done: we have updated the description of Eq.(15), which removed the incomplete sentence after the equation. We have also added detail in the text below Table 2 to explain the assumption that  $[Nd]_d$  and  $[Nd]_p$  are in equilibrium and defined the parameter which describes the ratio  $([Nd]_p/[Nd]_d)$ , that, based upon these assumptions, is the same irrespective of particle type and determines the scavenging efficiency in the model.

• L480: It took me a while to realize that the authors have used p instead of more usual  $\varrho$  (Greek rho)for seawater density. Could they replace p with  $\varrho$ ?

Done.

Eq. (17) Suggestion: Maybe the authors could also report RMSE (root mean square error, as done by *Sidall et al.* (2008) and *Pasquier et al.* (2022)) along MAE. (Also as a suggestion for the future work mentioned elsewhere: squared differences, like the mean square error (MSE), generally work well as the objective function for optimization routines, owing to their quadratic shape.)
I just stumbled upon this GMD highlight paper on MAE vs RMSE that the authors may find useful: Hodson, T. O.: Root-mean-square error (RMSE) or mean absolute error (MAE): when to use them or not, Geosci. Model Dev., 15, 5481–5487, https://doi.org/10.5194/gmd-15-5481-2022, 2022.

My understanding from that paper is that MAE should be used for [Nd] (exponentially distributed) and RMSE should be used for  $\epsilon$ Nd (normally distributed). I would still recommend reporting both MAE and RMSE however, to facilitate comparisons with past and future models, and also because the distribution assumptions are not exactly satisfied with the GEOTRACES IDP21 data:

---

## Author Comment (AC2)

**Reply to reviewers' comments: egusphere-2022-606**

S Robinson *et al*.

Correspondence: ee14s2r@leeds.ac.uk

Title: Simulating marine neodymium isotope distributions using ND v1.0 coupled to the ocean component of the FAMOUS-MOSES1 climate model: sensitivities to reversible scavenging efficiency and benthic source distributions

**Summary of Changes**

Blue text below is our response to the reviewer's comments (reproduced in black). Line numbers refer to the tracked-changes version of the manuscript.

**Response to community comment: Tristan Vadsaria**

The following comment of the preprint research article entitled "Simulating marine neodymium isotope distributions using ND v1.0 coupled to the ocean component of the FAMOUS-MOSES1 climate model: sensitivities to reversible scavenging efficiency and benthic source distributions" by Robinson et al. has been motivated by the next implementation of the Neodymium oceanic cycle in the *i*LOVECLIM model.
This article describes the modelling implementation of the Nd oceanic cycle in the fast global climate model (GCM) FAMOUS. After selecting the most appropriate reference simulation in terms of oceanic variable (temperature, salinity, AMOC pattern and strength), the authors scrutinized in detail the result of the implementation based on two decisive parameters. These parameters, being the scavenging coefficient and the Nd flux from the sediment, are known to be still poorly constrained by observational studies. The modelling work provided by Robinson et al., in line with the recent Nd modelling studies performed with GCM, helps to provide more insight on these processes with the "own" physics and parameterization of the FAMOUS model.

I found that the paper is overall very well written with good descriptions of the results which is very important to understand the relative importance of both the scavenging coefficient and the Nd flux from the sediment on the global oceanic distribution of Nd and εNd. I particularly like the effort put into reaching the most "appropriate" simulation regarding ocean physics in order to reduce the associated bias for the Nd and εNd interpretation. I also join the authors about the need of a shared protocol for an Nd modelling intercomparison project.
Below are some (minor) comments to 1) have some clarifications on some points of the manuscript, 2) suggest some modifications if it is feasible in time and in resources for the authors.
Because my field of expertise is mostly climate modelling, my comments will not focus on the geochemistry part of the paper.
Best regards,
Tristan Vadsaria

We really appreciate the time taken for the community comments given and the interest in the presented study.

**River forcings**

For this point I am not sure at 100% that my comment is pertinent, sorry if I misunderstood the manuscript in this regard. As far as I understood from other Nd modelers, the river forcings in terms of (dissolved) Nd concentration and εNd come indeed from Goldstein and Jacobsen (1987) but is often updated with recent observations such as coming from Blanchet (2019). As I assumed that the authors are using the most updated data to force the model, would it not be better to indicate this in the manuscript (especially for the caption of figure 5)?

We can clarify this here: in our presented scheme the riverine Nd source only refers to the dissolved riverine flux, and hence we used the estimated dissolved Nd concentration and isotope ratios from Goldstein and Jacobsen (1987). Recent schemes have updated the riverine source with particulate riverine samples from Blanchet (2019) and Robinson et al (2021). However, as we explain in lines 471-476, the $\varepsilon_{Nd}$ of dissolved and particulate river inputs is highly variable, and combining them is non-trivial. We therefore chose to keep the dissolved riverine source separate, and the sediment flux imposed (which includes the most recent compilation of published $\varepsilon_{Nd}$ from river sediment samples deposited on the continental shelf and slope) encompasses at least in part the $\varepsilon_{Nd}$ from a river particulate flux.

**What about the seafloor Nd concentration forcing?**

This comment is not really a suggestion nor a critic since I think it's beyond the scope of the study but rather an open question, I think it would need more discussion from and for the Nd modelling community.

For the Nd sediment source, I understand that this study is in line with the previous scheme initiated by Rempfer et al. (2011) that "[…]do not make any assumption regarding the nature of the Nd boundary source" and "[…]do not assume spatial variation...", later followed by Gu et al. (2019) with the same depth limitation and also by Pöppelmeier et al. (2020) but without the depth limitation. This approach is indeed convenient for tuning "fsed" which is today still not very well known. However, how far is it reasonable to apply the same scheme while considering the whole seafloor, i.e., to not consider the spatial Nd concentration of the seafloor into the calculation of the sediment source (question also valid for Pöppelmeier et al., 2020)?

While the Nd sediment source was "confined" to the continental margin in Rempfer et al. (2011) it seemed more "reasonable" to make their assumptions especially regarding the horizontal resolution of the Bern3D model. However, now, without the depth limitation, I would guess that the spatial variations of the Nd concentration of the seafloor would have an impact on the deep and bottom Nd dissolved seawater distribution, don't the authors think so?

In Arsouze et al. (2009), "Fsed is then determined for both [143]Nd and [144]Nd isotopes by multiplying this sediment flux to the concentration along the margin" and they fixed the sediment flux to only one value, without the ability to make a lot of simulation to tune this flux because of the resolution and the time consumption of their model. My question is: would it be possible to have an intermediate approach between Rempfer et al. (2011) and Arsouze et al. (2009), e. g., tuning the flux while applying widely the Nd seafloor concentration (obtained from the recent data of Robinson et al. 2021 for instance)?

*Small edit: As I kept thinking about the previous point, I realized that Nd seafloor data was indeed scarcer than εNd and that extrapolating a wide Nd seafloor map would be less relevant especially in the Pacific Ocean (cf attached figure in supplement using Nd data provided by Robinson et al., 2021). Anyway, would it be possible to imagine a regional seafloor Nd (or basin-scale) signal such as used for the dust εNd?*

These are interesting discussion points that do warrant thorough discussion in the community, so we are glad that our manuscript has people talking about this.

First, we return to the start of the comment. Our study primarily aimed to present the new Nd isotope scheme in FAMOUS and explore key model parameters under the context of our current knowledge of marine Nd cycling. The reasoning for the globally uniform sediment flux applied across the seafloor was to explore the emerging benthic flux hypotheses that suggests that a widespread benthic Nd flux, of similar flux magnitudes, and across diverse sedimentary environments may dominate the marine Nd cycle. In this study we wanted to focus on where this is and where this isn't the case before making assumptions on what environmental conditions drive spatially diverse fluxes, especially since pore water data is currently scarce. We think that this is an important first step towards tackling the big open questions that the comment poses.

In response here, and to another reviewer's comments, we have added more discussion in the manuscript surrounding the spatial variability of elevated Nd fluxes and regions that may act as Nd sinks.

To explore in more detail the spatial variability in Nd sediment sources, although out of the remit of this study (as highlighted in the comment), will be very useful and we agree that greater community focus on describing the environmental regions which drive elevated benthic fluxes is necessary to achieve this. A regional mask to explore elevated benthic fluxes is a great suggestion for future work, for example building upon the work in Pöppelmeier et al. (2020) where core top-bottom water offsets were used to create a simple regional elevated benthic flux map.

**Less sensitive response of Pacific εNd to reversible scavenging efficiency - deep seafloor wide sediment source**
Concerning that point, the authors said that their results "[...] contrast with results from Rempfer et al. (2011) ..." (line 751) and "attribute this difference primarily to the spatial variation in the sediment Nd flux" (lines 752-753) due to the different modelling scheme used for the Nd sediment source.
Even though that explanation seems obvious, I would like to see a simulation output to confirm what the authors are suggesting (i.e., simulations governed by deep seafloor wide sediment source) with the "own" FAMOUS ocean dynamics: A set of simulations similar to the scavenging coefficient sensitivity simulations (first part of the result) but with the "initial" depth limitation of 3000km would be the best (but maybe too much for that purpose).
What about retaining the best simulation ("EXPT_RS4" as far as I understood) for the scavenging coefficient and run a parallel simulation with the depth limitation of 3000km for the sediment source? I think that putting only the result (a couple of 2d maps) of this new simulation in the supplementary compared with "EXPT_RS4", while keeping the original

text in the main paper would be enough. Taking this comment into consideration is obviously up to the authors regarding its feasibility.

We highlight our companion paper to this manuscript (in discussion: https://egusphere.copernicus.org/preprints/2022/egusphere-2022-937/), where we first present an optimized version of the Nd isotope scheme. In this companion paper, which builds on the present manuscript, we begin to explore spatial variations in our optimised scheme (rather than in EXPT_RS4) by assessing the 'margin' vs 'benthic flux' as suggested in the comment.

**Wrong residence time value?**
As explained in the main manuscript, the residence time is equal to the Nd inventory divided by the total Nd flux (line 598). Following table 4 (and 5), it corresponds to (column 5*1000/column 4). If I apply this, I found the same results as the authors for all the experiments except for "EXPT_RS1" which should be (10.6*1000/5.27) = 2011 years, am I right? Anyway, it does not change anything to the results description and the conclusion since it is still the simulation with the highest residence time.

Done: typo corrected in the table and text ([Nd] inventory for EXPT_RS1 is 16 Tg, yielding a residence time of 3036 years. Thank you!

**Very minor suggestion (1)**
"The total global flux of river sourced dissolved Nd to seawater (friver) is $4.4 \times 10^8$g(Nd) yr⁻¹": this is the value that comes from the simulated runoff in FAMOUS combined with the prescribed Nd river concentration, isn't it (as confirmed by looking at Table1)? In that case, I would suggest adding "in the model" to the sentence to enhance clarity.

Done.

**Very minor suggestion (2)**
In my opinion, the use of "pronounced" in line 642, 894 and 992 to describe the behavior of the vertical gradient of [Nd]d is not very descriptive, but I may be wrong since I am not a native English speaker.

Done. Updated throughout text to '*overestimate the vertical [Nd] gradient with depth, where simulated [Nd] is too low in the surface and too high at depth compared to seawater measurements.*'

**Very minor suggestion (3)**
Figures 11, 12, 15 and 16 are overall very nice but I would suggest, regarding the depth profiles, for more clarity and visibility, to not match the color of the observational data with the color bar of the central 2d map. Maybe rather use a unique color, also with the dots connected together, to reduce the confusion with the color of the simulated profiles (but at this stage it's really a matter of taste).

These plots display a lot of information of the spatial and vertical distributions of simulated and measured Nd. We explored updating the plot to have a single colour and joined scatter (see left panel in the plots below) for the discrete observational data, but find the extra detail makes the plot harder to read compared to our original (see right panel). We have chosen not

to update the plots in this instance. Furthermore, although the modelled data can be depicted as continuous profiles we did not want to impose a linear fit between the discrete measured data points especially for $\varepsilon_{Nd}$.

[Figure]

**Very minor suggestion (4)**
Why not merge Table 2 and Table S3? I think that Table S3 would be very informative in the main text.

Done.

---

## Author Comment (AC3)

**Reply to reviewers' comments: egusphere-2022-606**

S Robinson *et al*.

Correspondence: ee14s2r@leeds.ac.uk

Title: Simulating marine neodymium isotope distributions using ND v1.0 coupled to the ocean component of the FAMOUS-MOSES1 climate model: sensitivities to reversible scavenging efficiency and benthic source distributions

**Summary of Changes**

Blue text below is our response to the reviewer's comments (reproduced in black). Line numbers refer to the tracked-changes version of the manuscript.

**Response to reviewer 2:  Catherine Jeandel (Referee)**

The manuscript egusphere#2022-606 proposes the implementation of the oceanic cycles of the Nd isotopes (143 and 144) in the FAMOUS-MOSES1 climate model. There are interesting novelties in this work as 1) the use of the detailed epsNd map established by the author and comprising the bottom sediment signatures (Robinson et al, 2021); 2) exhaustive sensitivity test of two main parameters driving Nd and epsNd cycles: the reversible scavenging and the external flux, mostly the sediment one here. Actually, this represents a tremendous work; the manuscript is well written (although sometimes a bit wordy) and illustrated. It certainly deserves publication in *egusphere*. Nevertheless, I have some comments that I submit here to the authors.

We thank the reviewer for her kind comments recognising the effort of the work presented.

- Sediment flux vs Boundary Exchange (BE) processes

I think there is a misunderstanding or a confusion between these two terms that needs tobe clarified.

At several places in the manuscript, it is written that sedimentary flux is encompassing Boundary Exchange (e.g lines 250-255, around 345 but also 998-1000 and at other places highlighted in the manuscript) while to me, it's the opposite (ie BE is encompassing sedimentary flux, down to 3000 m -which is already deep!) in our preceding works

*"Seafloor sedimentary fluxes, an umbrella term that refers to a multitude of processes encompassing boundary exchange (Lacan and Jeandel, 2005), submarine groundwater discharge (Johannesson and Burdige, 2007), and a benthic flux released from pore waters (Abbott et al., 2015a), are simulated via a combination of a sedimentary source applied across sediment-water interfaces together with a separate sink occurring via particle scavenging"*

I suggest to write this paragraph differently (as well the other places where it's a bit confusing, identified in my direct comments in the pdf). Indeed, when we proposed the "BE Concept" with F. Lacan (EPSL, 2005), we did not pretend to describe any specific processes that occur at the land-ocean interface and more specifically along the margins because we could not differentiate them. Later, I listed the potential processes that could explain the "BE (Jeandel, 2016). In other words, "BE" broadly comprises all the processes that could release Nd from the solid to the liquid but also those which would scavenge it, more or less at the same time and in the same area (note that this comprises reversible scavenging too!). More recently, one of the conclusions of the PAGES-GEOTRACES workshop (2018) pushed by Martin Franck was to "kill the BE", in other words to disentangle these processes, among them the seafloor sedimentary fluxes (either through early diagenesis or dissolution of resuspended sediments), low temperature hydrothermalism, SGD, benthic fluxes etc...

The point here is that the authors removed the depth limitation of 3 km which was forcing the model to consider sedimentary fluxes along the margins only. But the sedimentary flux they consider are occurring everywhere including along the slopes. This does not mean that the Sed Flux (a specific mechanism) encompasses the BE (a broader concept).This just means that this flux is extended to the whole ocean in the proposed work.

Done: we have updated the definition of the boundary exchange in the introduction, and throughout the text to:

*'The term 'boundary exchange' was then coined to describe significant modification of Nd isotopic composition by the co-occurrence of Nd release from sediment and boundary scavenging, without substantially changing [Nd] (Lacan and Jeandel, 2005).'* (e.g. line 91-93).

- 'Top down' vs. 'bottom-up' processes (issue linked to what is discussed by Reviewer 1)

I'd cautiously use this opposition which was never clear to me (and I had long debates with B. Haley on this issue). By the way, the earliest Nd budgets proposed 2 sources: dissolved rivers and hydrothermal (see the historical works of Goldstein, O'Nions etc…). To my knowledge, hydrothermal is not "top down". Consider now the most recent budgets: most of them invoke "Boundary exchange" which includes processes that occur in the deep waters, down to 3000 m depth (see above), in other words they include "bottom processes". Thus, although I agree with what is written line 103 and after (reported below), it seems to me that this was not "new" because the *benthic* flux is occurring at any place where there is a contact between sediment and water, in other words everywhere from the beach to the deepest parts of the ocean. Again, there is a confusion between the processes and the location. Thus, I strongly suggest to the authors to be cautious here. What was "new" is that it could concern sediments below 3 km depth.

"Recent pore fluid concentration profiles measured on the Oregon margin in the Pacific Ocean indicate that there may be a *benthic* flux of Nd from sedimentary pore fluids, presenting a new, potentially major seafloor-wide source of Nd to seawater (Abbott et al., 2015b, a)."

Done: we have removed the 'top down' vs 'bottom up' statements from the manuscript and replaced with more detail on the processes under each statement. For example, line 1604-1607: '*This could be interpreted as evidence against a globally widespread benthic flux driven model of the marine Nd cycle with a spatially constant flux across diverse sedimentary environments in favour of the more distinct [Nd]$_d$ distributions that may be achieved under a model of marine Nd cycling with larger and more heterogenous surface and near surface Nd sources, and a greater dominance of reversible scavenging*'

We have updated the sentence mentioned to be clear that recent benthic flux measurements have led to the suggestion that the sedimentary fluxes are no longer limited to the continental margins, lines 142-144: '*Recent pore fluid concentration profiles measured on the Oregon margin in the Pacific Ocean indicate that there may be a benthic flux of Nd from sedimentary pore fluids, presenting a potential major seafloor-wide (i.e., no longer limited to the continental margins) source of Nd to seawater (Abbott et al., 2015b, a).*'

▪ The choice of the Ndp/Ndd ratio to conduct the Fsed sensitivity test.

I did not understand why the authors did not kept the value of 0.004 instead of that of 0.003 to do these sensitivity tests. Indeed, as underlined in Table 5, the value of 0.003 leas to residence times larger than 1000 y, leading to more moderate range of 40 years difference in the *f*sed simulations Nd

I did not see a clear justification of this choice in section 3.1 and would be keen to see the same sensitivity tests but with the Ndp/Ndd ratio of 0.004, which was the most consistent with the data.

Done: we have added further clarification to our explanation of this in lines 827-834. To paraphrase: due to the computational demands, all simulations were run in parallel, and hence we had to decide the $[Nd]_p/[Nd]_d$ value before we had the results from the $[Nd]_p/[Nd]_d$ sensitivity experiment. Our choice of 0.003 is in the middle of the range (0.001-0.006) identified by previous modelling studies (Rempfer et al., 2011; Arsouze et al., 2009; Gu et al., 2019) and direct observations (Jeandel et al., 1995; Stichel et al., 2020; Zhang et al., 2008; Lagarde et al., 2020; Paffrath et al., 2021). Now that we have the results from the $[Nd]_p/[Nd]_d$ sensitivity experiment, we can see that 0.004 would have been a better choice, and propose that future work to refine model performance begins with this.

• **The discussion on the reasons leading to epsNd modelled profiles that do not fit the data** is often too shy and not clear enough. Perhaps the sedimentary flux is too strong? Or the choice to attribute a constant flux for the deepest (bottom) and shallowest sediment (margin) is not appropriate? This is well exemplified by Figure 17 and the discussion lines 930-950. What would happen if the Fsed would allow differentiating the strength of the SedFlux deposited on the margin (fresh deposits fromrivers, easy to remobilize) vs that of the bottom (too strong, "counterproductive" as it is written line 942)?

Done: also to address a similar comment from Reviewer 1, we have updated the discussion to include additional arguments to explain model-data offsets, including suggestions that the too low [Nd] in the surface may be a result of too weak surface sources (dust, rivers and continental margins), and too high [Nd] at depth may be due to too strong sediment source

(especially due to the globally uniform Nd source escaping all sediment-water interfaces).

- **Minor comments:** they are highlighted in the attached pdf. I also identified some unit issues and rare typos but they are already listed by B. Pasquier (who I thank for the exhaustive list!).

  Done: many thanks – we have addressed these.

As a whole, I'd also suggest to the authors to shorten the manuscript by 10%-15% if possible.

Done: we have shortened the main text in the revised version by 10 % (cut >1,200 words) following revisions to focus the presented points.

Please also note the supplement to this comment:
https://egusphere.copernicus.org/preprints/2022/egusphere-2022-606/egusphere-2022-6

Thank you - we have used these annotations to correct and revise the manuscript.

---

## Author Comment (AC4)

**Reply to reviewers' comments: egusphere-2022-606**

S Robinson *et al*.

Correspondence: ee14s2r@leeds.ac.uk

Title: Simulating marine neodymium isotope distributions using ND v1.0 coupled to the ocean component of the FAMOUS-MOSES1 climate model: sensitivities to reversible scavenging efficiency and benthic source distributions

**Summary of Changes**

Blue text below is our response to the reviewer's comments (reproduced in black). Line numbers refer to the tracked-changes version of the manuscript.

**Response to reviewer 3:  Ed Hathorne (Referee)**

This paper describes the results of a model of the marine Nd cycle implemented in the ocean part of the fast climate model derived from the Hadley center GCM. The use of such a model for simulating the Nd isotopes of seawater is a useful development as the fast run times allow more experiments to be conducted, although care must be taken that the ocean circulation is resolved correctly as this is what we hope to trace with Nd isotopes. From the standpoint of a geochemical oceanographer this paper is very interesting because it directly tests the hypothesis that the distribution of Nd isotopes in seawater is mostly controlled by a flux from marine sediments, sometimes known as the "bottom flux hypothesis". Along with other models of the marine Nd isotope cycle published very recently, this work affords many insights into the processes that are likely, and unlikely, to control the distribution of Nd isotopes in seawater. The discussions paper is rather long but with some editing, clarification in places and discussion of the other very recently published works, I would gladly recommend this for publication.

We thank the reviewer for his positive comments. His and other reviewers' specific comments have been valuable for making the presentation of this work even stronger in the revised manuscript.

This could be an important contribution as the authors have the most up to date data compilation available, but this should be utilised throughout. For example, in Figure 9 the global marine Nd inventory of 4.2 x10^12 g from Tachikawa et al. (2003) is used to assess which reversable scavenging scenarios are realistic. Although this ground breaking study is clearly still relevant, many samples have been taken and measured in the intervening decades as shown in Figure 8. Would it not make sense to estimate the marine Nd inventory with all the available data? Perhaps it will make little difference but in 2003 there were very few data available for the entire Southern Ocean and North Pacific (Table 1 in Tachikawa et al., 2003).

This is an interesting point, and we fully support the suggestion to calculate a new marine Nd inventory with all the available data, updating the estimate made by Tachikawa et al. in

2003. However, to do this robustly is a big piece of work that goes beyond the data compilation and model results presented here. This is because it requires precise characterization of water mass Nd with a sophisticated interpolation between discrete point measurements that accurately distinguishes (in 3D space) the different water masses of the global ocean (and some oceanic regions remain sparsely measured). A potentially useful way to undertake this activity would be within the framework of a multi-model intercomparison, using the output from multiple models all running the same careful experiment design in combination with the observations to characterise global ocean Nd. This could even be achieved without activating an Nd isotope scheme, for example, using the ensemble of CMIP6 historical simulations to identify distinct water mases and their mixing, and combining this ocean structure with our presented catalogue of seawater measurements. Such a substantial undertaking would be a very useful piece of research, but it is beyond the scope of our study.

In light of this, we will continue to use the estimate by Tachikawa et al. (2003) as the best available 'target' [Nd] inventory for evaluating our global budget. However, we can also report an independent estimate of the global Nd inventory, which we derive from our best performing simulation for [Nd], EXPT_SED2, based on it returning the lowest MAE and RMSE with respect to our newly compiled [Nd] database. Furthermore, we can provide a basin-by-basin breakdown of that budget, noting that the total [Nd] from this best performing simulation is likely an underestimate because of missing marginal seas in our model. Whilst subject to the model's limitations, this estimate does have the advantage of having used FAMOUS to undertake the complex task of producing a globally continuous marine Nd field, and it does make use of the updated catalogue of observations, since this is what has been used to evaluate the model's performance and identify the 'best' simulation.

We have edited the text (Section 3.2, lines 1375-1386) and added Table 4 (shown below) to the main text to include this new result.

| Ocean Region | Nd inventory (Tg) |
|---|---|
| Global | 3.89 |
| Arctic Ocean | 0.05 |
| North Atlantic | 0.33 |
| South Atlantic | 0.45 |
| North Pacific | 0.96 |
| South Pacific | 0.76 |
| Indian Ocean | 0.63 |
| Southern Ocean | 0.28 |

Using these realistic reversable scavenging values (can it please be clarified if this is also 100% released like in Tachikawa et al., 2003?) a very simple universal sediment flux is tested.

In the scheme, biogenic particles follow dissolution profiles, when the particles dissolve 100% of the Nd associated with the particles is released back to seawater.

Although it is very interesting that this fails to simulate the tails of the observed data, both

radiogenic in the Pacific and unradiogenic in the N Atlantic, this is not proof that the bottom flux hypothesis is wrong. Assuming a constant flux over the entire ocean bottom is clearly unrealistic and this point should be clearly stated. With rare earth element concentrations >2 times that of shale, the red clay sediments covering large parts of the abyssal Pacific (e.g. Kato et al., 2011, Nature Geoscience 4) are most likely a sink for Nd. Here and also in areas influenced by hydrothermal particles (German et al., 1990, Nature 345, 516-518) the bottom flux is likely to be negative. The fact that Pasquier et al. (2022) use a parameterisation which increases the sediment flux at both radiogenic and unradiogenic extremes of sediment composition should be mentioned in the context of a constant bottom flux not simulating the highest and lowest seawater values.

Done: we have modified the text to clarify our discussion, pointing to the literature and other modelling studies to explore the need for more precise constraints on the benthic source, highlighting the diverse sedimentary regions/environmental conditions that may drive enhanced benthic fluxes or act as effective sinks. See lines 1555-1559 for a discussion on the benthic processes in Pacific red clays.

Detailed comments and suggestions are provided in an annotated PDF. I still hope publishers will provide a tool for extracting comments from PDFs. Please also note the supplement to this comment: https://egusphere.copernicus.org/preprints/2022/egusphere-2022-606/egusphere-2022-6 06-RC4-supplement.pdf

Thank you! We have used the annotations to improve the text throughout.

---

## Author Comment (AC5)

**Reply to reviewers' comments: egusphere-2022-606**

S Robinson *et al.*

Correspondence: ee14s2r@leeds.ac.uk

Title: Simulating marine neodymium isotope distributions using ND v1.0 coupled to the ocean component of the FAMOUS-MOSES1 climate model: sensitivities to reversible scavenging efficiency and benthic source distributions

**Summary of Changes**

Blue text below is our response to the reviewer's comments (reproduced in black). Line numbers refer to the tracked-changes version of the manuscript.

**Response to reviewer 4:  Torben Stichel (Referee)**

This discussion paper by Robinson et al. discusses the incorporation of Nd (isotopes and elemental concentration) into FAMOUS GCM's ocean component to better understand the GLOBAL marine Nd cycle. I really like the paper and it addresses the current debate on the direction of control in Nd (and REE) distribution. Acknowledging the paper has set its focus on the sensitivity of the scavenging efficiency and benthic fluxes, it leans towards comparing itself with previous modelling studies (Rempfer et al., Siddall et al., Pöppelmeier et al). The overall reality test is done by comparing their results with a global data base. They conclude that reversible scavenging is important for the Atlantic-Pacific gradient in eNd, but again the modelled Pacific Ocean does not match the observed data there. They admit that a global constant sediment flux in the model runs could be the issue here as the Pacific Ocean supposedly provides more reactive material (young, mafic rocks) than the Atlantic, which would support different sediment fluxes within these basins. As a non-modeller, I would like to avoid evaluating the technical parts of this paper, but I would like to highlight some important aspects on the biogeochemical cycles of Nd. Overall, I found the paper very well written, a bit wordy though.

We thank the reviewer for his positive and constructive comments, which have helped us to revise the manuscript. We have also shortened the text.

I do admit, I am bit surprised that recent particle studies (e.g. Lagarde et al. 2020, Paffrath et al., Stichel et al. 2020) were very marginally used in this paper. From those studies, we now have information on eNd in particles, different mineral fractions, different kDs etc. from the same locations as the dissolved fraction. We also know that pNd/dNd unfortunately is not uniform in the ocean. Also, in the last paragraph of the discussion (lines 927 ff.), where the authors compared their model outcome with observations in the North Atlantic – the area where the aforementioned papers have their study area – the composition of particles would help to assess the NADW composition (e.g. fig.6 in Stichel et al. 2020). In the search for end member composition, the authors might want to consider the very dynamic particle composition in that area (pointing towards different sources) and not necessarily from bulk sediments. If I am not mistaken, in those papers the observational

pNd/dNd are often one order of magnitude higher in the North Atlantic, compared to the global average assumed in the modelling studies (0.001 to 0.006). Or do I miss something here? Of course, it is reasonable to assume that Pacific pNd are very much lower than in the North Atlantic and therefore the pNd/dNd is very much skewed towards lower values. This discrepancy should at least be mentioned and justified.

Done: we have included these key studies and discussion points within the manuscript (e.g., lines 1008-1010), including highlighting and discussing the simplified assumption of a globally uniform $[Nd]_p/[Nd]_d$, which is mainly a pragmatic choice in the model development.
We have also updated the discussion on the North Atlantic to include information on the provenance of particles: lines 1008-1015.

The suggested studies are important, reporting measured Kd values for Nd, alongside measuring $[Nd]_p/[Nd]_d$ in the ocean. The reported $[Nd]_p/[Nd]_d$ values in the suggested studies are a magnitude higher compared to the global average reported in modelling studies. Unfortunately, a direct comparison between the modelled and measured $[Nd]_p/[Nd]_d$ is difficult to make, here we explain why. In the observed values reported, it appears that all particulate matter is totally digested, which includes detrital particles. Thus, the measured particulate Nd concentrations cannot be compared to simulated particulate concentrations directly, because we define these in our scheme (as do other similar schemes) as the adsorbed/authigenic fraction only. This difference is most clearly visible in the bottom most stations that sampled benthic nepheloid layers. The nepheloid layers can have very high detrital particle concentrations, which explain the [Nd]p/[Nd]d ratios in these samples. "True" (or adsorbed only particulate fraction) [Nd]p/[Nd]d ratios of Nd would therefore be lower. For comparison, the particulate to dissolved ratio of Th, which is much better studied and has a greater particle affinity to particles should be between 1 and 5%.

I would like to point out another rather minor issue, which is the database used. I acknowledge that with the now very impressive global Nd data sets available, it is very convenient to cite the GEOTRACES IDP. However, the authors want to double-check whether data actually IS in the data product. For instance, large parts of the eNd from GA03 (or US-GEOTRACES North Atlantic Zonal Section) are not included in the IDP 2021 but was used a citation here. For those data sets you can find the correct citations here: http://data.bco-dmo.org/jg/info/BCO/GEOTRACES/NorthAtlanticTransect/Nd_GT10%7Bdir =data.bco-dmo.org/jg/dir/BCO/GEOTRACES/NorthAtlanticTransect/,data=data.bco-dmo.or g:80/jg/serv/BCO/GEOTRACES/NorthAtlanticTransect/Nd_GT10_v8_joined.html0%7D?. I apologise for this rather shameless self-advertisement…

We think there is a misunderstanding here: the data sources highlighted by the reviewer are all included in the previous compilations referenced in our manuscript (Osborne et al., 2017, 2015; GEOTRACES Intermediate Data Product), as described in line 706-710:
*'The observational data used in this assessment are from the seawater REE compilation used by Osborne et al. (2017, 2015), augmented with more recent measurements including data in the GEOTRACES Intermediate Data Product 2021 (GEOTRACES Intermediate Data Product Group, 2021) from GEOTRACES cruises (GA02, GA08, GP12, GN02, GN03 and GIPY05).'* and lines 715-718: *'The location and spatial distribution of all observational records used in this study are shown in Fig. 8, and full details of the*

*seawater compilation including a full list of all the references for the data sources are provided in Supplementary Information: Table S3.'*

We realise that citing the full list of original data sources and methods papers would be preferable, but such a list comprises too many references (141 in Table S4) to all be included here. Hence in the main text we cite the published compilations that include the original studies – Osborne et al., (2017, 2015), which includes the $\varepsilon_{Nd}$ from GA03 (e.g. Stichel et al., 2015*), plus the additional cruises in the GEOTRACES Intermediate Data Product 2021 (which adds in $\varepsilon_{Nd}$ from cruises GA02, GA08, GP12, GN02, GN03 and GIPY05) – and in Table S3 we have attempted to adequately reference the original work. It would be better if journals had a way of indexing original data sources when a data compilation is cited in order to correctly attribute the credit (e.g. if we could link the citations in our Table S4), and we would certainly welcome any innovation to implement this.

*Stichel, T., Hartman, A., Duggan, B., Goldstein, S.L., Scher, H. and Pahnke, K. (2015). Separating biogeochemical cycling of neodymium from water mass mixing in the Eastern North Atlantic. Earth and Planetary Science Letters, 412:245-260.
The other two citations included at the referenced web page (Hartman et al. and Duggan et al.) are listed as 'in prep', and we have been unable to find them elsewhere (e.g. with a Google scholar title or author search)

I fully support the publication of this paper eventually. It is an important work and will be key for a better understanding of the Nd cycle as it is one of (if not) the most complete modelling papers for marine Nd isotopes and concentrations. The supplement's profound eNd/YREE data set is also great! Thanks for providing this with your publication.

Thank you very much for the positive comments, a lot of work has gone into this, and it is gratifying to see it so well received.